# Frequency-Aware Perceptual Optimization for Low-Complexity Implicit Image Compression

**Haotian Wu** [1 2]  **Gen Li** [2]  **Di You** [2]  **Pier Luigi Dragotti** [2]  **Deniz Gündüz** [2]

## Abstract

We propose a frequency-aware perceptual optimization framework for low-complexity image compression, realized as a ***Re**alism-enhanced **Re**gion-based Implicit Codec* (Re²IC). Re²IC models visual perception via saliency-guided region partitioning and local–global perceptual modulation. To enhance realism under complexity constraints, we introduce wavelet–Wasserstein distortion (WA-WD), a frequency-decomposed perceptual distortion that balances fidelity and realism through subband-aware modeling and provides a more reliable approximation than standard Wasserstein distortion. Together, these designs enable fine-grained spatial–spectral optimization, allowing Re²IC to achieve superior rate–perception trade-offs, outperforming generative codecs such as HiFiC while using less than $1\%$ of their decoding cost. Extensive experiments demonstrate state-of-the-art perceptual performance among overfitted codecs. Beyond compression, WA-WD serves as a standalone, tunable perceptual metric with strong alignment to human preference (Pearson 94.6%, Spearman 92.3%) and competitive performance across multiple IQA benchmarks. Project page: https://eedavidwu.github.io/ReReIC/

## 1. Introduction

Image compression has long relied on hand-crafted transforms to capture statistical correlations and perceptual redundancies (Sullivan et al., 2012; Bross et al., 2021). Recent deep learning approaches replace these with neural networks (Ballé et al., 2018; Cheng et al., 2020; He et al.,

2022), achieving strong rate–distortion (RD) gains. However, optimizing RD alone often fails to preserve perceptual quality, especially at low bit rates, which has spurred growing interest in perceptual compression.

Perceptual quality, or "realism" (Hamdi et al., 2025; Qiu et al., 2024), measures how closely reconstructions resemble natural images. Prior work (Blau & Michaeli, 2019) shows a fundamental trade-off between rate, distortion, and perception, implying that optimizing RD alone cannot guarantee naturalness. To bridge the gap, recent learned codecs tend to shift from deterministic decoding to sampling from conditional distributions with generative priors. GAN (Mentzer et al., 2020) and diffusion-based approaches (Theis et al., 2022) produce compelling reconstructions but often rely on complex models and large-scale training, exceeding the cost of commercial codecs. Achieving perceptual quality at low complexity, therefore, remains an open challenge.

Meanwhile, implicit codecs provide a low-complexity alternative by fitting each image with compact implicit neural representations (INRs) (Dupont et al.; Ladune et al., 2023; Wu et al., 2025), achieving promising RD performance with minimal decoding cost. However, their perceptual potential remains largely unexplored. Recent studies show that Wasserstein distortion (WD) (Qiu et al., 2024) improves rate–perception (RP) trade-offs by unifying fidelity and realism through saliency-weighted WD optimization. However, practical WD estimation (Ballé et al., 2025) typically relies on independence assumptions across features, neglecting spatial correlations. In addition, saliency-only weighting captures spatial importance but overlooks regional structure and frequency-dependent visual characteristics. As a result, it may over-resample textures in smooth, non-salient areas (Fig. 31) or enforce excessive fidelity in high-frequency salient regions (Fig. 32). Moreover, further realism gains typically require more complex INRs or costly optimization. These limitations highlight the need for a new framework that explicitly incorporates regional structure and frequency-aware saliency to achieve perceptual quality at low complexity.

Human visual perception is spatially and spectrally non-uniform, with high acuity in the fovea, reduced sensitivity in the periphery, and attention biased toward salient, frequency-

---

[1]College of Electrical Engineering, Zhejiang University, Hangzhou, 310027, China. [2]Department of Electrical and Electronic Engineering, Imperial College London, London SW7 2AZ, U.K. Correspondence to: Di You <dy22@ic.ac.uk>.

*Proceedings of the $43^{rd}$ International Conference on Machine Learning*, Seoul, South Korea. PMLR 306, 2026. Copyright 2026 by the author(s).

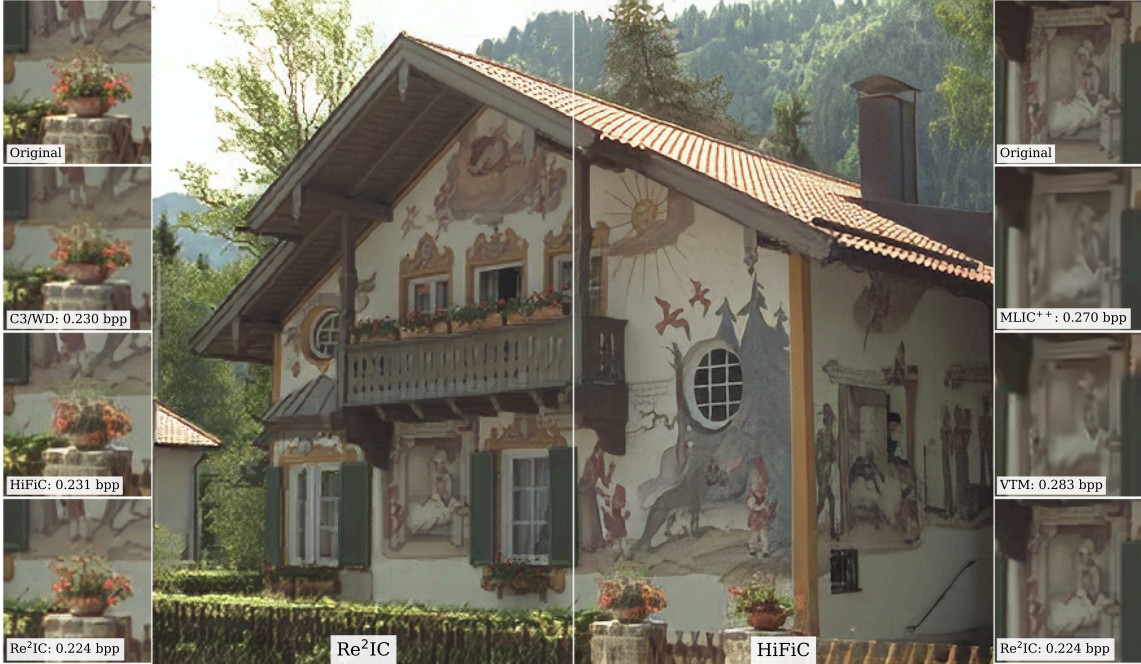

*Figure 1.* Visual comparison between codecs: Re$^2$IC delivers the best perceptual realism, closely matching the original image, while C3/WDs and HiFiC can exhibit artifacts. Even with higher bit budgets, codecs such as VTM and MLIC$^{++}$ lose texture. Much more static and dynamic comparisons across methods, bitrates, and datasets are provided in the Appendix and our project page.

specific content. Motivated by this, we propose Re$^2$IC, a ***Re**alism-enhanced **Re**gion-based **I**mplicit **C**odec*, that models visual perception in a region-wise manner. Re$^2$IC introduces two main innovations: (1) *Region-wise perceptual modeling*: improving RP performance with faster convergence and cheaper decoding; and (2) *Wavelet–Wasserstein control*: decomposing WD into subbands to balance fidelity and realism for fine-grained optimization.

Rather than overfitting the entire image with a single network, Re$^2$IC partitions it into salient regions, each modeled by a dedicated local perceptual network (LPN). Arbitrary region boundaries are efficiently encoded using low-cost chain coding (Chang et al., 1992), with the decoded contour for identifying the corresponding LPN. A global perceptual modulator (GPM) is introduced to provide global context to mitigate artifacts from purely local modeling. This design simplifies coding, accelerates convergence, and keeps each LPN lightweight, thereby lowering both encoding and decoding costs. Finally, Re$^2$IC decomposes perception modeling into wavelet subbands, enabling explicit control over fidelity and realism across frequencies. The multi-scale wavelet structure naturally supports WD approximation, while saliency-guided band adaptation enables fine-grained bit allocation that accounts for both regional structure and frequency characteristics, achieving perceptually enhanced compression at low complexity (see Fig. 1).

Compared with leading neural codecs (Jiang et al., 2025;

Fu et al., 2024), generative-based methods (Mentzer et al., 2020), diffusion-based methods (Yang & Mandt, 2023), and prior overfitted codecs (Ballé et al., 2025), Re$^2$IC achieves higher perceptual realism at substantially lower decoding complexity, delivering superior RP performance across user studies and various quantitative metrics. Notably, Re$^2$IC surpasses HiFiC with less than $1\%$ of its decoding cost. By combining wavelet–Wasserstein optimization with local–global perceptual modulation, Re$^2$IC enables fine-grained fidelity–realism control, faster convergence, and low-complexity decoding, challenging the long-standing trade-off between perceptual quality, speed, and efficiency.

Our main contributions are summarized as follows:

- We propose Re$^2$IC, a realism-enhanced implicit codec that models visual perception *region-wise*, achieving strong RP performance while maintaining low decoding complexity.

- Imitating human perception, Re$^2$IC integrates saliency-based partitioning with local–global perceptual modulation, and applies wavelet–Wasserstein optimization for fine-grained spatial–spectral control, enabling flexible bit allocation and efficient coding.

- Extensive experiments and user studies demonstrate that Re$^2$IC achieves state-of-the-art RP performance among overfitted codecs, surpassing HiFiC while requiring substantially lower decoding cost.

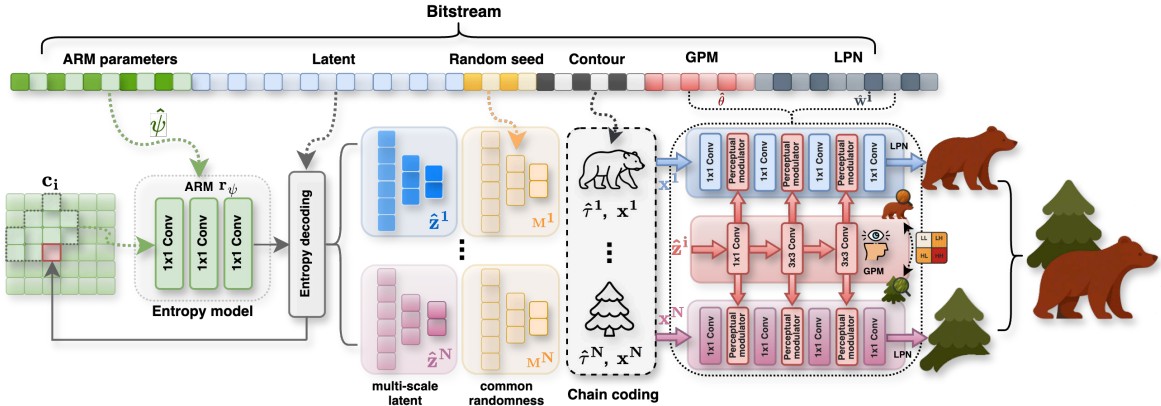

*Figure 2.* Workflow of Re$^2$IC: the image is partitioned into $N$ regions, each reconstructed by a local perceptual network. A shared global perceptual modulator provides global modulation for perception compensation, using latent vectors decoded from a shared entropy model.

- We introduce wavelet–Wasserstein distortion as a standalone, tunable perceptual metric, which adapts to diverse scenarios and aligns strongly with human ratings (PCC 94.6%, SRCC 92.3%). Additional image quality assessment experiments further confirm its leading performance as a general perceptual metric.

## 2. Related work

**Implicit codec.** Early work, such as (Dupont et al.) trained INRs with quantized parameters forming the bitstream, later extended via meta-learning (Dupont et al., 2022a). COM-BINER (Guo et al., 2023) improved RD with variational INRs and relative entropy coding. COOL-CHIC and successors (Ladune et al., 2023; Kim et al., 2024) introduced learnable latents, entropy models, and architectural advances, surpassing HEVC (Sullivan et al., 2012). More recently, LotteryCodec (Wu et al., 2025) and MoRIC (Li et al., 2026) employ random networks and region-wise coding for better RD performance, and (Ballé et al., 2025) incorporated saliency-guided Wasserstein distortion for RP optimization. Nevertheless, perception-oriented overfitted codecs remain underdeveloped and fall short of generative methods, underscoring the need for realism-enhanced yet low-cost designs.

**Region partitioning for INRs.** INRs model signals as continuous functions and benefit greatly from spatial decomposition, as shown by input partitioning (Jiang et al., 2020; Tretschk et al., 2020), local latent modulation (Mehta et al., 2021), and coarse-to-fine optimization strategies (Liu et al., 2023; Ashkenazi & Treister, 2024). In the compression setting, although region- and shape-based coding attracted research interest in the past (Schmalz, 2004), its adoption was constrained by the difficulty of designing transforms for arbitrary shapes (Sikora et al., 1995; Li &

Li, 2000). In contrast, INRs naturally support arbitrary shapes by training directly on irregular regions, which can be well-suited for modeling non-uniform human perception, enabling perception-enhanced region-based coding.

**Perception neural codec.** Early realism-enhanced codecs employed GANs such as HiFiC (Mentzer et al., 2020) and MS-ILLM (Muckley et al., 2023), as well as latent generative models (Jia et al., 2024), to synthesize visually plausible reconstructions at low bitrates. More recent methods leverage diffusion priors (Yang & Mandt, 2023; Hoogeboom et al., 2023; Ohayon et al., 2025) and text-to-image generation capabilities (Careil et al., 2024; Jiang et al., 2023; Lee et al., 2024) for strong realism at extremely low bpp. However, these approaches require orders-of-magnitude higher computation than commercial codecs, limiting their practicality. This motivates a realism-enhanced implicit codec that models perception directly, rather than learning a generative distribution, achieving competitive RP performance at far lower decoding complexity.

## 3. Methodology

### 3.1. Preliminary.

**Region-based overfitted codec.** Re$^2$IC adopts a region-based perceptual coding framework, partitioning each image into $N$ regions $\{S^1, \ldots, S^N\}$, each defined by a contour $\tau^i$ and spatial coordinates $x^i$. Each region is independently modeled using a dedicated INR $f_{W^i}$ and a latent code $z^i$, both quantized and entropy-coded (see Fig. 2). Contours $\tau^i$ are explicitly compressed using chain coding. During decoding, they guide the selection of $\{x^i, z^i, W^i\}$ for region-wise reconstruction via $\hat{S} = \bigcup_{i=1}^{N} f_{\hat{W}^i}(x^i, \hat{z}^i)$, where $\bigcup$ denotes the merging of all regions, $\hat{W}^i$ is the quantized

parameters, and $\hat{z}^i$ is the quantized latent codes. The perceptual distortion and the rate cost can be expressed as:

$$P = \mathbb{E}_{\boldsymbol{S} \sim p_s}\Big[ d(\boldsymbol{S}, \hat{\boldsymbol{S}}) \Big],$$

$$R = \mathbb{E}_{\boldsymbol{S} \sim p_s}\Bigg[ \sum_{i=0}^{N} \Big( -\log_2 p_{\hat{\psi}}(\hat{\boldsymbol{z}}^i) $$
$$- \log_2 p(\hat{\boldsymbol{W}}^i) - \log_2 p(\boldsymbol{\tau}^i) \Big) \Bigg] + R_{\hat{\psi}}, \quad (1)$$

where $d$ denotes perceptual distortion, $p_{\hat{\psi}}$ is a shared entropy model for all $\hat{\boldsymbol{z}}^i$, and $R_{\hat{\psi}}$ is the associated parameter cost.

**Wasserstein Distortion.** WD (Qiu et al., 2024) measures perceptual differences in feature space by explicitly modeling *foveal* and *peripheral* sensitivity through a spatially varying $\sigma$-map. Derived from saliency, this $\sigma$-map modulates tolerance to *texture resampling*, thereby enabling *flexible, perception-guided* bit allocation in image compression. Formally, local WD for a feature map $\boldsymbol{z}$ at location $(x, y)$ is defined as the 2-Wasserstein distance between feature distributions within a pooling window, where the underlying WD is computed with respect to a feature space distortion measure $d(\cdot)$. The pooling weights follow a family of probability mass functions (PMFs) $q_\sigma(k)$ parameterized by the pooling width $\sigma(x, y)$. This yields locally pooled measures $\boldsymbol{y}_\sigma$ over which WD is evaluated, and the final distortion is obtained by spatial averaging across all locations. For efficiency, $\boldsymbol{y}_\sigma$ are often approximated as *independently distributed Gaussians* (Olkin & Pukelsheim, 1982). Under this approximation, the local WD between each feature component $z_i$ and its reconstruction $\hat{z}_i$ is: $D_{i,\sigma} = (\mu_i - \hat{\mu}_i)^2 + \left(\sqrt{V_i} - \sqrt{\hat{V}_i}\right)^2$, where $\mu_i = \sum_k q_\sigma(k) z_{i+k}$, and $V_i = \sum_k q_\sigma(k) \left(z_{i+k} - \mu_i\right)^2$ denote the local mean and variance of each element, with $k \in \mathbb{Z}$ indexing spatial offsets within the pooling window determined by $\sigma$-map. The final WD is averaged over all locations $\mathcal{I}$ as: $WD(\boldsymbol{z}, \hat{\boldsymbol{z}}) = \frac{1}{|\mathcal{I}|} \sum_{i \in \mathcal{I}} D_{i,\sigma}$. Detailed derivations, interpretations, theoretical properties, and practical implementations are provided in Appendix E.1.

### 3.2. Wavelet Wasserstein Distortion

While WD provides a foundation for perceptual compression, its practical computation often relies on independence assumptions that can be restrictive in neural feature spaces. Moreover, saliency-only optimization emphasizes spatial importance but does not explicitly capture regional structure or frequency-dependent perceptual characteristics. These limitations motivate wavelet-WD (WA-WD), a more interpretable and perception-consistent modeling approach for practical perceptual optimization. Detailed motivations, comparisons, theoretical properties, and validations are given in Appendix E.2.

**Formulation.** WA-WD measures perceptual differences in a frequency-decomposed feature space, producing subband-aware distortion that enable fine-grained control and more flexible bit allocation. Specifically, WA-WD applies an orthonormal discrete wavelet transform (DWT) to decompose feature maps into orthogonal subbands and then computes WD. This absorbs intra-block correlations into subband-specific variances, making the Gaussianized independence assumption more accurate. Subband-wise evaluation naturally disentangles fidelity (LL) from realism (LH/HL/HH), and multi-level decompositions further decorrelate structural and textural statistics.

Consider a $2 \times 2$ feature block from the source image $\mathbf{x}$ with vectorized form $\mathbf{w}_i \in \mathbb{R}^4$. The Haar transform projects this block onto basis vectors $\boldsymbol{h_b} \in \mathbb{R}^4$, yielding $z_i^b = \boldsymbol{h_b}^\top \boldsymbol{w_i}$, with $b \in \{LL, HL, LH, HH\}$. Given the pooling PMF $q_\sigma$, the resultant pooled mean and variance are: $\mu_i^b = \sum_k q_\sigma(k) z_{i+k}^b = h_b^\top \left( \sum_k q_\sigma(k) \boldsymbol{w_{i+k}} \right) \triangleq \boldsymbol{h_b}^T \bar{\boldsymbol{w}}_i$, $V_i^b = \sum_k q_\sigma(k) \left( z_{i+k}^b - \mu_i^b \right)^2 = \boldsymbol{h_b^T C_i h_b}$, where $\boldsymbol{C_i} \triangleq \sum_k q_\sigma(k)(\boldsymbol{w_{i+k}} - \bar{\boldsymbol{w}}_i)(\boldsymbol{w_{i+k}} - \bar{\boldsymbol{w}}_i)^\top$ denote a pooled $4 \times 4$ *block covariance matrix*. Similarly to WD, with independence assumption, the per-band WD and the overall WA-WD are given as: $D_{i,\sigma}^b = (\mu_i^b - \hat{\mu}_i^b)^2 + \left(\sqrt{V_i^b} - \sqrt{\hat{V}_i^b}\right)^2$ and $WD_{\mathrm{wave}}(\boldsymbol{z}, \hat{\boldsymbol{z}}) = \frac{1}{4} \sum_{b \in \{LL, LH, HL, HH\}} \frac{1}{|\mathcal{I}_b|} \sum_{i \in \mathcal{I}_b} D_{i,\sigma}^b$, respectively. We use equal subband weights by default for simplicity, though they can be tuned for improved perceptual evaluation or different degradations and bpp regimes, as detailed in the later section.

**Theoretical properties.** A core design principle of WA-WD is to maintain the desirable theoretical guarantees of WD while improving perceptual alignment and region-wise controllability. Since DWT is orthonormal and invertible, applying WD in wavelet-domain preserves metric structure.

**Theorem 3.1** (Metric preservation). *Let $\boldsymbol{W} \in \mathbb{R}^{d \times d}$ be an orthonormal DWT and define $WD_{\mathrm{wave}}(\boldsymbol{z}, \hat{\boldsymbol{z}}) \triangleq WD(\boldsymbol{Wz}, \boldsymbol{W\hat{z}})$. Then: (1) If the PMFs $q_\sigma$ have no spectral nulls and $d(\cdot)$ is a metric (i.e, $WD(\boldsymbol{z}, \hat{\boldsymbol{z}})^{1/p}$ is a metric), then $WD_{\mathrm{wave}}(\boldsymbol{z}, \hat{\boldsymbol{z}})^{1/p}$ is a metric. (2) If the feature extractor $\phi(\cdot)$ is invertible (i.e, $WD(\boldsymbol{x}, \hat{\boldsymbol{x}})^{1/p}$ is a metric), then $WD_{\mathrm{wave}}(\boldsymbol{x}, \hat{\boldsymbol{x}})^{1/p}$ is also a metric.*

Theorem 3.1 shows that WA-WD preserves the metric structure as a perceptual measure (with proof in Appendix B). Beyond this, its diagonal-Gaussian approximation bound becomes tighter after an orthonormal DWT due to its decorrelating effect, as formalized in the following Corollary:

**Corollary 3.2** (Tighter error bound for diagonal-Gaussian approximation). *Let $\boldsymbol{X}$ and $\boldsymbol{Y}$ be local Gaussian patches with covariances $\boldsymbol{\Sigma_X}$ and $\boldsymbol{\Sigma_Y}$. Let $\widetilde{WD}$ and $\widetilde{WD}_{\mathrm{wave}}$ denote their diagonal-Gaussian approximations in the original and wavelet domains. They both satisfy error bounds of the*

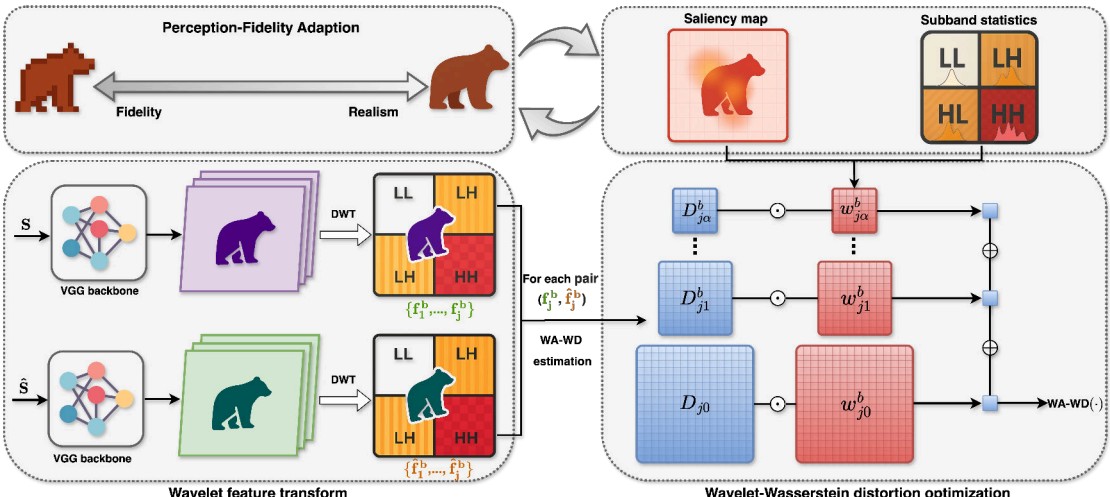

*Figure 3.* Illustration of WA-WD, where paired images are processed by a VGG backbone and DWT, decomposing features into orthogonal subbands for multi-scale WA-WD estimation. Combined with saliency and subband statistics, WA-WD enables fine-grained realism–fidelity optimization.

*same form:*

$$\left| WD(\boldsymbol{X}, \boldsymbol{Y}) - \widetilde{WD}(\boldsymbol{X}, \boldsymbol{Y}) \right| \leq C \sum_{Z \in \{\boldsymbol{X}, \boldsymbol{Y}\}} \|\text{offdiag}(\boldsymbol{\Sigma_Z})\|_F, \tag{2}$$

$$\left| WD_{\text{wave}}(\boldsymbol{X}, \boldsymbol{Y}) - \widetilde{WD}_{\text{wave}}(\boldsymbol{X}, \boldsymbol{Y}) \right| \leq C \sum_{Z \in \{\boldsymbol{X}, \boldsymbol{Y}\}} \|\text{offdiag}(\boldsymbol{W}\boldsymbol{\Sigma_Z}\boldsymbol{W}^\top)\|_F, \tag{3}$$

*where the constant $C > 0$ follows Lemma B.2 and* offdiag$(\Sigma) \triangleq \Sigma - \text{diag}(\Sigma)$. *Since wavelet coefficients of natural images are typically far more decorrelated, the off-diagonal terms of $\boldsymbol{W}\boldsymbol{\Sigma_X}\boldsymbol{W}^\top$ and $\boldsymbol{W}\boldsymbol{\Sigma_Y}\boldsymbol{W}^\top$ are much smaller, yielding a* tighter error bound in practice.

Consequently, the error bound in Corollary 3.2 can be generally tighter in wavelet domain for natural images, indicating that the diagonal-Gaussian approximation in WA-WD can be statistically more accurate (with proof in Appendix B). Evaluating WD within individual wavelet subbands places the computation in a domain where the independence assumptions hold more closely, resulting in more stable and perceptually consistent optimization.

**Practical implementation.** WA-WD is approximated by pre-computed local moments at discrete pooling scales, same as (Ballé et al., 2025). As shown in Fig. 3, original image $\boldsymbol{S}$ and its reconstruction $\hat{\boldsymbol{S}}$ are passed through a VGG backbone and a Haar wavelet transform, yielding multiscale features $\boldsymbol{f}_j^b$ and $\hat{\boldsymbol{f}}_j^b$. Each feature pair is further downsampled by with scale factor $\alpha$ to compute

element-wise first- and second-order moments (Eqns. 25, 26), from which a WD map $\boldsymbol{D}_{j\alpha}^b$ is obtained with associated pooling scales $\sigma_{j\alpha}^b$. Similarly, $\sigma_{j\alpha}^b$ are derived from EML-Net saliency maps, where scores $s \in [0, 1]$ are converted into a spatial likelihood $p = p_{\min} + (1 - p_{\min}) \cdot \frac{s}{\bar{s}}$, with $p_{\min} = 0.5$ ensuring positivity and unit mean. The $\sigma$ field is then set as $\sigma = \sigma_{\max}^b \cdot \frac{p_{\min}}{p(x,y)}$, with $\sigma_{\max}^b$ subband-dependent. The integration of VGG and DWT yields multi-scale features, naturally aligned with pooling widths, serving as references for estimating WD at arbitrary $\sigma$. WA-WD is then interpolated with a weight map $\boldsymbol{w}_{j\alpha}^b$ (with elements from $\max(0, 1 - |\log_2 \sigma_{j\alpha}^b - \alpha|)$, and aggregated across different feature and subbands as: $WD_{\text{wave}} = \sum_{b \in \{LL, LH, HL, HH\}} \sum_{j,\alpha} (\boldsymbol{w}_{j\alpha}^b \odot \boldsymbol{D}_{j\alpha}^b)$. See (Ballé et al., 2025) for more approximation details.

### 3.3. Region-wise perception modeling

As shown in Fig. 2, Re$^2$IC comprises five components: region partition, chain coding, local–global perceptual networks (LPNs $f_{\boldsymbol{W}^i}$ and GPM $g_{\boldsymbol{\theta}}$), latent vectors ($\boldsymbol{z}^i$ and randomness prior $\boldsymbol{M}^i$), and a shared entropy model ($r_{\boldsymbol{\psi}}$). Details of each component are provided in Appendix D.

**Region partition.** To encode an image, Re$^2$IC partitions images into regions using saliency predictions. An EML-Net (Jia & Bruce, 2020) is employed to generate a saliency map, from which we threshold and extract $N - 1$ largest connected components as sub-regions for LPN allocation.

**Chain coding.** To compress contours efficiently, we adopt adaptive chain coding (Chang et al., 1992). It encodes curves with a starting point and directional steps, adapting to curvature for low bit cost. Re$^2$IC further applies contour

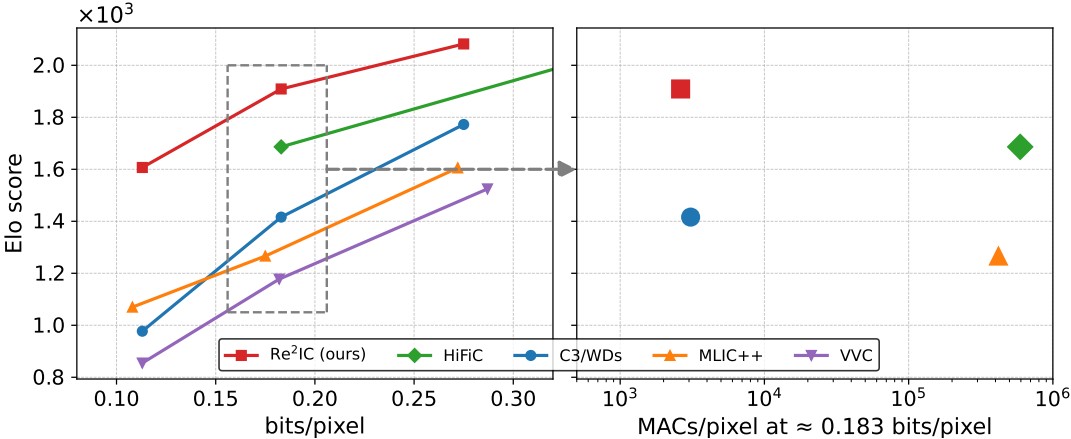

*Figure 4.* Human rating results and decoder complexity on Kodak. Left: Evaluation of different methods vs. bit rate. Right: Decoding complexity at the middle bit-rate regime. Re$^2$IC achieves the best perceptual quality by human preference while maintaining the lowest decoding complexity.

dilation, as saliency maps typically lack precise pixel-level semantics, thereby improving robustness to lossy contours and high-curvature boundaries while reducing cost.

**Latent vectors.** For the $i$-th region, the learnable latent vectors are defined as $z^i \triangleq \{z^i_1, z^i_2, \ldots, z^i_L\}$, where each $z^i_j \in \mathbb{R}^{\frac{K^i}{4^{j-1}}}$ corresponds to a $2^{j-1} \times 2^{j-1}$ downsampling of the region, with $K^i$ denoting its pixel count. To improve perceptual quality, we introduce common randomness via a Gaussian matrix $M^i \sim \mathcal{N}(0,1)$ (same shape as $z^i$), generated synchronously by the same random seed at both ends without coding cost. The decoded $z^i$ and $M^i$ are concatenated and fed into the GPM for perceptual compensation.

**Local-global perceptual networks.** Each LPN reconstructs pixel values from region coordinates $x^i$, while the GPM injects global context using the quantized latent vectors as input. As shown in Fig. 10, each $f_{W^i}$ consists of four convolutional layers, each modulated by $\mathcal{M}(\cdot)$, which fuses the layer output with a vector from $g_\theta$ through a shared MLP. Sharing $\mathcal{M}(\cdot)$ across regions reduces bitrate and implicitly blends local and global features for improved perceptual modeling. The GPM $f_\theta$ itself comprises an upsampling module followed by four convolutional layers.

### 3.4. Rate-perception optimization.

To balance RP trade-off, Re$^2$IC with an $N$-region configuration is trained to overfit each data sample into the parameter set: $\Omega \triangleq \left\{ \psi, \theta, \bigcup_{i=1}^{N} \left( z^i, \tau^i, W^i \right) \right\}$, by modeling visual perception with the following loss: $\mathcal{L}(\Omega) = \left[ WD_{\text{wave}}(S^i, \bigcup_{i=1}^{N} \hat{S}^i) + \lambda R(\hat{z}^i) \right]$, where $\hat{S}^i = f_{W^i}(f_\theta(\hat{z}^i, M^i), x^i)$ denotes the $i$-th region reconstruction, $WD_{\text{wave}}(\cdot)$ is Wavelet–WD, $R(\hat{z}^i)$ is the latent rate term, and $\lambda$ is a RP trade-off hyperparameter. Further im-

plementation details are given in Appendix D.2.

## 4. Experimental results

### 4.1. Experiment setup

We evaluate Re$^2$IC[1] on the Kodak (Kodak, 1991) and the CLIC2020 professional validation set (Toderici et al., 2020), using both human rater studies and quantitative assessments. Baselines include the classical VTM 19.1 (Bross et al., 2021), state-of-the-art autoencoder codecs MLIC$^+$ and MLIC$^{++}$ (Jiang et al., 2025), the generative codec HiFiC (Mentzer et al., 2020), and perception-enhanced implicit codec C3/WDs (Ballé et al., 2025). For each method, we target $\{0.075, 0.15, 0.3\}$ bpp on CLIC, and on Kodak $\{0.113, 0.183, 0.275\}$ bpp to align with open-sourced baselines. More baselines, such as TACO (Lee et al., 2024), CDC (Yang & Mandt, 2023), and WeConvene (Fu et al., 2024) are provided in the Appendix (Tables 6, 7, and 14).

For quantitative evaluation, we use PSNR for fidelity; MS-SSIM, LPIPS, DISTS, FID, and KID for reference-based perception; CLIP-IQA, MUSIQ, and MANIQA as learned no-reference preference metrics; and NIQE as a handcrafted naturalness prior. We further measure MACs/pixel and latency for decoding efficiency. For user studies, we test all three target rates for baselines and the two lower rates for HiFiC, giving 14 method–rate combinations per dataset.

### 4.2. Rate-perception performance

**User study.** Our evaluation protocol follows the CLIC framework with its open-source rating model. User is required to select one between two reconstructions closer

---

[1]In all experiments, Re$^2$IC is trained for 80k steps with fast convergence though further gains are possible.

*Table 1.* Quantitative comparison of different methods on Kodak. We report scores on BPP↓, PSNR↑, MS-SSIM↑, LPIPS↓, DISTS↓, FID↓, NIQE↓, MUSIQ↑, ClipIQA↑, and MANIQA↑. **Red** and blue indicate the best and second-best results, respectively. HiFiC is excluded from the high-BPP regime comparisons since its rate is significantly higher. Results for the $Re^2IC$ counterparts with sub-band tuning are also included, demonstrating adaptive trade-offs for additional RP gains.

| | Model | BPP↓ | PSNR↑ | MS-SSIM↑ | LPIPS↓ | DISTS↓ | FID↓ | NIQE↓ | MUSIQ↑ | ClipIQA↑ | MANIQA↑ |
|---|---|---|---|---|---|---|---|---|---|---|---|
| Low | VTM | 0.113 | 28.51 | 0.9279 | 0.3246 | 0.2118 | 186.59 | 5.7412 | 66.74 | 0.3595 | 0.3564 |
| | MLIC$^{++}$ | 0.108 | **29.16** | **0.9339** | 0.3043 | 0.2076 | 175.71 | 5.9582 | **71.21** | 0.4803 | **0.4289** |
| | C3/WDs | 0.113 | 24.41 | 0.8958 | 0.1717 | 0.1179 | 72.83 | 5.6089 | 64.98 | 0.4803 | 0.3538 |
| | HiFiC | - | - | - | - | - | - | - | - | - | - |
| | Re$^2$IC | 0.113 | 24.67 | 0.8802 | **0.1458** | **0.0975** | **63.89** | **3.9574** | 68.37 | **0.5255** | 0.3932 |
| | ↪ Tuning | 0.113 | 23.76 | 0.8408 | 0.1432 | 0.0860 | 58.21 | 3.9574 | 69.96 | 0.5797 | 0.4125 |
| Medium | VTM | 0.182 | 30.10 | 0.9520 | 0.2538 | 0.1767 | 147.87 | 5.1808 | 70.68 | 0.4827 | 0.4084 |
| | MLIC$^{++}$ | 0.175 | **30.72** | **0.9555** | 0.2358 | 0.1749 | 141.69 | 5.3280 | 73.33 | 0.5552 | 0.4495 |
| | C3/WDs | 0.183 | 26.23 | 0.9324 | 0.1150 | 0.0786 | 49.87 | 5.5448 | 68.49 | 0.5062 | 0.3899 |
| | HiFiC | 0.183 | 27.56 | 0.9456 | **0.0665** | 0.0889 | 54.89 | **2.7894** | **73.76** | **0.6562** | **0.4576** |
| | Re$^2$IC | 0.183 | 26.13 | 0.9168 | 0.1022 | **0.0616** | **40.01** | 4.0660 | 70.62 | 0.5988 | 0.4209 |
| | ↪ Tuning | 0.183 | 25.65 | 0.9006 | 0.1017 | 0.0572 | 37.05 | 4.0302 | 71.86 | 0.6149 | 0.4388 |
| High | VTM | 0.287 | 31.84 | 0.9694 | 0.1864 | 0.1416 | 106.87 | 4.6125 | 73.13 | 0.5623 | 0.4586 |
| | MLIC$^{++}$ | 0.272 | **32.33** | **0.9719** | 0.1720 | 0.1425 | 111.99 | 4.6692 | **74.72** | 0.6027 | **0.4692** |
| | C3/WDs | 0.275 | 27.79 | 0.9529 | 0.0802 | 0.0537 | 35.39 | 4.8479 | 70.15 | 0.5491 | 0.4085 |
| | HiFiC | 0.351 | 29.65 | 0.9707 | 0.0428 | 0.0639 | 34.37 | 2.9543 | 74.19 | **0.6721** | 0.4588 |
| | Re$^2$IC | 0.274 | 27.27 | 0.9406 | **0.0752** | **0.0411** | **28.23** | 4.2442 | 71.98 | 0.6264 | 0.4411 |
| | ↪ Tuning | 0.275 | 26.74 | 0.9231 | 0.0764 | 0.0374 | 26.09 | 3.8981 | 72.27 | 0.6446 | 0.4446 |

to the original, and Elo scores for each method and rate are computed by minimizing the cross-entropy between observed and predicted preferences (details provided in Appendix C.1). The main results on Kodak are shown in Fig. 4. While C3/WDs outperforms MLIC$^{++}$ and VVC, it lags behind generative-based HiFiC. Re$^2$IC closes this gap, surpassing HiFiC while operating at two orders of magnitude lower decoding complexity. Since HiFiC's open-sourced rates are higher than those of other baselines, we further conducted direct pairwise comparisons between HiFiC and Re$^2$IC at its released rates (see Fig. 9), further providing a clear performance gain for Re$^2$IC.[2] This highlights the promise of Re$^2$IC: by modeling perception region-by-region rather than learning a distribution, it can achieve superior RP performance at drastically reduced complexity.

**Quantitative evaluations.** Comprehensive quantitative assessments are reported in Table 1 for Kodak and Table 2 for CLIC2020. Across all bitrates in Table 2, Re$^2$IC consistently leads in perceptual metrics (such as LPIPS, DISTS, FID, KID, NIQE, and CLIP-IQA), and requires significantly less decoding complexity. Similar advantages can also be observed in Table 1. Interestingly, overfitted codecs often show greater performance gains at higher bitrates (Kim et al., 2024; Wu et al., 2025), but for perceptual quality, the

---

[2]More comparisons on CLIC2020 and Kodak are provided in supplementary materials, with interactive flipping comparisons.

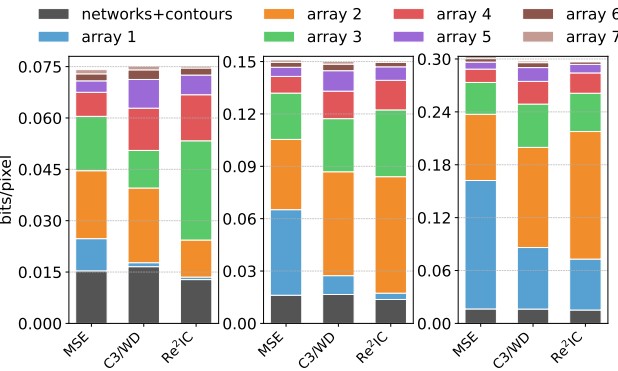

*Figure 5.* Bit allocation across latents from array 1-7 (highest to lowest resolution) at 0.075, 0.15, and 0.3 bpps.

advantage is more pronounced at low bitrates.

Notably, as shown in Table 2, Re$^2$IC not only surpasses C3/WDs in perceptual quality but also improves distortion, with up to a 0.82 dB PSNR gain, underscoring its leading role among overfitted codecs. Moreover, Re$^2$IC achieves better WD than C3/WDs despite not being directly trained on it. This highlights the effectiveness of Re$^2$IC's design, making WD optimization more effective and enhancing overall performance. Considering that WA-WD is tunable via learnable subband weights, we also report tuned Re$^2$IC results in Table 1, demonstrating its potential for improved

*Table 2.* Quantitative comparison of different methods on CLIC2020. We report performance on PSNR↑, MS-SSIM↑, LPIPS↓, DISTS↓, FID↓, KID↓, NIQE↓, MUSIQ↑, ClipIQA↑, and MANIQA↑. **Red** and blue indicate the best and second-best results, respectively. VTM and CDC is excluded from the high-BPP regime comparisons since its rate is significantly higher or lower.

| | Model | BPP↓ | PSNR↑ | MS-SSIM↑ | LPIPS↓ | DISTS↓ | FID↓ | KID↓ | NIQE↓ | MUSIQ↑ | ClipIQA↑ | MANIQA↑ | WD↓ |
|---|---|---|---|---|---|---|---|---|---|---|---|---|---|
| Low | VTM | 0.0790 | 30.37 | 0.9440 | 0.3205 | 0.2196 | 114.88 | 57.29 | 5.9644 | 52.96 | 0.3471 | 0.3263 | 2.5756 |
| | MLIC$^+$ | 0.0780 | **30.92** | **0.9488** | 0.3035 | 0.2201 | 105.93 | 46.66 | 5.9993 | **57.11** | 0.4321 | **0.3506** | 2.4406 |
| | C3/WDs | 0.0740 | 25.97 | 0.8972 | 0.1228 | 0.0844 | 24.57 | 5.79 | 5.1572 | 53.40 | 0.4773 | 0.3199 | 1.3082 |
| | HiFiC | - | - | - | - | - | - | - | - | - | - | - | - |
| | Re$^2$IC | 0.0750 | 26.57 | 0.9072 | **0.1136** | **0.0708** | **17.16** | **2.34** | **4.3611** | 55.58 | **0.5226** | 0.3352 | **1.1131** |
| Medium | VTM | 0.1480 | 32.39 | 0.9665 | 0.2495 | 0.1837 | 89.83 | 47.54 | 5.3504 | 57.34 | 0.4230 | 0.3521 | 1.1709 |
| | MLIC$^+$ | 0.1490 | **32.99** | **0.9703** | 0.2334 | 0.1840 | 85.65 | 38.75 | 5.3692 | 59.84 | 0.4777 | **0.3539** | 1.5845 |
| | C3/WDs | 0.1490 | 27.94 | 0.9357 | 0.0767 | 0.0459 | 13.71 | 2.63 | 4.7688 | 56.38 | 0.5140 | 0.3352 | 0.6487 |
| | HiFiC | 0.1420 | 29.83 | 0.9594 | **0.0589** | 0.0695 | 18.81 | 2.33 | **3.3044** | 59.91 | 0.5140 | 0.3390 | 1.0509 |
| | Re$^2$IC | 0.1500 | 28.53 | 0.9422 | 0.0754 | **0.0381** | **11.82** | **1.41** | 4.4016 | 57.24 | **0.5509** | 0.3429 | **0.5498** |
| High | VTM | 0.3880 | 36.14 | 0.9865 | 0.1571 | 0.1224 | 53.41 | 32.07 | 4.5570 | 61.57 | 0.5335 | 0.3815 | 0.7759 |
| | MLIC$^+$ | 0.3020 | **35.44** | **0.9843** | 0.1708 | 0.1417 | 64.27 | 32.40 | 4.7365 | **61.97** | 0.5263 | **0.3748** | 0.9233 |
| | C3/WDs | 0.2950 | 29.59 | 0.9572 | 0.0498 | 0.0223 | 7.90 | 1.23 | 4.5353 | 58.59 | 0.5432 | 0.3440 | 0.3086 |
| | HiFiC | 0.2720 | 31.84 | 0.9774 | **0.0380** | 0.0492 | 12.74 | 1.40 | **3.4455** | 60.23 | 0.5142 | 0.3406 | 0.6168 |
| | Re$^2$IC | 0.2970 | 30.41 | 0.9636 | 0.0483 | **0.0187** | **7.10** | **0.80** | 4.1808 | 58.89 | **0.5492** | 0.3486 | **0.2411** |

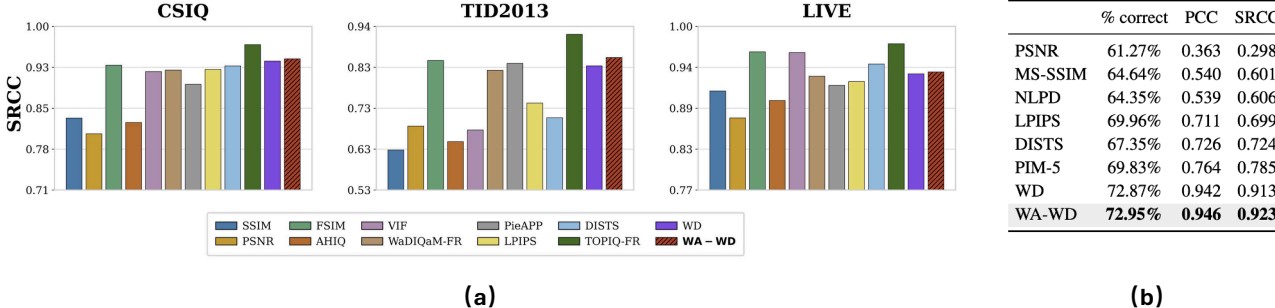

| | % correct | PCC | SRCC |
|---|---|---|---|
| PSNR | 61.27% | 0.363 | 0.298 |
| MS-SSIM | 64.64% | 0.540 | 0.601 |
| NLPD | 64.35% | 0.539 | 0.606 |
| LPIPS | 69.96% | 0.711 | 0.699 |
| DISTS | 67.35% | 0.726 | 0.724 |
| PIM-5 | 69.83% | 0.764 | 0.785 |
| WD | 72.87% | 0.942 | 0.913 |
| WA-WD | **72.95%** | **0.946** | **0.923** |

**(a)**                    **(b)**

*Figure 6.* Evaluation of WA-WD: (a) performance across IQA datasets; (b) human-rating prediction.

RP performance and flexible R–D–P trade-offs. More quantitative results validating the advantages of Re$^2$IC are also provided in Tables 6,7 and Figs. 13,14.

**Bit-breakdown.** A breakdown of bit cost in Fig. 5 shows that, Re$^2$IC allocates bits more evenly across latent vectors, capturing both low-frequency structures (typically in low-resolution latents) and high-frequency details (in high-resolution latents). This balanced allocation encourages richer cross-frequency interactions, consistent with our WA-WD design intuition, leading to more efficient bit usage and a better PD trade-off. Latent visualizations in Fig. 20 further confirm that Re$^2$IC enhances these interactions.

### 4.3. Perception metric

To validate WA-WD as an effective perceptual indicator for compression, we directly use it to predict user preferences in the open-source human rating study (Ballé et al., 2025), where prediction quality was measured using Pearson (PCC) and Spearman (SRCC) correlations. As shown in Fig. 6 (b), WA-WD achieved the highest alignment with Elo scores.

We further evaluate WA-WD on multiple image quality assessment (IQA) datasets (Fig. 6 (a)), where it consistently outperforms WD and ranks among the top perceptual metrics. Notably, these results rely on a plain VGG backbone; even larger gains are expected with WA-WD tuning or more advanced backbones.

### 4.4. Ablation studies

**Convergence speed.** Another advantage of Re$^2$IC is fast convergence. We evaluate the convergence speed of both RD and RP tasks. For RD (Fig. 7 b), Re$^2$IC reaches the BD-rate of C3 (Kim et al., 2024) (100k encoding steps) with only 25k steps, highlighting the effectiveness of the partition strategy. For RP (Fig. 7 a), Re$^2$IC delivers satisfactory perceptual quality within 2k steps, far fewer than C3/WDs (Ballé et al., 2025) (more examples in Fig. 22).

**PD trade-off.** We compare Re$^2$IC with variants trained using MSE and WD. As shown in Fig. 7 (c), optimizing for MSE alone significantly reduces perceptual quality. Re$^2$IC/WDs outperforms C3/WDs, validating the benefit of

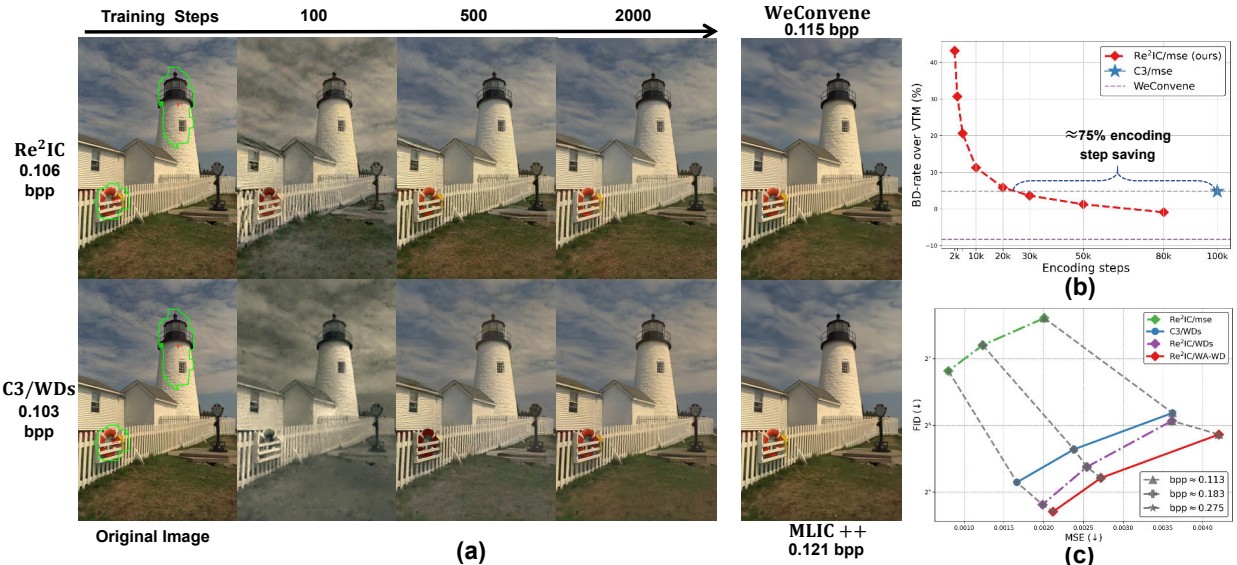

*Figure 7.* Encoding convergence: (a) RP and (b) RD; (c) P–D trade-offs on Kodak.

region-wise coding, while adding wavelet control provides further perceptual gains. This progression, from fidelity-only to full Re$^2$IC paradigm, highlights the cumulative effect of each module for improved perceptual performance and aligns with the R-D-P trade-off. More ablation studies for each component are provided in Tables 11 and 12.

**Contour accuracy.** We examine the effect of artifacts from lossy contour compression. As shown in Fig. 21, for Re$^2$IC/mse at low bitrates, artifacts may arise at LPN boundaries or where distributions change abruptly. In contrast, Re$^2$IC/WA-WD avoids such issues. This demonstrates that WA-WD, by modeling visual inhomogeneity and integrating low- and high-frequency interactions, improves visual robustness at region boundaries.

**Visualization.** Example visual comparisons are provided for the Kodak in Figs. 23, 25, 26, 27, and 28, and for the CLIC2020 in Figs. 29, 30, 31, and 32, demonstrating significantly improved RP performance of Re$^2$IC. More detailed image-wise comparisons are available on our project page.

## 5. Conclusion

We presented Re$^2$IC, a realism-enhanced region-based implicit codec for perceptual image compression at low complexity. By combining saliency-driven region partitioning with lightweight local-global modulation, Re$^2$IC enables efficient region-wise perception modeling with low decoding cost. We further introduced wavelet–Wasserstein distortion, which provides explicit frequency-aware control over the fidelity–realism trade-off and supports fine-grained optimization and flexible bit allocation. Extensive experiments show that Re$^2$IC outperforms state-of-the-art implicit

codecs and even generative methods such as HiFiC, while requiring significantly lower decoding cost. In addition, wavelet–WD also serves as a standalone, tunable perceptual metric that aligns more closely with human preference. Together, these results demonstrate that region- and frequency-aware perceptual modeling provides an effective foundation for low-complexity, perception-driven image compression.

**Limitations and future work.** While Re$^2$IC accelerates convergence, its reliance on a VGG backbone increases encoding complexity. It is therefore most suitable for scenarios with one-time encoding but massive reuse, such as streaming, where lightweight decoding ensures high perceptual quality on resource-limited devices. In the future, Re$^2$IC can be explored with efficient perception-tailored backbones to reduce encoding complexity. Another promising direction is to extend it to video compression by leveraging dynamic region partitioning guided by motion and temporal attention.

## Impact Statement

This paper presents work whose goal is to advance the field of Machine Learning. There are many potential societal consequences of our work, none of which we feel must be specifically highlighted here.

## Acknowledgments

This work was started while Haotian Wu was with Imperial College London. This work received funding from the UKRI for the projects INFORMED-AI (EP/Y028732/1) and AI-R (ERC Consolidator Grant, EP/X030806/1).

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

# Appendix Contents

## A. Open resources

**(a) Chain coding method.**    We open-source an adaptive chain-coding method (Chang et al., 1992), which can support both lossy and lossless coding for contour.

**(b) Code and reconstructions.**    Code and all reconstructions are released in our project page, with interactive comparisons.

**(c) Detailed evaluation results at various rates.**    We provide reconstructions with their rates on Kodak and CLIC2020 for different methods to facilitate future research and comparisons.

*Given the subjectivity of perception comparisons, we refer to our project page, which provides detailed one-by-one comparisons.*

## B. Analysis and Proof

### B.1. Proof of Theorem 3.1: Metric Preservation

Let $\boldsymbol{W} \in \mathbb{R}^{d \times d}$ denote an orthonormal discrete wavelet transform (DWT) and $WD(\cdot)$ denote the Wasserstein distortion. Define the wavelet Wasserstein distortion as $WD_{\text{wave}}(\boldsymbol{z}, \hat{\boldsymbol{z}}) = WD(\boldsymbol{W}\boldsymbol{z}, \boldsymbol{W}\hat{\boldsymbol{z}})$. Then the following hold:

- If PMFs $q_\sigma$ have no spectral nulls and $d(\cdot)$ is a metric (i.e, $WD(\boldsymbol{z}, \hat{\boldsymbol{z}})^{1/p}$ is a metric), $WD_{\text{wave}}(\boldsymbol{z}, \hat{\boldsymbol{z}})^{1/p}$ is a metric.

- If $\phi(\cdot)$ is invertible (i.e, $WD(\boldsymbol{x}, \hat{\boldsymbol{x}})^{1/p}$ is a metric), $WD_{\text{wave}}(\boldsymbol{x}, \hat{\boldsymbol{x}})^{1/p}$ is also a metric.

*Proof.*  Since $WD(\boldsymbol{z}, \hat{\boldsymbol{z}})^{1/p}$ is a metric, whenever $d(\cdot)$ is a metric and $q_\sigma$ has no spectral nulls, and since $\boldsymbol{W}$ is linear and invertible, it follows that $WD_{\text{wave}}(\boldsymbol{z}, \hat{\boldsymbol{z}})^{1/p}$ inherits all metric properties from $WD(\boldsymbol{z}, \hat{\boldsymbol{z}})^{1/p}$. In particular, $WD_{\text{wave}}$ is symmetric and satisfies $WD_{\text{wave}}(\boldsymbol{z}, \boldsymbol{z}) = 0$.

For any $\boldsymbol{z} \neq \hat{\boldsymbol{z}}$, the invertibility of $\boldsymbol{W}$ guarantees $\boldsymbol{W}\boldsymbol{z} \neq \boldsymbol{W}\hat{\boldsymbol{z}}$. Because $q_\sigma$ has no spectral nulls, therefore

$$q_\sigma * (\boldsymbol{W}\boldsymbol{z}) \;\neq\; q_\sigma * (\boldsymbol{W}\hat{\boldsymbol{z}}), \tag{4}$$

where $*$ denotes circular convolution (Oppenheim, 1999). Let $\boldsymbol{y}$ and $\hat{\boldsymbol{y}}$ denote these pooled signals. Since $W_p$ is a metric (Villani, 2008), $\boldsymbol{y} \neq \hat{\boldsymbol{y}}$ implies $W_p^p(y_l, \hat{y}_l) > 0$ for some $l$, and hence $WD_{\text{wave}}(\boldsymbol{z}, \hat{\boldsymbol{z}}) > 0$.

If $\phi$ is invertible, then $\boldsymbol{x} \neq \hat{\boldsymbol{x}}$ implies $\boldsymbol{z} \neq \hat{\boldsymbol{z}}$, and thus $WD_{\text{wave}}(\boldsymbol{x}, \hat{\boldsymbol{x}}) > 0$ as well. The triangle inequality for $WD_{\text{wave}}(\boldsymbol{x}, \hat{\boldsymbol{x}})$ and $WD_{\text{wave}}(\boldsymbol{z}, \hat{\boldsymbol{z}}) > 0$ follows directly from the fact that $W_p$ is a metric and from Minkowski's inequality.

In a nutshell, $WD_{\text{wave}}^{1/p}$ is a metric on the feature space whenever PMF has no spectral nulls, and it becomes a metric on the input space when the feature extractor $\phi$ is invertible. $\qquad\square$

*Remark* B.1 (Practical blockwise implementation).  In practice, with the diagonal-Gaussian assumption, WA-WD can be implemented by computing WD independently in each subband and summing their contributions with equal weight.

Since the DWT is assumed orthonormal, which can be decomposed as $\boldsymbol{W}^\top = \begin{bmatrix} \boldsymbol{W}_{LL}^\top & \boldsymbol{W}_{LH}^\top & \boldsymbol{W}_{HL}^\top & \boldsymbol{W}_{HH}^\top \end{bmatrix}$. Under the practical diagonal-Gaussian assumption, the Wasserstein distance becomes blockwise additive: $WD(\boldsymbol{W}\boldsymbol{z}, \boldsymbol{W}\hat{\boldsymbol{z}}) = \sum_{b \in \{LL, LH, HL, HH\}} WD(\boldsymbol{W}_b\boldsymbol{z}, \boldsymbol{W}_b\hat{\boldsymbol{z}})$.

**Lemma B.2.** *The constant $C$ in Corollary 3.2 depends only on the eigenvalue bounds of $\mathbf{\Sigma_X}, \mathbf{\Sigma_Y}$. If the eigenvalues lie in $[m, M]$, an admissible choice is $C = 2\sqrt{d}(1 + \frac{M}{2m} + \frac{M^{3/2}}{2m^{3/2}})$*

### B.2. Proof of Lemma B.2: error bound for diagonal-Gaussian approximation

Let $\mathbf{X}$ and $\mathbf{Y}$ be local Gaussian patches with covariances $\mathbf{\Sigma_X}$ and $\mathbf{\Sigma_Y}$, and let $\widetilde{WD}$ denote the diagonal-Gaussian approximation of $WD$. Then there exists a constant $C > 0$, depending only on the eigenvalue bounds of $\mathbf{\Sigma_X}$ and $\mathbf{\Sigma_Y}$, such that

$$\left| WD(\mathbf{X}, \mathbf{Y}) - \widetilde{WD}(\mathbf{X}, \mathbf{Y}) \right| \leq C \left( \|\mathrm{offdiag}(\mathbf{\Sigma_X})\|_F + \|\mathrm{offdiag}(\mathbf{\Sigma_Y})\|_F \right). \tag{5}$$

If the eigenvalues satisfy

$$0 < m \leq \lambda_{\min}(\mathbf{\Sigma_X}), \ \lambda_{\min}(\mathbf{\Sigma_Y}) \leq \lambda_{\max}(\mathbf{\Sigma_X}), \ \lambda_{\max}(\mathbf{\Sigma_Y}) \leq M < \infty, \tag{6}$$

then the constant is:

$$C = 2\sqrt{d}\left( 1 + \frac{M}{2m} + \frac{M^{3/2}}{2m^{3/2}} \right). \tag{7}$$

*Proof.* Consider $\mathbf{X} \sim \mathcal{N}(\boldsymbol{\mu}_X, \mathbf{\Sigma_X})$ and $\mathbf{Y} \sim \mathcal{N}(\boldsymbol{\mu}_Y, \mathbf{\Sigma_Y})$ be $d$-dimensional Gaussian feature vectors with symmetric positive definite covariance matrices $\mathbf{\Sigma_X}, \mathbf{\Sigma_Y} \in \mathbb{R}^{d \times d}$. The exact squared 2-Wasserstein distance between the two Gaussians is:

$$WD(\mathbf{X}, \mathbf{Y}) = \|\boldsymbol{\mu}_X - \boldsymbol{\mu}_Y\|_2^2 + \mathrm{Tr}\left( \mathbf{\Sigma_X} + \mathbf{\Sigma_Y} - 2(\mathbf{\Sigma_Y}^{1/2} \mathbf{\Sigma_X} \mathbf{\Sigma_Y}^{1/2})^{1/2} \right). \tag{8}$$

Meanwhile, the diagonal-Gaussian approximation in (Qiu et al., 2024, Eq. 12), obtained by replacing the covariances with their diagonal parts, is: $\widetilde{WD}(\mathbf{X}, \mathbf{Y}) = \|\boldsymbol{\mu}_X - \boldsymbol{\mu}_Y\|_2^2 + \|\boldsymbol{\sigma}_X - \boldsymbol{\sigma}_Y\|_2^2$, where $\boldsymbol{\sigma}_X, \boldsymbol{\sigma}_Y$ collect the square roots of the diagonal entries of $\mathbf{\Sigma_X}, \mathbf{\Sigma_Y}$. Define the diagonal parts $\mathbf{\Sigma_X}^d \triangleq \mathrm{diag}(\mathbf{\Sigma_X})$ and $\mathbf{\Sigma_Y}^d \triangleq \mathrm{diag}(\mathbf{\Sigma_Y})$:

$$\widetilde{WD}(\mathbf{X}, \mathbf{Y}) = \|\boldsymbol{\mu}_X - \boldsymbol{\mu}_Y\|_2^2 + \mathrm{Tr}\left( \mathbf{\Sigma_X}^d + \mathbf{\Sigma_Y}^d - 2((\mathbf{\Sigma_Y}^d)^{1/2} \mathbf{\Sigma_X}^d (\mathbf{\Sigma_Y}^d)^{1/2})^{1/2} \right). \tag{9}$$

Using linearity of the trace and the triangle inequality, we can bound this difference by a constant (depending on $d$) times

$$\left| WD(\mathbf{X}, \mathbf{Y}) - \widetilde{WD}(\mathbf{X}, \mathbf{Y}) \right|$$

$$= \left| \mathrm{Tr}\left( \mathbf{\Sigma_X} + \mathbf{\Sigma_Y} - 2(\mathbf{\Sigma_Y}^{1/2}\mathbf{\Sigma_X}\mathbf{\Sigma_Y}^{1/2})^{1/2} \right) - \mathrm{Tr}\left( \mathbf{\Sigma_X}^d + \mathbf{\Sigma_Y}^d - 2((\mathbf{\Sigma_Y}^d)^{1/2}\mathbf{\Sigma_X}^d(\mathbf{\Sigma_Y}^d)^{1/2})^{1/2} \right) \right|$$

$$= \left| \mathrm{Tr}\left( (\mathbf{\Sigma_X} + \mathbf{\Sigma_Y} - 2(\mathbf{\Sigma_Y}^{1/2}\mathbf{\Sigma_X}\mathbf{\Sigma_Y}^{1/2})^{1/2}) - (\mathbf{\Sigma_X}^d + \mathbf{\Sigma_Y}^d - 2((\mathbf{\Sigma_Y}^d)^{1/2}\mathbf{\Sigma_X}^d(\mathbf{\Sigma_Y}^d)^{1/2})^{1/2}) \right) \right|$$

$$= \left| \mathrm{Tr}\left( (\mathbf{\Sigma_X} - \mathbf{\Sigma_X}^d) + (\mathbf{\Sigma_Y} - \mathbf{\Sigma_Y}^d) - 2[(\mathbf{\Sigma_Y}^{1/2}\mathbf{\Sigma_X}\mathbf{\Sigma_Y}^{1/2})^{1/2} - ((\mathbf{\Sigma_Y}^d)^{1/2}\mathbf{\Sigma_X}^d(\mathbf{\Sigma_Y}^d)^{1/2})^{1/2}] \right) \right|$$

$$\leq \left| \mathrm{Tr}(\mathbf{\Sigma_X} - \mathbf{\Sigma_X}^d) \right| + \left| \mathrm{Tr}(\mathbf{\Sigma_Y} - \mathbf{\Sigma_Y}^d) \right| + 2\left| \mathrm{Tr}((\mathbf{\Sigma_Y}^{1/2}\mathbf{\Sigma_X}\mathbf{\Sigma_Y}^{1/2})^{1/2} - ((\mathbf{\Sigma_Y}^d)^{1/2}\mathbf{\Sigma_X}^d(\mathbf{\Sigma_Y}^d)^{1/2})^{1/2}) \right|$$

$$\leq C_0\left( \|\mathbf{\Sigma_X} - \mathbf{\Sigma_X}^d\|_F + \|\mathbf{\Sigma_Y} - \mathbf{\Sigma_Y}^d\|_F + \left\| (\mathbf{\Sigma_Y}^{1/2}\mathbf{\Sigma_X}\mathbf{\Sigma_Y}^{1/2})^{1/2} - ((\mathbf{\Sigma_Y}^d)^{1/2}\mathbf{\Sigma_X}^d(\mathbf{\Sigma_Y}^d)^{1/2})^{1/2} \right\|_F \right), \tag{10}$$

where $C_0 = 2\sqrt{d}$, derived from $|\mathrm{Tr}(\mathbf{M})| \leq \sqrt{d}\|\mathbf{M}\|_F$.

With (Higham, 2008, Theorem 6.2), the matrix square root is Fréchet differentiable and Lipschitz continuous on the set of symmetric positive definite matrices: $\|\mathbf{A}^{1/2} - \mathbf{B}^{1/2}\| \leq L\|\mathbf{A} - \mathbf{B}\|$, where $L = \frac{1}{\lambda_{min}(\mathbf{A})^{1/2} + \lambda_{min}(\mathbf{B})^{1/2}}$, where $\lambda_{min}$ is the smallest eigenvalues. We can then bound the last term: $\|\mathbf{A} - \mathbf{B}\|_F = \left\| (\mathbf{\Sigma_Y}^{1/2}\mathbf{\Sigma_X}\mathbf{\Sigma_Y}^{1/2}) - ((\mathbf{\Sigma_Y}^d)^{1/2}\mathbf{\Sigma_X}^d(\mathbf{\Sigma_Y}^d)^{1/2}) \right\|_F \leq$

$\|T_1\|_F + \|T_2\|_F$, where

$$
\begin{aligned}
A - B &= \boldsymbol{\Sigma}_{\boldsymbol{Y}}^{1/2} \boldsymbol{\Sigma}_{\boldsymbol{X}} \boldsymbol{\Sigma}_{\boldsymbol{Y}}^{1/2} - (\boldsymbol{\Sigma}_{\boldsymbol{Y}}^d)^{1/2} \boldsymbol{\Sigma}_{\boldsymbol{X}}^d (\boldsymbol{\Sigma}_{\boldsymbol{Y}}^d)^{1/2} \\
&= \underbrace{\boldsymbol{\Sigma}_{\boldsymbol{Y}}^{1/2} \boldsymbol{\Sigma}_{\boldsymbol{X}} \boldsymbol{\Sigma}_{\boldsymbol{Y}}^{1/2} - \boldsymbol{\Sigma}_{\boldsymbol{Y}}^{1/2} \boldsymbol{\Sigma}_{\boldsymbol{X}}^d \boldsymbol{\Sigma}_{\boldsymbol{Y}}^{1/2}}_{\text{depends only on } \boldsymbol{\Sigma}_{\boldsymbol{X}} - \boldsymbol{\Sigma}_{\boldsymbol{X}}^d} + \underbrace{\boldsymbol{\Sigma}_{\boldsymbol{Y}}^{1/2} \boldsymbol{\Sigma}_{\boldsymbol{X}}^d \boldsymbol{\Sigma}_{\boldsymbol{Y}}^{1/2} - (\boldsymbol{\Sigma}_{\boldsymbol{Y}}^d)^{1/2} \boldsymbol{\Sigma}_{\boldsymbol{X}}^d (\boldsymbol{\Sigma}_{\boldsymbol{Y}}^d)^{1/2}}_{\text{depends only on } \boldsymbol{\Sigma}_{\boldsymbol{Y}} - \boldsymbol{\Sigma}_{\boldsymbol{Y}}^d} \\
&= \underbrace{\boldsymbol{\Sigma}_{\boldsymbol{Y}}^{1/2} (\boldsymbol{\Sigma}_{\boldsymbol{X}} - \boldsymbol{\Sigma}_{\boldsymbol{X}}^d) \boldsymbol{\Sigma}_{\boldsymbol{Y}}^{1/2}}_{T_1} + \underbrace{\left( \boldsymbol{\Sigma}_{\boldsymbol{Y}}^{1/2} \boldsymbol{\Sigma}_{\boldsymbol{X}}^d \boldsymbol{\Sigma}_{\boldsymbol{Y}}^{1/2} - (\boldsymbol{\Sigma}_{\boldsymbol{Y}}^d)^{1/2} \boldsymbol{\Sigma}_{\boldsymbol{X}}^d (\boldsymbol{\Sigma}_{\boldsymbol{Y}}^d)^{1/2} \right)}_{T_2}.
\end{aligned}
\tag{11}
$$

For $T_1$ with $C_1 = \|\boldsymbol{\Sigma}_{\boldsymbol{Y}}^{1/2}\|_2^2 \le M$:

$$
\|T_1\|_F = \|\boldsymbol{\Sigma}_{\boldsymbol{Y}}^{1/2} (\boldsymbol{\Sigma}_{\boldsymbol{X}} - \boldsymbol{\Sigma}_{\boldsymbol{X}}^d) \boldsymbol{\Sigma}_{\boldsymbol{Y}}^{1/2}\|_F \le \|\boldsymbol{\Sigma}_{\boldsymbol{Y}}^{1/2}\|_2^2 \|\boldsymbol{\Sigma}_{\boldsymbol{X}} - \boldsymbol{\Sigma}_{\boldsymbol{X}}^d\|_F \le C_1 \|\boldsymbol{\Sigma}_{\boldsymbol{X}} - \boldsymbol{\Sigma}_{\boldsymbol{X}}^d\|_F.
\tag{12}
$$

For $T_2$, with $C_2' = \|D_X\|_2 (\|A_Y\|_2 + \|B_Y\|_2) \le 2M^{3/2}$, and $C_2 = \frac{M^{3/2}}{\sqrt{m}}$: let $A_Y = \boldsymbol{\Sigma}_{\boldsymbol{Y}}^{1/2}$, $B_Y = (\boldsymbol{\Sigma}_{\boldsymbol{Y}}^d)^{1/2}$ and $D_X = \boldsymbol{\Sigma}_{\boldsymbol{X}}^d$, where $T_2 = A_Y D_X A_Y - B_Y D_X B_Y = (A_Y - B_Y) D_X A_Y + B_Y D_X (A_Y - B_Y)$. We can have:

$$
\begin{aligned}
\|T_2\|_F &\le \|(A_Y - B_Y) D_X A_Y\|_F + \|B_Y D_X (A_Y - B_Y)\|_F \\
&\le \|A_Y - B_Y\|_F \|D_X\|_2 \|A_Y\|_2 + \|B_Y\|_2 \|D_X\|_2 \|A_Y - B_Y\|_F \\
&= \left( \|D_X\|_2 \|A_Y\|_2 + \|B_Y\|_2 \|D_X\|_2 \right) \|A_Y - B_Y\|_F \\
&= C_2' \|A_Y - B_Y\|_F \le C_2 \|\boldsymbol{\Sigma}_{\boldsymbol{Y}} - \boldsymbol{\Sigma}_{\boldsymbol{Y}}^d\|_F.
\end{aligned}
\tag{13}
$$

Using equation 11, equation 12, and equation 13, we can have:

$$
\|A - B\|_F \le C_1 \|\boldsymbol{\Sigma}_{\boldsymbol{X}} - \boldsymbol{\Sigma}_{\boldsymbol{X}}^d\|_F + C_2 \|\boldsymbol{\Sigma}_{\boldsymbol{Y}} - \boldsymbol{\Sigma}_{\boldsymbol{Y}}^d\|_F.
\tag{14}
$$

Finally, substitute into the last term of equation 10, with $L_1 = \frac{1}{2m}$:

$$
\begin{aligned}
\|A^{1/2} - B^{1/2}\|_F &\le L_1 \|A - B\|_F \\
&\le L_1 (C_1 + C_2) \left( \|\boldsymbol{\Sigma}_{\boldsymbol{X}} - \boldsymbol{\Sigma}_{\boldsymbol{X}}^d\|_F + \|\boldsymbol{\Sigma}_{\boldsymbol{Y}} - \boldsymbol{\Sigma}_{\boldsymbol{Y}}^d\|_F \right).
\end{aligned}
\tag{15}
$$

Let $L = L_1(C_1 + C_2) \le \frac{1}{2m}(M + \frac{M^{3/2}}{\sqrt{m}})$, we can obtain:

$$
\left\| \left( \boldsymbol{\Sigma}_{\boldsymbol{Y}}^{1/2} \boldsymbol{\Sigma}_{\boldsymbol{X}} \boldsymbol{\Sigma}_{\boldsymbol{Y}}^{1/2} \right)^{1/2} - \left( (\boldsymbol{\Sigma}_{\boldsymbol{Y}}^d)^{1/2} \boldsymbol{\Sigma}_{\boldsymbol{X}}^d (\boldsymbol{\Sigma}_{\boldsymbol{Y}}^d)^{1/2} \right)^{1/2} \right\|_F \le L \left( \|\boldsymbol{\Sigma}_{\boldsymbol{X}} - \boldsymbol{\Sigma}_{\boldsymbol{X}}^d\|_F + \|\boldsymbol{\Sigma}_{\boldsymbol{Y}} - \boldsymbol{\Sigma}_{\boldsymbol{Y}}^d\|_F \right).
\tag{16}
$$

The first two terms are exactly $\|\text{offdiag}(\boldsymbol{\Sigma}_{\boldsymbol{X}})\|_F$ and $\|\text{offdiag}(\boldsymbol{\Sigma}_{\boldsymbol{Y}})\|_F$. Combining these estimates and absorbing constants into a single $C > 0$ (depends only on the $\boldsymbol{\Sigma}_{\boldsymbol{X}}, \boldsymbol{\Sigma}_{\boldsymbol{Y}}$) yields:

$$
\begin{aligned}
\left| WD(\boldsymbol{X}, \boldsymbol{Y}) - \widetilde{WD}(\boldsymbol{X}, \boldsymbol{Y}) \right| &\le C \left( \|\boldsymbol{\Sigma}_{\boldsymbol{X}} - \boldsymbol{\Sigma}_{\boldsymbol{X}}^d\|_F + \|\boldsymbol{\Sigma}_{\boldsymbol{Y}} - \boldsymbol{\Sigma}_{\boldsymbol{Y}}^d\|_F \right) \\
&= C \left( \|\text{offdiag}(\boldsymbol{\Sigma}_{\boldsymbol{X}})\|_F + \|\text{offdiag}(\boldsymbol{\Sigma}_{\boldsymbol{Y}})\|_F \right),
\end{aligned}
\tag{17}
$$

which proves the claim with $C = 2\sqrt{d}(1 + \frac{M}{2m} + \frac{M^{3/2}}{2m^{3/2}})$. $\qquad\square$

*Remark* B.3 (The bounded-eigenvalue assumption). Bounding the eigenvalues of $\boldsymbol{\Sigma}_{\boldsymbol{X}}$ and $\boldsymbol{\Sigma}_{\boldsymbol{Y}}$ within $[m, M]$ is a standard requirement (Wang et al., 2024) ensuring the matrix square root is Lipschitz continuous. In deep feature spaces, this assumption is mild: local covariances are empirically well-conditioned due to batch normalization, bounded nonlinearities. As a result, feature statistics can remain within a compact region across images and datasets, making the bounded-eigenvalue assumption compatible with WD-based image compression.

**B.3. Proof of Corollary 3.2: Tighter diagonal-Gaussian approximation**

*Proof.* Let $\boldsymbol{X}$ and $\boldsymbol{Y}$ be local Gaussian patches with covariances $\boldsymbol{\Sigma_X}$ and $\boldsymbol{\Sigma_Y}$. Let $\widetilde{WD}$ and $\widetilde{WD}_{\text{wave}}$ denote their diagonal-Gaussian approximations in the original and wavelet domains. Suppose the original covariances $\boldsymbol{\Sigma_X}, \boldsymbol{\Sigma_Y}$ satisfy the eigenvalue bounds $mI \preceq \boldsymbol{\Sigma_X}, \boldsymbol{\Sigma_Y} \preceq MI$, and let $\tilde{\boldsymbol{\Sigma}}_{\boldsymbol{X}} = W\boldsymbol{\Sigma_X}W^\top, \tilde{\boldsymbol{\Sigma}}_{\boldsymbol{Y}} = W\boldsymbol{\Sigma_Y}W^\top$ be the covariances after an orthonormal wavelet transform $W$. Since orthonormal transforms preserve eigenvalues, both $\tilde{\boldsymbol{\Sigma}}_{\boldsymbol{X}}$ and $\tilde{\boldsymbol{\Sigma}}_{\boldsymbol{Y}}$ satisfy the same bounds:

$$mI \preceq \tilde{\boldsymbol{\Sigma}}_{\boldsymbol{X}}, \tilde{\boldsymbol{\Sigma}}_{\boldsymbol{Y}} \preceq MI. \tag{18}$$

Under the same constant $C = 2\sqrt{d}\left(1 + \frac{M}{2m} + \frac{M^{3/2}}{2m^{3/2}}\right)$. appearing in Lemma B.2, the approximation errors satisfy:

$$|WD(\tilde{\boldsymbol{X}}, \tilde{\boldsymbol{Y}}) - \widetilde{WD}(\tilde{\boldsymbol{X}}, \tilde{\boldsymbol{Y}})| \leq C\left(\|\text{offdiag}(\tilde{\boldsymbol{\Sigma}}_{\boldsymbol{X}})\|_F + \|\text{offdiag}(\tilde{\boldsymbol{\Sigma}}_{\boldsymbol{Y}})\|_F\right), \tag{19}$$

$$|WD(\boldsymbol{X}, \boldsymbol{Y}) - \widetilde{WD}(\boldsymbol{X}, \boldsymbol{Y})| \leq C(\|\text{offdiag}(\boldsymbol{\Sigma_X})\|_F + \|\text{offdiag}(\boldsymbol{\Sigma_Y})\|_F), \tag{20}$$

and because natural-image wavelet coefficients are well known to be approximately decorrelated (Figueiredo & Nowak, 2001; Simoncelli, 1999):

$$\|\text{offdiag}(\tilde{\boldsymbol{\Sigma}}_{\boldsymbol{X}})\|_F \ll \|\text{offdiag}(\boldsymbol{\Sigma_X})\|_F, \qquad \|\text{offdiag}(\tilde{\boldsymbol{\Sigma}}_{\boldsymbol{Y}})\|_F \ll \|\text{offdiag}(\boldsymbol{\Sigma_Y})\|_F, \tag{21}$$

we obtain a tighter error bound for WA-WD:

$$C\left(\|\text{offdiag}(\tilde{\boldsymbol{\Sigma}}_{\boldsymbol{X}})\|_F + \|\text{offdiag}(\tilde{\boldsymbol{\Sigma}}_{\boldsymbol{Y}})\|_F\right) \leq C(\|\text{offdiag}(\boldsymbol{\Sigma_X})\|_F + \|\text{offdiag}(\boldsymbol{\Sigma_Y})\|_F). \tag{22}$$

Thus, WA-WD generally yields a tighter error bound of WD whenever the wavelet transform reduces off-diagonal covariance energy. $\square$

*Remark* B.4 (Analyzing bounds rather than pointwise comparisons). A direct pointwise comparison of WD approximation error before and after a wavelet transform is generally infeasible, since the exact Wasserstein term depends on image content and local feature correlations. Thus, the quantity $|WD - \widetilde{WD}|$ can vary across patches and cannot be compared fairly without strong assumptions over source distributions. This motivates analyzing *error bounds* rather than pointwise differences. Bounding the approximation error in terms of off-diagonal covariance energy provides a content-independent criterion. Because DWT substantially reduces cross-correlation in natural images, these bounds become generally tighter after DWT. Hence, a bound-based analysis captures the intrinsic advantage of WA-WD even when pointwise comparison is not tractable.

**Corollary B.5** (Statistical properties of WA-WD). *WA-WD satisfies the following statistical properties: it (a). preserves the same first-order statistics as WD; (b). captures intra-block correlations that WD ignores under its diagonal-Gaussian approximation; (c) reduces to MSE when $\sigma = 0$.*

Corollary B.5 shows that WA-WD explicitly captures intra-block correlations ignored by WD under diagonal-Gaussian approximation, with additional examples given in Table 3. Proof is provided in Appendix B, with more visualizations for validation in Appendix F.5.

**B.4. Proof of Corollary B.5: Statistical properties of WA-WD**

*Proof.* As shown in Sections 3.1 and 3.2, under independent Gaussian approximation, the standard local WD between each feature component $z_i$ and its reconstruction $\hat{z}_i$ is:

$$D_{i,\sigma} = (\mu_i - \hat{\mu}_i)^2 + \left(\sqrt{V_i} - \sqrt{\hat{V}_i}\right)^2, \tag{23}$$

where $\mu_i$ and $V_i$ denote the local mean and variance of each element:

$$\mu_i = \sum_k q_\sigma(k)\, z_{i+k}, V_i = \sum_k q_\sigma(k)\left(z_{i+k} - \mu_i\right)^2. \tag{24}$$

The final WD is averaged over all locations as: $WD(\boldsymbol{z}, \hat{\boldsymbol{z}}) = \frac{1}{|\mathcal{I}|} \sum_{i \in \mathcal{I}} D_{i,\sigma}$.

For WA-WD, the resultant pooled mean and variance are:

$$\mu_i^b = \sum_k q_\sigma(k)\, z_{i+k}^b = h_b^\top \Big( \sum_k q_\sigma(k)\, w_{i+k} \Big) \triangleq \boldsymbol{h_b^T} \bar{\boldsymbol{w}}_{\boldsymbol{i}}, \tag{25}$$

$$V_i^b = \sum_k q_\sigma(k) \left( z_{i+k}^b - \mu_i^b \right)^2 = \sum_k q_\sigma(k) \big( \boldsymbol{h_b^T}(\boldsymbol{w_{i+k}} - \bar{\boldsymbol{w}}_{\boldsymbol{i}}) \big)^2 = \boldsymbol{h_b^T} \boldsymbol{C_i} \boldsymbol{h_b}, \tag{26}$$

where $\boldsymbol{C_i} \triangleq \sum_k q_\sigma(k)(\boldsymbol{w_{i+k}} - \bar{\boldsymbol{w}}_{\boldsymbol{i}})(\boldsymbol{w_{i+k}} - \bar{\boldsymbol{w}}_{\boldsymbol{i}})^\top$ denotes a pooled $4 \times 4$ *block covariance matrix*, $z_i^b = \boldsymbol{h_b^\top} \boldsymbol{w_i}$ denotes the Haar transform coefficients, with the Haar basis $\boldsymbol{h_b} \in \mathbb{R}^4$, and $\boldsymbol{w_i} \in \mathbb{R}^4$ is the vectorization of each feature block. With independence approximation, the per-band WD and the overall WA-WD are given as: $D_{i,\sigma}^b = (\mu_i^b - \hat{\mu}_i^b)^2 + \big( \sqrt{V_i^b} - \sqrt{\hat{V}_i^b} \big)^2$ and $WD_{\text{wave}}(\boldsymbol{z}, \hat{\boldsymbol{z}}) = \frac{1}{4} \sum_{b \in \{LL, LH, HL, HH\}} \frac{1}{|\mathcal{I}_b|} \sum_{i \in \mathcal{I}_b} D_{i,\sigma}^b$, respectively.

1. **Mean preservation.** Compare Eqn. 24 and Eqn. 25, since both pooling and the Haar transform are linear,

$$\mu_i^b = \boldsymbol{h_b^\top} \boldsymbol{\mu_i},$$

   meaning WA-WD preserves the first-order statistics of WD up to an orthonormal transform.

2. **Variance decomposition with correlation sensitivity.** From Eqn. 26, the variance of the each subband for WA-WD is obtained as

$$V_i^b = \boldsymbol{h_b^\top} \boldsymbol{C_i} \boldsymbol{h_b}.$$

   This explicitly depends on all entries of $C_i$, including cross-covariances (off-diagonal terms). The variance for WD (as shown in Eqn. 24), in contrast:

$$V_i = \sum_k q_\sigma(k) \left( z_{i+k} - \mu_i \right)^2.$$

   uses only $\text{diag}(\boldsymbol{C_i})$. Hence WA-WD captures intra-block correlations that standard WD ignores.

3. **Reduction to MSE when $\sigma = 0$.** When the pooling scale $\sigma(i) = 0$ at a location $i$, the pooling kernel degenerates to a single spatial point. This yields $D_{i,0} = (z_i - \hat{z}_i)^2$, i.e., WD reduces to the pointwise squared error. For WA-WD, the same happens in the wavelet domain. By orthonormality of $W$ (Parseval's theorem), we can have:

$$\sum_b (z_i^b - \hat{z}_i^b)^2 = \|W z_i - W \hat{z}_i\|_2^2 = \|z_i - \hat{z}_i\|_2^2.$$

   Thus, when $\sigma = 0$ everywhere, both WD and WA-WD reduce exactly to mean squared error (MSE), reflecting pure fidelity distortion without any pooling.

$\square$

To further illustrate their variance difference, we provide several block-distribution examples in Table 3, demonstrating how WA-WD captures intra-block correlations.

## C. Baseline and experimental setting

### C.1. Experiment setup

We evaluate Re$^2$IC on the Kodak dataset (24 images)(Kodak, 1991) and the CLIC2020 professional validation set (41 images)(Toderici et al., 2020), using both human rater studies and quantitative assessments. Baselines include classical codecs (VTM (Bross et al., 2021)), currently state-of-the-art autoencoder codecs (MLIC$^+$, MLIC$^{++}$(Jiang et al., 2025)), the generative codec HiFiC (Mentzer et al., 2020), and perception-enhanced overfitted codecs (C3/WDs (Ballé et al., 2025)), and fidelity-oriented overfitted codecs (C3/MSE (Kim et al., 2024)). Additionally, diffusion-based perception codecs, such

*Table 3.* Special cases of block covariance structures and their effect on wavelet-WD, where $p$ means the indexes of pixels in a block and $\sigma_p^2$ denotes their variance ($\sigma^2$ is the common variance).

| Block distribution | Effect over Wavelet-WD |
| --- | --- |
| Independent | $C = \mathrm{diag}(\sigma_1^2, \sigma_2^2, \sigma_3^2, \sigma_4^2)$. No cross-terms; wavelet variances reduce to weighted sums of pixel variances. The difference from WD is minimal. |
| Fully correlated | $C = \sigma^2 \mathbf{1} \mathbf{1}_{4\times4}$. All energy collapses into $LL$: $V^{LL} = \sigma^2$, while $V^{LH} = V^{HL} = V^{HH} = 0$. Wavelet-WD focuses on low-frequency distortion for smooth areas. |
| i.i.d. noise | $C = \sigma^2 I_{4\times4}$. Energy spreads evenly across subbands: $V^{LL} = V^{LH} = V^{HL} = V^{HH} = \sigma^2$. Represents uncorrelated, isotropic fluctuations. |
| Directional correlation | Strong horizontal correlation (horizontal edges) $\Rightarrow V^{LH}$ dominates. Strong vertical correlation (vertical edges) $\Rightarrow V^{HL}$ dominates. |
| Checkerboard correlation | Variance concentrates in $HH$ subband, which captures diagonal alternations. |
| Partial correlation | $C = \sigma^2[(1-\rho)I_{4\times4} + \rho \mathbf{1}_{4\times4}]$, interpolating between i.i.d. ($\rho = 0$) and fully correlated ($\rho = 1$). Then $V^{LL} = \sigma^2(1 + 3\rho)$, while $V^{LH} = V^{HL} = V^{HH} = \sigma^2(1 - \rho)$. Energy shifts smoothly from high-frequency to low-frequency as $\rho$ increases. |

as CDC (Yang & Mandt, 2023), DWT-based codec WeConvene (Fu et al., 2024), and TACO (Lee et al., 2024) are also introduced in Tables 6, 7, and 14.

For each method, we evaluate compressed reconstructions at three dataset-average bitrates (e.g., low, medium, and high bpp regimes). On CLIC2020, we target $\{0.075, 0.15, 0.3\}$ bpp (as in the CLIC competition) and all results come from the open-sourced reconstructions from (Ballé et al., 2025). For Kodak, we use $\{0.113, 0.183, 0.275\}$ bpp to align with open-sourced baselines.[3] Note that MLIC$^{++}$ is used for Kodak to match the target rate regime, while MLIC$^+$ is used for CLIC2020 to align with prior work (Ballé et al., 2025).

For quantitative evaluation, we use PSNR for fidelity; MS-SSIM, LPIPS, DISTS, FID (patch 256), and KID (patch 128, multiplied by $10^3$) for reference-based perceptual similarity; CLIP-IQA, MUSIQ, and MANIQA as learned no-reference preference metrics; and NIQE as a handcrafted naturalness prior. We also measure MACs per pixel and coding latency to assess decoding complexity and efficiency.

For the user study, our evaluation protocol follows the CLIC framework and uses their open-source rating model. In each trial, raters are shown a random $512 \times 512$ crop from Kodak, with the original on one side and two reconstructions on the other. By flipping between reconstructions, they select the one closer to the original, and Elo scores for each method and rate are computed by minimizing the cross-entropy between observed and predicted preferences. We included all three target rates for the baselines and the two lower rates for HiFiC, yielding 14 method–rate combinations per dataset and thus 14 reconstructions per image for both Kodak. In total, 10 users participate in our rating. For the evaluation of the WA-WD metric, we directly employ open-resourced data in (Ballé et al., 2025), as they have more comprehensive assessments for a better verification of our method.

For IQA experiments, we consider CSIQ (Larson & Chandler, 2010), TID2013 (Ponomarenko et al., 2015), LIVE (Sheikh, 2005) datasets, and report SRCC.

---

[3]All results are adopted from their officially released reconstructions or codes.

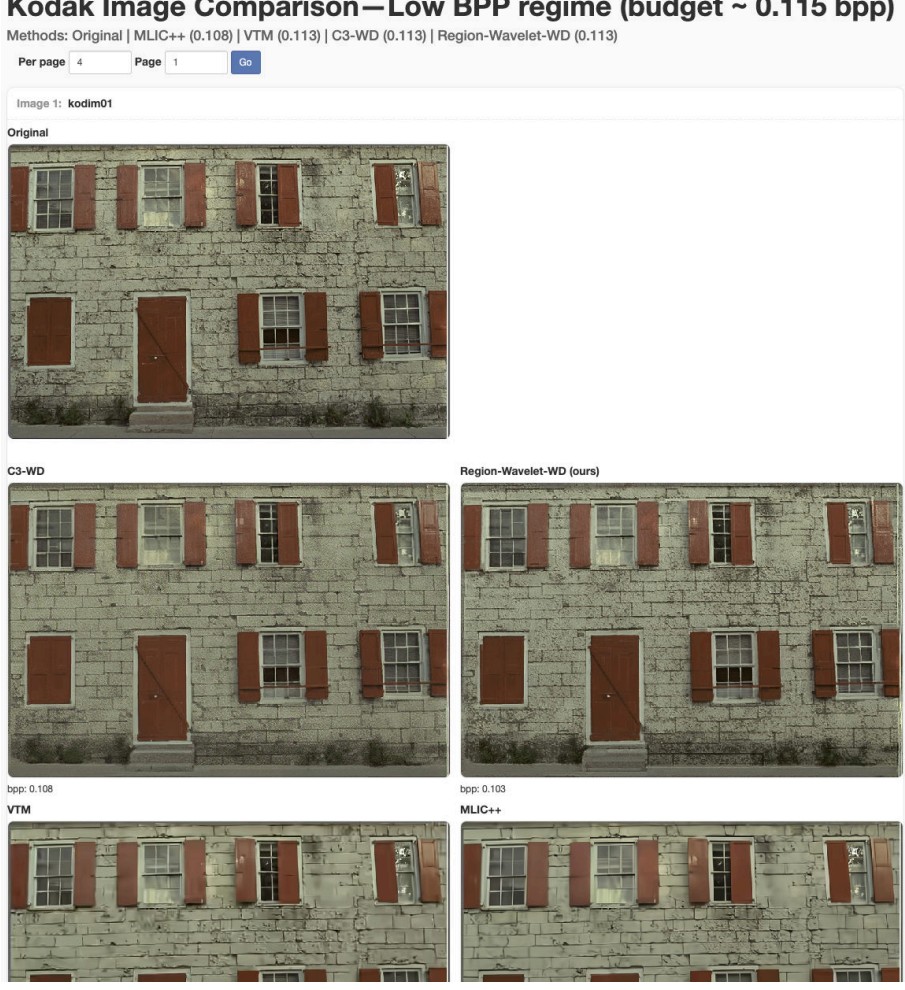

*Figure 8.* Example of our interactive comparison interface in the project page, which supports zooming and flip-toggle comparisons across methods, enabling clear visualization of the perceptual advantages of our approach.

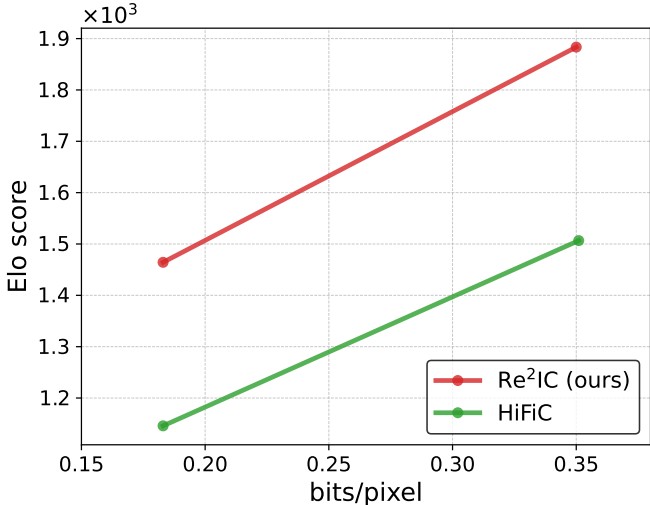

*Figure 9.* Human rating results for Re$^2$IC and HiFiC across more rates.

# D. Re$^2$IC implementation

## D.1. Region partition and contour compression

We partition images into regions using saliency predictions. An EML-Net (Jia & Bruce, 2020) is employed to generate a saliency map, from which we extract $N-1$ connected components as sub-regions for LPN. While high-resolution images could benefit from more sub-networks for finer perception modeling, we find that one or two are typically sufficient for a competitive RP performance. Detailed illustrations of the region partitioning and contour coding strategy are provided below. The process first thresholds the saliency map to generate a binary mask, followed by dilation and morphological opening to remove noise and merge nearby salient regions. Each region's perceptual score, given by mean saliency $\times$ area, is used to rank and select the top-$N$ regions, discarding those below a minimum area ratio.

---

Region partition and contour coding

```
## Region partition:
regions = extract_top_saliency_regions(saliency_map,N=2,dilate_kernel_size=3,
          threshold=np.percentile(saliency_map, 90), min_area_ratio=0.005)
## Chain coding for contour:
total_bits = contour_compression(input_contour, step_length, step_angle)
```

---

## D.2. Quantization and entropy coding details

In Re$^2$IC, the latent modulations $\boldsymbol{z^i}$, LPN parameters $\boldsymbol{W^i}$, and those of ARM and GPM are quantized and entropy-coded using the same methods with (Kim et al., 2024; Ladune et al., 2024; Wu et al., 2025). During training, latent modulation $\boldsymbol{z^i}$ is soft-rounded with Kumaraswamy noise, while hard rounding is applied at inference. Rate terms for $\{\boldsymbol{\tau^i}, \boldsymbol{\theta}, \boldsymbol{W^i}, \boldsymbol{\psi}\}$ are omitted from the loss due to their negligible contributions and simplified optimization. In practice, network parameters are quantized and entropy-coded with a non-learned distribution.

**Quantization-aware optimization.** Each $\boldsymbol{z^i}$ is optimized in a continuous space during the training, with quantization approximated via soft-rounding and Kumaraswamy noise for differentiability in two stages:

$$\hat{\boldsymbol{z}}^{\boldsymbol{i}} = \begin{cases} \mathcal{S}_T(\boldsymbol{z^i}) + \boldsymbol{u}_{kum}, & \text{Stage I (80,000 steps)} \\ Q(\boldsymbol{z^i}), & \text{Stage II (8000 steps),} \end{cases} \tag{27}$$

where $\mathcal{S}_T$ is the temperature for soft rounding, $Q$ is hard rounding (same with the inference phase), and $\boldsymbol{u}_{kum}$ is Kumaraswamy noise. Noise and temperature are gradually annealed for stable training.

**Entropy coding.** For entropy coding, we adopt a factorized auto-regressive model (ARM) $r_\psi$ (Ballé et al., 2018; Ladune et al., 2023). Each latent element $\hat{z}_{k,j}^i$ ($i$-th region, $k, j$ position) is conditioned on $C$ neighbors $\boldsymbol{c_{k,j}^i} \in \mathbb{Z}^C$: $p_\psi(\hat{\boldsymbol{z^i}}) = \prod_{k,j} p_\psi(\hat{z}_{k,j}^i | \boldsymbol{c_{k,j}^i})$, where $p_\psi(\hat{\boldsymbol{z^i}})$ is modeled as an integrated Laplace distribution $p_\psi(\hat{z}_{k,j}^i | \boldsymbol{c_{k,j}^i}) = \int_{\hat{z}_{k,j}^i - 0.5}^{\hat{z}_{k,j}^i + 0.5} g(z) dz$, and $g \sim \mathcal{L}(\mu_{k,j}^i, \sigma_{k,j}^i)$, $\mu_{k,j}^i, \sigma_{k,j}^i = r_\psi(\boldsymbol{c_{k,j}^i})$. With range coding [4], the rate for $\hat{\boldsymbol{z^i}}$ becomes:

$$R(\hat{\boldsymbol{z^i}}) = -\log_2 p_\psi(\hat{\boldsymbol{z^i}}) = \sum_{k,j} -\log_2 p_\psi(\hat{z}_{k,j}^i | \boldsymbol{c_{k,j}^i}). \tag{28}$$

The final bit cost for all latent vectors are $\boldsymbol{b_z} = \sum_{i=0}^{N} R(\hat{\boldsymbol{z^i}})$.

**Model compression.** Parameters of all networks are quantized with scalar quantizers $Q(\cdot, \Delta)$ and entropy-coded using Laplace priors ($\boldsymbol{W^i}$ with a step size $\Delta$ as an example): $\hat{\boldsymbol{W}}^{\boldsymbol{i}} = Q(\boldsymbol{W^i}, \Delta_{\boldsymbol{W^i}})$, and $p(\hat{W}_k) =$

---

[4] We employ the range-coder in PyPI for range-coding. Note that here fixed-point fractional-multiplier quantization is necessary to ensure bit-exact consistency for reliable range decoding.

---

**Algorithm 1** Encoding stage of the $\text{Re}^2\text{IC}$

---

**Input:** Source image $\boldsymbol{S}$, coordinate $\boldsymbol{x}^i$, Learning rates for Stage I and II: $\alpha$, $\beta$,
Latent and random vectors: $(\boldsymbol{z}^i, \boldsymbol{M}^i)$, and networks $(\boldsymbol{W}^i, \boldsymbol{\theta}^i, \text{ and } \boldsymbol{\psi})$,
**Output:** Bits stream of $\boldsymbol{b_z}, \boldsymbol{b_\tau}, \boldsymbol{b_W}, \boldsymbol{b_\theta}, \boldsymbol{b_\psi}$.

1: $\boldsymbol{S_a} = \text{EML}_{net}(\boldsymbol{S})$, $\boldsymbol{S_a} \to \{\boldsymbol{S}^1, \ldots, \boldsymbol{S}^N\}; \{\boldsymbol{\tau}^1, \ldots, \boldsymbol{\tau}^N\}; \{\boldsymbol{x}^1, \ldots, \boldsymbol{x}^N\}$,        # Partition the source

2: $\boldsymbol{z}^i \triangleq \{\boldsymbol{z}_1^i, \ldots, \boldsymbol{z}_L^i\}, \boldsymbol{M}^i \triangleq \{\boldsymbol{m}_1^i, \ldots, \boldsymbol{m}_L^i\}$ with each $\boldsymbol{z}_k^i, \boldsymbol{m}_k^i \in \mathbb{R}^{\frac{K^i}{4^{k-1}}}$     # Initialize latent vectors

3: **for** the $k$-step within the Stage I **do**

4:  **for** each $i$-th region within the $N$ regions **do**

5:    $\hat{\boldsymbol{z}_i} = \mathcal{S}_T(\boldsymbol{z}_i) + \boldsymbol{u}_{kum}, \hat{\boldsymbol{S}_i} = f_{\boldsymbol{W}^i}(f_{\boldsymbol{\theta}}(\hat{\boldsymbol{z}^i}, \boldsymbol{M}^i), \boldsymbol{x}^i)$      # Quantization-aware training

6:  **end for**

7:  $\hat{\boldsymbol{S}} = \bigcup_{i=1}^N \hat{\boldsymbol{S}^i}$,                      # Merge regions

8:  $\mathcal{L}(\boldsymbol{\Omega}) = WD_{\text{wave}}(\boldsymbol{S}, \hat{\boldsymbol{S}}) + \sum_{i=0}^N \lambda R(\hat{\boldsymbol{z}^i})$,      # Compute the PD loss function

9:  $\boldsymbol{\theta} \leftarrow \boldsymbol{\theta} - \alpha \nabla_{\boldsymbol{\theta}} \mathcal{L}$,              # Update the GPM for all regions

10:  $\boldsymbol{z}^i \leftarrow \boldsymbol{z}^i - \alpha \nabla_{\boldsymbol{z}^i} \mathcal{L}$,           # Update the latent for each region

11:  $\boldsymbol{W}^i \leftarrow \boldsymbol{W}^i - \alpha \nabla_{\boldsymbol{W}^i} \mathcal{L}$,         # Update the LPN for each region

12:  $\boldsymbol{\psi} \leftarrow \boldsymbol{\psi} - \alpha \nabla_{\boldsymbol{\psi}} \mathcal{L}$,           # Update ARM for all regions

13: **end for**

14: **for** the $k$-th step within the Stage II **do**

15:  **for** each $i$-th region within the $N$ regions **do**

16:    $\hat{\boldsymbol{z}^i} = Q(\boldsymbol{z}^i), \hat{\boldsymbol{S}_i} = f_{\boldsymbol{W}^i}(f_{\boldsymbol{\theta}}(\hat{\boldsymbol{z}^i}, \boldsymbol{M}^i), \boldsymbol{x}^i)$,       # Hard rounding

17:  **end for**

18:  $\hat{\boldsymbol{S}} = \bigcup_{i=1}^N \hat{\boldsymbol{S}^i}$,                    # Merge regions

19:  $\mathcal{L}(\boldsymbol{\Omega}) = WD_{\text{wave}}(\boldsymbol{S}, \hat{\boldsymbol{S}}) + \sum_{i=0}^N \lambda R(\hat{\boldsymbol{z}^i})$,    # Compute the PD loss function

20:  $\boldsymbol{\theta} \leftarrow \boldsymbol{\theta} - \beta \nabla_{\boldsymbol{\theta}} \mathcal{L}$,              # Update the GPM for all regions

21:  $\boldsymbol{z}^i \leftarrow \boldsymbol{z}^i - \beta \nabla_{\boldsymbol{z}^i} \mathcal{L}$,           # Update the latent for each region

22:  $\boldsymbol{W}^i \leftarrow \boldsymbol{W}^i - \beta \nabla_{\boldsymbol{W}^i} \mathcal{L}$,         # Update the LPN for each region

23:  $\boldsymbol{\psi} \leftarrow \boldsymbol{\psi} - \beta \nabla_{\boldsymbol{\psi}} \mathcal{L}$,           # Update ARM for all regions

24: **end for**

25: $\boldsymbol{b_z} = \sum_{i=0}^N \mathcal{A}(Q(\boldsymbol{z}^i))$,          # Quantization and entropy coding over $\boldsymbol{z}^i$

26: $\boldsymbol{b_\tau} = \sum_i^N \mathcal{C}(\boldsymbol{\tau}^i)$,           # Lossy chain coding for contours

27: $\Delta_\theta, \Delta_{\boldsymbol{W}^i}, \Delta_\psi = \mathcal{G}(\boldsymbol{\theta}, \boldsymbol{W}^i, \boldsymbol{\psi})$     # Search for optimal quantization steps for networks

28: $\boldsymbol{b_\theta} = \mathcal{A}(Q(\boldsymbol{\theta}, \Delta_\theta)), \boldsymbol{b_\psi} = \mathcal{A}(Q(\boldsymbol{\psi}, \Delta_\psi)) \, \boldsymbol{b_W} = \sum_{i=0}^N \mathcal{A}(Q(\boldsymbol{W}^i, \Delta_{\boldsymbol{W}^i}))$,
                 # Quantization and entropy coding over all network parameters

---

**Algorithm 2** Decoding stage of the $\text{Re}^2\text{IC}$

---

**Input:** Bits stream of $\boldsymbol{b_z}, \boldsymbol{b_\tau}, \boldsymbol{b_W}, \boldsymbol{b_\theta}, \boldsymbol{b_\psi}$.
**Output:** Reconstruction of image $\hat{\boldsymbol{S}}$.

1: $\hat{\boldsymbol{\tau}}^i, \boldsymbol{x}^i \leftarrow \mathcal{C}(\boldsymbol{b_\tau})$,          Decode contour to identify region partition

2: $\hat{\boldsymbol{\psi}} = Q(\mathcal{A}(\boldsymbol{b_\psi})), \hat{\boldsymbol{\theta}} = Q(\mathcal{A}(\boldsymbol{b_\theta})), \hat{\boldsymbol{W}}^i = Q(\mathcal{A}(\boldsymbol{b_{W^i}}))$,      Decode networks

3: $\hat{\boldsymbol{z}^i} = Q(\mathcal{A}_{\hat{\tau}}(\boldsymbol{b_z})), \boldsymbol{M}^i \sim \mathcal{N}(0, 1)$,  Decode the latent vectors and synchronize a random commonness

4: **for** each $i$ region within $N$ regions **do**

5:  $\hat{\boldsymbol{S}_i} = f_{\hat{\boldsymbol{W}}^i}(f_{\hat{\boldsymbol{\theta}}}(\hat{\boldsymbol{z}^i}, \boldsymbol{M}^i), \boldsymbol{x}^i)$,         # Reconstruct each local region

6: **end for**

7: $\hat{\boldsymbol{S}} = \bigcup_{i=1}^N \hat{\boldsymbol{S}^i}$,                Merge the source image

---

$\int_{\hat{W}_k - 0.5}^{\hat{W}_k + 0.5} g(W) dW$, with $g \sim \mathcal{L}(0, \text{stddev}(\hat{\boldsymbol{W}}))$. The total parameter rate is:

$$R_{\text{MLP}} = -\sum_{k,i} \log_2 p(\hat{W}_k^i) - \sum_k \log_2 p(\hat{\theta}_k) - \sum_k \log_2 p(\hat{\psi}_k). \tag{29}$$

Quantization steps are searched via greedy PD optimization:

$$\min_{\Delta \hat{W}^i, \Delta \hat{\psi}, \Delta \hat{\theta}} d(\boldsymbol{S^i}, \bigcup_{i=1}^N [f_{\hat{\boldsymbol{W}}^i}(f_{\hat{\boldsymbol{\theta}}}(\hat{\boldsymbol{z}^i}), \boldsymbol{x}^i)]) + \lambda \sum_i^N R(\hat{\boldsymbol{z}^i}) + \lambda R_{MLP}. \tag{30}$$

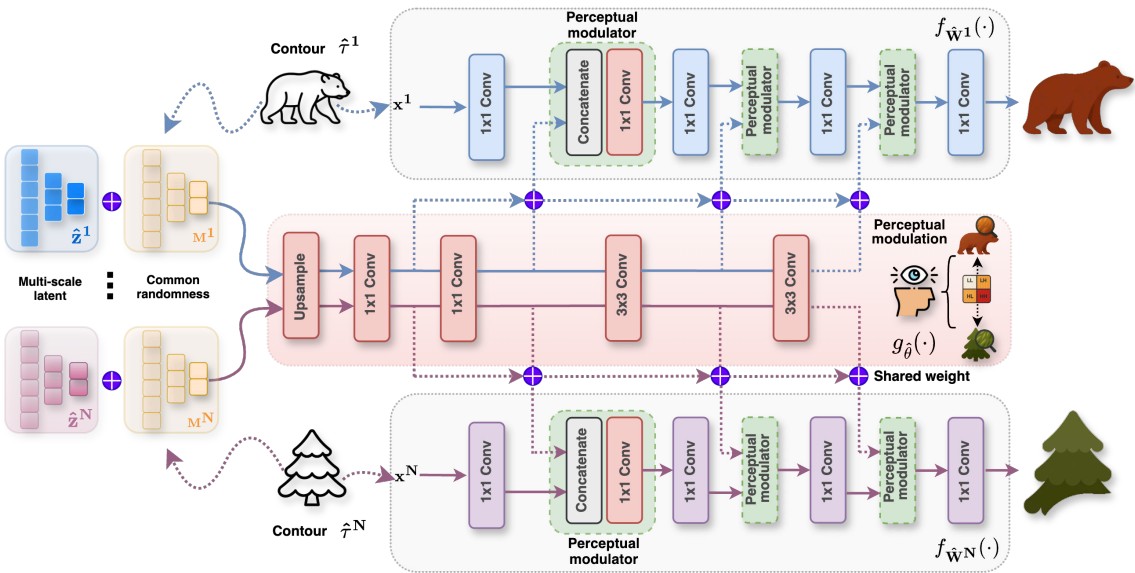

*Figure 10.* Global-local design of Re$^2$IC: each LPN is optimized with WA-WD to model visual perception, while red GPM blocks denote parameter-shared components that inject global context by modulating multi-scale features, thereby mitigating artifacts from pure local overfitting.

**Model architecture.** For the ARM model with 24 contextual inputs, we use three linear (or $1 \times 1$ convolutional) layers with GELU activations, mapping $(24 \times 24) \to (24 \times 24) \to (24 \times 2)$.

The GPM model takes $L = 7$ resolution-layered latent vector as input with 4 convolutional layers, mapping $(7 \times 18) \to (18 \times 3) \to (3 \times 3) \to (3 \times 3)$. Specifically, as shown in Fig. 10, GPM $f_{\theta}$ consists of an upsampling module and 4 convolutions. The quantized $\hat{z}^i$ is first upsampled by transposed convolutions to $U_0^i$, then to generate hierarchical modulation vectors for each LPN: $U_k^i = f_{\theta}^{(k)}(U_{k-1}^i)$, where $U_{k-1}^i$ denotes the $k$-th layer output of $g_{\theta}$ for $i$-the region. Inspired by (Mehta et al., 2021; Wu et al., 2025; Perez et al., 2018), GPM modulates each region perception in a concatenated fashion: $M_k^i \triangleq \text{Concatenate}(U_1^i, \dots, U_k^i)$ for the $k$-th layer of $f_{W^i}$. At each layer of LPN, $M_k^i$ is concatenated with the current output and processes it through a shared MLP, enabling consistent global context injection and dynamic refinement.

Each LPN maps pixel coordinates $x^i$ to RGB with 4 layers, modulated by $\mathcal{M}_k^i$ using modulations from $f_{\theta}$. The input and output dimensions are given as: $(2 \times 3) \to ([3 + 18 + 3] \times 3) \to (3 \times 3) \to ([3 + 18 + 3 + 3] \times 3) \to (3 \times 3) \to ([3 + 18 + 3 + 3 + 3] \times 3) \to (3 \times 3)$, where the orange MLP layers are globally shared across regions.

**Optimization details.** We introduce a 700-step warm-up stage to stabilize region-based coding, training 7 candidates with different random seeds and retaining the best for the remaining training. Adam optimizer with a learning rate of $1e - 4$ is employed. For WA-WD optimization, we use a single-level DWT to reduce computation while maintaining satisfactory performance, though further gains are expected with deeper decompositions of DWT.

**Model parameters.** Table 4 summarizes all parameter settings in our experiments. To support future research and reproducibility, we release the full $\lambda$ configuration used in our experiments. (We note that rate selection across methods, particularly under WD or WA-WD optimization, costs a lot of computational resources to ensure fair comparison.)

**Pseudocode for the algorithm.** A detailed pseudocode of encoding and decoding process of Re$^2$I is presented in Algorithm 1 and Algorithm 2, respectively.

*Table 4.* Hyper-parameter settings, with all reconstruction results provided on our project page.

| **Main experiments** | | |
|---|---|---|
| Values of $\lambda$ for Kodak | $\{4, 8.9, 23\}$ | |
| Values of $\lambda$ for CLIC2020 | $\{1.1, 4.3, 15.5\}$ | |
| **Abaltion experiments** | | |
| Values of $\lambda$ for Tuning WA-WD on Kodak | $\{3, 7, 20.5\}$ | |
| Values of $\lambda$ for Ablation study on Kodak w/o common randomness | $\{4, 8.9, 23\}$ | |
| Values of $\lambda$ for Ablation study on Kodak w/o LPN | $\{6, 12, 26\}$ | |
| Values of $\lambda$ for Ablation study on Kodak w/o GPM | $\{5.1, 11.2, 27\}$ | |
| Values of $\lambda$ for Ablation study on Kodak w/o partition | $\{5.7, 11.5, 25\}$ | |
| Values of $\lambda$ for Ablation study on Kodak w/o wavelet | $\{4.5, 10, 25\}$ | |
| Values of $\lambda$ for RP convergence on Kodak for C3/WDs and Re$^2$IC | $\{27$ vs $28\}$ | |
| Values of $\lambda$ for RD convergence on Kodak | $\{8e^{-3}, 2e^{-3}, 1e^{-3}, 5e^{-4}, 2e^{-4}, 1e^{-4}\}$ | |
| **Hyper parameter** | **Initial values** | **Final values** |
| **Quantization – Stage I** | | |
| Learning rate $\beta$ | $10^{-2}$ | $0$ |
| Temperature $T$ for soft rounding | $0.3$ | $0.1$ |
| Noise strength $\alpha$ for Kumaraswamy noise | $2.0$ | $1.0$ |
| Number of encoding steps | $8 \times 10^4$ | |
| Scheduler for learning rate / Soft-rounding with Kumaraswamy noise | Cosine / Linear scheduler | |
| **Quantization – Stage II** | | |
| Learning rate | $10^{-4}$ | $10^{-8}$ |
| Number encoding steps | $8 \times 10^3$ | |
| Decay learning rate if loss has not improved for this many steps | $40$ | |
| Decay factor | $0.8$ | |
| Temperature $T$ for soft rounding | $10^{-4}$ | |
| **Warm-up and WA-WD setting** | | |
| Round 1 candidates / steps | 7 / 400 | |
| Round 2 candidates / steps | 3 / 400 | |
| $\sigma_{max}$ for Kodak / CLIC2020 | 8 / 16 | |
| DWT level / PMF level / VGG slices | 1 / 5 / 5 | |
| Salient network / Salient region numbers | EML-Net / $1 \sim 2$ | |
| **Architecture** | | |
| **Entropy model** | | |
| Output channel numbers for ARM model | $(24 \times 24) \rightarrow (24 \times 24) \rightarrow (24 \times 2)$ | |
| Activation function | GELU | |
| Log-scale of Laplace is shifted before $\exp$ | $4$ | |
| Scale parameter of Laplace is clipped to | $[10^{-2}, 150]$ | |
| **Latent modulations** | **Values** | |
| Number of latent vectors $L$ / Initialization of $\boldsymbol{z}$ | 7/0 | |
| **Global perceptual modulator** | **Values** | |
| Upsampling kernel | $8 \times 8$ | |
| Output channels of the $1 \times 1$ convolutions / $3 \times 3$ convolutions | $(7 \times 18) \rightarrow (18 \times 3)/(3 \times 3) \rightarrow (3 \times 3)$ | |
| Modulation vector dimension | $\{3 \rightarrow 3\}$ | |
| **Local perceptual network** | **Values** | |
| Output dimensions of each layer | $\{2 \rightarrow 3 \rightarrow 3 \rightarrow 3\}$ | |
| Out-channels of each shared perceptual modulator | | |
| $\{[3 + 18 + 3] \rightarrow 3\}; \{[3 + 18 + 3 + 3] \rightarrow 3\}; \{[3 + 18 + 3 + 3 + 3] \rightarrow 3\}$ | | |

# E. Wavelet-Wasserstein distortion

## E.1. Wasserstein distortion

**Definition.** Similar to LPIPS (Zhang et al., 2018) and DISTS (Ding et al., 2022), WD (Qiu et al., 2024) is a perceptual distance metric that measures image similarity in a feature-embedding space. Specifically, WD incorporates models of foveal and peripheral vision by computing pointwise feature-space distances with a spatially varying $\sigma$-map, *typically derived from saliency map*, which determines how permissive the model is to *texture resampling*.

Let $\boldsymbol{x} = \{x_n\}_{n=-\infty}^{\infty}$ denote the source sequence with infinite length (with straightforward extension to images) and $\boldsymbol{z} = \{z_n\}_{n=-\infty}^{\infty}$ with $z_n = \phi(\boldsymbol{x}) \in \mathbb{R}^d$ denote its feature representation from a neural network $\phi$. WD is defined through a family of pooling probability mass functions (PMFs) $q_\sigma(k)$ (parameterized by a $\sigma$-map) over integers $k$, satisfying the *two-sided geometric distribution*[5]:

$$q_\sigma(k) = \begin{cases} \frac{e^{1/\sigma}-1}{e^{1/\sigma}+1} \cdot e^{-|k|/\sigma} & \text{if } \sigma > 0 \\ 1 & \text{if } \sigma = 0 \text{ and } k = 0 \\ 0 & \text{otherwise.} \end{cases} \tag{31}$$

This induces local empirical measures $\boldsymbol{y}_\sigma = \{y_{n,\sigma}\}_{n=-\infty}^{\infty}$, where $y_{n,\sigma} \triangleq \sum_{k=-\infty}^{\infty} q_\sigma(k)\delta_{z_{n+k}}$, and $\delta$ is Dirac delta measure. Each $y_{n,\sigma}$ represents the *pooled feature statistics* around location $n$ with an *effective pooling width* $\sigma \geq 0$.

For a reconstruction $\hat{\boldsymbol{x}} = \{\hat{x}_n\}_{n=-\infty}^{\infty}$ with features $\hat{\boldsymbol{z}} = \{\hat{z}_n\}_{n=-\infty}^{\infty}$ and pooled measures $\hat{\boldsymbol{y}}_\sigma = \{\hat{y}_{n,\sigma}\}_{n=-\infty}^{\infty}$, the local distortion is:

$$D_{n,\sigma} = W_p^p\left(y_{n,\sigma}, \hat{y}_{n,\sigma}\right), \tag{32}$$

where $W_p$ denotes the Wasserstein distance of order $p$ (Villani, 2008):

$$W_p(\rho, \hat{\rho}) = \inf_{Z \sim \rho, \hat{Z} \sim \hat{\rho}} \mathbb{E}\left[d^p(Z, \hat{Z})\right]^{1/p}, \tag{33}$$

where $\rho$ and $\hat{\rho}$ are probability measures on $\mathbb{R}^d$ and $d(\cdot)$ is a metric.

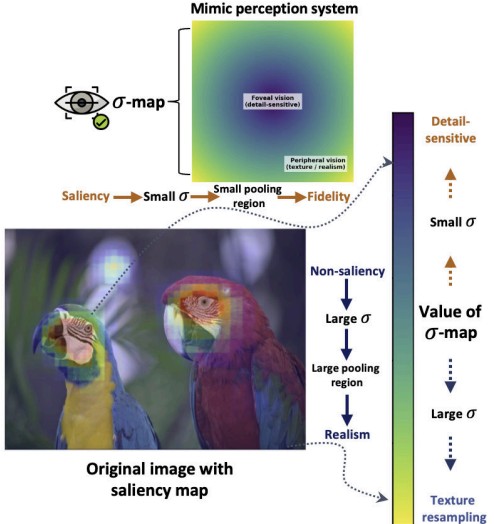

*Figure 11.* WD for foveal-peripheral vision.

The overall WD is the spatial average as:

$$WD(\boldsymbol{x}, \hat{\boldsymbol{x}}) = \frac{1}{2N+1}\sum_{n=-N}^{N} D_{n,\sigma(n)} = \frac{1}{2N+1}\sum_{n=-N}^{N} W_p^p\left(y_{n,\sigma(n)}, \hat{y}_{n,\sigma(n)}\right),$$

where $\sigma(n)$ is the spatially varying pooling width (i.e., the $\sigma$-map).

**Interpretation.** As illustrated in Fig. 11, saliency assigns spatially varying $\sigma$ values that modulate pooling statistics and balance the trade-off between realism and fidelity. When the $\sigma$-map approaches 0 in regions of high visual importance (e.g., foveal or salient areas), WD reduces towards a pointwise distance (can be equivalent to MSE), emphasizing fine detail. Conversely, larger $\sigma$ values mimic peripheral vision by expanding the pooling extent, thereby prioritizing realistic texture over fine-grained accuracy.

In summary, the $\sigma$-map determines the *pooling extent* used in the distortion computation and is explicitly shaped to reflect *foveal and peripheral vision*, thereby mimicking human perceptual behavior. In the context of *image compression*, $\sigma$-map controls how permissive WD is to *texture resampling*, **which in turn enables more flexible *bit allocation*.**

---

[5]The PMF is designed for an adjustable, visually compliant pooling kernel without spectral zeros. This also precisely motivates our WA-WD design, where wavelet subbands offer a *spectrally complete* and *human vision consistent* feature basis that **reinforces** these theoretical requirements.

**Theoretical properties.** The key theoretical properties of WD can be summarized as:

- **Property 1. Fidelity–Realism Parameterization:** It unifies *fidelity* (small $\sigma$) and *realism* (large $\sigma$) within a *single*, continuously parameterized framework by allowing $\sigma$ to vary over the full real-valued range.

- **Property 2. Metric Consistency:** If the PMF $q_\sigma(\cdot)$ has *no spectral nulls* and the base distance $d$ *is a metric*, then $WD(\boldsymbol{z}, \hat{\boldsymbol{z}})^{1/p}$ itself satisfies positivity, symmetry, and the triangle inequality, thus *is a metric*. If $\phi$ is *invertible*, $WD(\boldsymbol{x}, \hat{\boldsymbol{x}})^{1/p}$ *is also a metric*.

- **Property 3. Saliency-Adaptive Bit Allocation:** When combined with *human-perception saliency*, WD enables *flexible bit allocation* that adapts to varying fidelity–realism requirements across regions in image compression, leading to improved RP performance.

**Practical implementation.** For computational efficiency (Olkin & Pukelsheim, 1982), Eqn.33 can be replaced by a proxy (Dupont et al., 2022b; Heusel et al., 2017; Liu et al., 2021):

$$|\mu - \hat{\mu}|_2^2 + \text{Tr}\left(C + \hat{C} - 2(\hat{C}^{1/2} C \hat{C}^{1/2})^{1/2}\right), \tag{34}$$

which is equivalent to a 2-Wasserstein distance when $p = 2$, $d$ is Euclidean distance, and the distributions of probability measures $\rho$ are Gaussian with means $\mu$ and covariances $C$. In papers (Qiu et al., 2024; Ballé et al., 2025), a simplified version of Eqn. 34 is employed as:

$$\sum_{i=1}^{d} (\mu_i - \hat{\mu}_i)^2 + \left(\sqrt{V_i} - \sqrt{\hat{V}_i}\right)^2, \tag{35}$$

where $\mu_i$ and $V_i$ are the mean and variance of the $i$-th component. [6]

Even under the independent Gaussian assumption, computing WD in image compression for learned codec optimization still remains computationally impractical, as it requires aggregating local statistics using a distinct pooling kernel at every spatial location. A practical solution is to approximate WD by discretizing $\sigma$ to powers of two, replacing the original PMF with Gaussian downsampling kernels, and leveraging the inherent downsampling in VGG, together serving as an implicit pooling (PMFs) mechanism (see (Ballé et al., 2025) for details).

While efficient and empirically effective, these assumptions can be unrealistic in practice: (1). Neural features are rarely independent; (2). Log-scale approximations for WD at different $\sigma$ values can exacerbate correlation effects; (3). The design of implicit PMFs may introduce spectral blind points, leading to metric degeneracy for *Property 2* of WD. These considerations motivate our WA-WD formulation, which strengthens the robustness of WD, with more details provided in Appendix E.2.

### E.2. Towards Wavelet-WD

Although WD provides a principled foundation for perceptual compression, several practical challenges remain, including the independence assumption over features, the implicit PMF approximation, and unstable gradients across mixed-frequency content. Motivated by human visual perception and by the need for fine-grained frequency control in implicit codecs, we introduce WA-WD for Re²IC. An illustration of WA-WD is presented in Fig. 12. This section outlines the motivation, interpretation, theoretical properties, validation experiments, and tunable strategy for this design.

### Motivation

- **Consistency with Human Visual System (HVS)** Visual neuroscience shows that early vision performs multi-scale, orientation-selective filtering (V1) followed by larger pooling (V2) (Freeman & Simoncelli, 2011; Balas et al., 2009). WD models V2 via its $\sigma$-map pooling, while wavelet subbands naturally mirror V1 stage. WA-WD actually follows this two-stage HVS structure in feature space, providing an interpretable motivation: wavelet decomposition (V1-like) + WD pooling (V2-like) yields a distortion that captures human sensitivity to structure and texture.

---

[6]However, Eqn. 35 *assumes feature components are uncorrelated*, an assumption that rarely holds in practice. Our WA-WD addresses this by explicitly modeling intra-block correlations before WD computation.

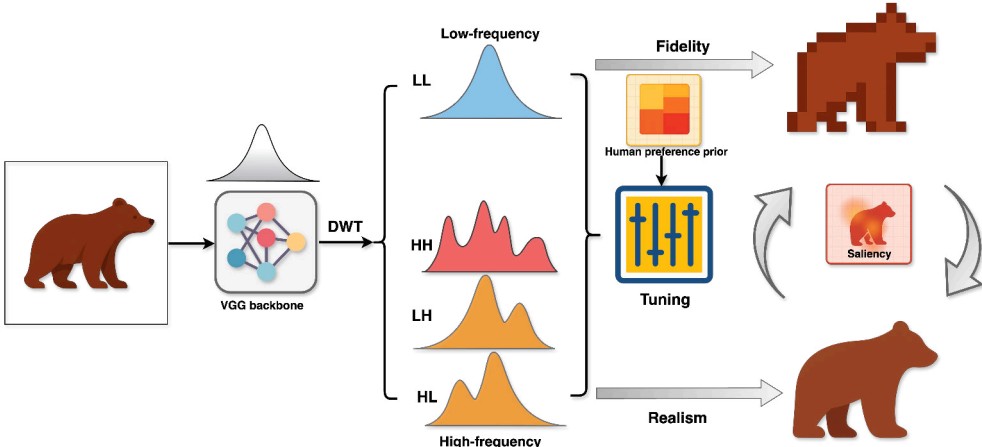

*Figure 12.* Illustration of Wavelet-WD: features are decomposed into subbands. The LL band captures structural content and can be optimized with a lower $\sigma$ range, while other bands can be adapted into a slightly higher $\sigma$ range to better model perceptual realism.

- **Inheriting the metric property from WD.** A core design principle of WA-WD is to retain the desirable theoretical properties of the original WD (Section E.1) while improving perceptual alignment and region-wise controllability for compression. Rather than modifying the WD formulation, we apply an invertible orthonormal wavelet transform to the feature representation and compute WD independently within each subband. Since $W$ is orthonormal and invertible, the metric properties of WD are preserved under this transformation: if $WD(\cdot, \cdot)^{1/p}$ is a *metric*, then $WD_{\text{wave}}(\cdot, \cdot)^{1/p}$ is *also* a metric (see Theorem 3.1).

- **Making the independence assumption more justified.** Practical WD computation relies on approximating local feature statistics as independent Gaussians (Ballé et al., 2025), assumptions that are often violated due to strong spatial and cross-channel correlations in neural features. Wavelet subbands, however, are known to be approximately decorrelated across scale and orientation for natural images (Figueiredo & Nowak, 2001; Simoncelli, 1999), making them far more compatible with the diagonal-covariance model used in WD approximation. Computing WD per subband thus make its statistical assumptions hold more closely. As formalized in Corollary 3.2, WA-WD provides a tighter bound on the approximation error, resulting in more perceptually consistent optimization for compression.

- **Fine-grained optimization.** Optimizing WD purely via saliency can make high-frequency variance dominate, suppressing structural gradients and causing drift in non-salient regions. As a result, WD may overlook regional structure and frequency characteristics, leading to over-resampled textures in smooth non-salient areas (Fig. 31) or excessive fidelity in high-frequency salient regions (Fig. 32). To address this, LL/LH/HL/HH are optimized separately in Re$^2$IC, where LL provides clean structural-fidelity signals, while the high-frequency bands guide texture realism. Combined with the design of Re$^2$IC, this frequency and region-aware optimization can yield more perceptually consistent compression while allowing tunable subband/region weights for better HVS alignment.

- **Tunable metric.** Serving as a metric, WD is computed directly in neural feature space, making its behavior largely determined by the backbone and therefore difficult to adjust. WA-WD introduces subband-level control (via tunable weights or $\sigma$-ranges). This allows adaption to different perceptual preferences or degradation characteristics and provides a more flexible RDP trade-off.

**Interpretation**

- Design of $\sigma$-map: $\sigma$ is a spatial map controlling the pooling size. For small $\sigma$, WA-WD models foveal vision under different sub-bands, which enforces high-fidelity reconstruction. For large $\sigma$, it models peripheral vision with more tolerance to texture resampling. Thus, $\sigma$ directly controls the realism–fidelity trade-off in a way that mimics the HVS.

*Table 5.* Comparison of standard overfitted codec with WD and Re$^2$IC with WA-WD, where the advantages are summarized in green.

| Difference | WD codec (C3/WDs) | WA-WD codec (Re$^2$IC) |
|---|---|---|
| Feature space (Frequency-aware) | VGG feature domain. Optimization with mixed frequency gradients. | Wavelet-decomposed space. Optimization with **frequency-aware gradients**. |
| HVS alignment (Explicit and more aligned) | HVS-V2 processing. Modeling fovea-to-periphery integration with WD $\tilde{\leftrightarrow}$ HVS-V2. | HVS-V1 + HVS-V2 processing. DWT for **explicit** multi-scale, orientation decomposition $\tilde{\leftrightarrow}$ HVS-V1; WD $\tilde{\leftrightarrow}$ HVS-V2. |
| Feature assumption (Better justified) | Independent Gaussian over VGG. Neural features are highly correlated, so the independence assumption is often violated. | Independent Gaussians after DWT. DWT yields **orthogonal, decorrelated** subbands, making independence assumptions more realistic. |
| Granularity control (Adaptive tuning) | Coarse and scale-only control. Perception modeling (pooling) is controlled only by the pyramid scale, smoothing all frequencies jointly at each level. | Fine-grained and subband-wise control. Perception modeling is performed per subband, enabling **subband-wise control** and **frequency-aware modulation** (flexible structure, texture modeling). |
| Coding strategy (Towards "Faster, cheaper, better") | Standard instance-wise overfitting. Standard overfitting each sample with a WD target, where each sample is fitted via a single neural function. | Region-wise perception modeling. Modeling perception region-by-region leads to **faster** convergence with **lower-complexity**, while achieving a **better** RP performance. |
| Tunable metric (More aligned to human preference) | WD computed in the VGG feature space. WD is computed directly in the neural space, whose behavior is largely determined by the network backbone, with limited flexibility. | Tunable WD in decomposed feature space. Each subband's contribution to WD can be **independently tuned**, enabling **adaptation** to various preference and a more **flexible** R-D-P trade-off. |

- Frequency-aware optimization: Unlike WD, which mixes all frequencies, WA-WD separately optimizes distortions in structural (LL) and textural (LH/HL/HH) subbands. As a result, WA-WD measures perceptual differences in *multi-scale, frequency-decomposed feature statistics*, producing frequency-aware distortion and gradient that supports fine-grained DP control and more flexible, perceptually aligned bit allocation.

In summary, WA-WD preserves the core structure of WD while placing it in a more *interpretable* and *HVS-consistent* form for image compression. Specifically, this change has several practical implications: the *independence assumptions* underlying the WD approximation hold more closely and the *subband-wise decomposition* naturally supports region-wise perceptual modulation. Together, these properties improve the *stability*, *flexibility*, and *effectiveness* of bit allocation for RP optimization in image compression, while preserving the original metric behavior of WD. A detailed comparison between Re$^2$IC (WA-WD) and C3 (WDs) is provided in Table 5.

**Theoretical Properties** WA-WD satisfies several desirable theoretical properties:

- **Metric preservation.** As shown in Theorem 3.1, WA-WD inherits the metric property of WD: if $WD(\cdot, \cdot)^{1/p}$ is a *metric*, then $WD_{\text{wave}}(\cdot, \cdot)^{1/p}$ is *also* a metric.

- **Tighter approximation.** As shown in Corollary 3.2, WA-WD better satisfies the independent-Gaussianized approximation used in WD, yielding a tighter bound on the resultant approximation error.

- **Statistics properties.** As shown in Corollary B.5, compared with WD, WA-WD preserves a similar mean but differs in variance, where it incorporates cross-covariance terms and makes the metric sensitive to intra-block correlations. When $\sigma = 0$, both WD and WA-WD can reduce to mean squared error, measuring only fidelity distortion.

**Validation experiments** To evaluate the benefits of WA-WD for frequency-aware and region-wise optimization, we provide two illustrative examples in Fig. 31 and Fig. 32. Additional experiments over the loss function can be seen in Table 12. Experiments on tunable WA-WD are shown in Fig. 14. Experiments for the DWT effect can be seen in Figs. 15, 18, and 19. The comparisons with other DWT-based codec can be seen in Figs. 16, 17.

*Table 6.* Experiments for additional baselines: TACO (Lee et al., 2024) and CDC (Yang & Mandt, 2023) on CLIC2020, where we select the baseline with their most relevant rate regimes. **Red** and blue denote the best and second-best results. Re$^2$IC consistently delivers **state-of-the-art or leading** performance across FID, DISTS, KID, CLIP-IQA, and MANIQA, even compared with the current strongest model, TACO, at the target medium rate.

| | Model | BPP↓ | PSNR↑ | MS-SSIM↑ | LPIPS↓ | DISTS↓ | FID↓ | KID↓ | NIQE↓ | MUSIQ↑ | ClipIQA↑ | MANIQA↑ |
|---|---|---|---|---|---|---|---|---|---|---|---|---|
| Medium | VTM | 0.1480 | **32.39** | **0.9665** | 0.2495 | 0.1837 | 89.83 | 47.54 | 5.3504 | 57.34 | 0.4230 | **0.3521** |
| | HiFiC | 0.1420 | 29.83 | 0.9594 | 0.0589 | 0.0695 | 18.81 | 2.33 | **3.3044** | **59.91** | 0.5140 | 0.3390 |
| | TACO | 0.1425 | 30.79 | 0.9555 | **0.0394** | 0.0771 | 19.42 | 4.17 | 4.4278 | 58.95 | 0.5419 | 0.3168 |
| | Re$^2$IC | 0.1500 | 28.53 | 0.9422 | 0.0754 | **0.0381** | **11.82** | **1.41** | 4.4016 | 57.24 | **0.5509** | 0.3429 |
| High | HiFiC | 0.2720 | 31.84 | **0.9774** | 0.0380 | 0.0492 | 12.74 | 1.40 | **3.4455** | 60.23 | 0.5142 | 0.3406 |
| | CDC | 0.2439 | 27.51 | 0.9522 | 0.0836 | 0.0508 | 13.12 | 1.84 | 4.0203 | 59.78 | 0.4528 | 0.3453 |
| | TACO | 0.2390 | **33.45** | 0.9756 | **0.0351** | 0.0756 | 18.10 | 3.71 | 4.1096 | **60.73** | 0.5409 | 0.3379 |
| | Re$^2$IC | 0.2970 | 30.41 | 0.9636 | 0.0483 | **0.0187** | **7.10** | **0.80** | 4.1808 | 58.89 | **0.5492** | **0.3486** |

**Tunable strategy**    As an independent perceptual metric (shown in Fig. 12), WA-WD can be tuned for different scenarios. In this way, even within same saliency region, different bands can be optimized differently according to their characteristics. The design principle is to emphasize fidelity in salient regions with low-frequency features, while assigning more realism to non-salient regions with high-frequency components. Here we list two typical tuning methods:

- **Tuning the sub-band weight.** Assigning different weights to subbands provides a simple way to control their distortion contributions. In general, the LL subband receives a higher weight, while HL, LH, and HH are assigned lower, equal weights. Specifically, in our setting, for simplicity, they are learnable parameters.

- **Tuning the $\sigma$-map per subband.** For a more fine-grained optimization, we can compute WD separately for each subband with distinct $\sigma$ ranges: LL uses a narrow range to preserve fidelity, HL and LH share a moderately larger range, and HH is assigned the largest range to better capture high-frequency distortions.

# F. Additional experiments and visualizations

## F.1. Experiments over Kodak, CLIC2020, and tuned parameters

We can observe that while the proposed Re2IC achieves state-of-the-art perceptual performance, the PSNR and MS-SSIM are noticeably lower than other methods like VTM and MLIC+. The observed gap arises from the inherent rate–distortion–perception (RDP) trade-off: at a fixed rate, improving perceptual quality necessarily degrades distortion. This phenomenon has been widely observed in prior perceptual codecs (e.g., C3/WDs, HiFiC). Distortion-oriented codecs, such as VTM and MLIC++, are optimized for pixel-wise fidelity, which leads to strong PSNR performance but can result in over-smoothed textures under limited bitrates. In contrast, Re2IC emphasizes perceptual quality through distribution alignment and region-aware bit allocation, preserving salient structures while allowing texture resampling in less important regions. This yields more realistic visual quality at the cost of reduced pixel-wise fidelity, leading to lower PSNR/MS-SSIM.

## F.2. Additional perceptual-codec baselines

This section presents additional perceptual codecs, with results reported in Tables 6 and 7. Here, we focus solely on perceptual performance and intentionally ignore computational complexity to provide a broader comparison. For fairness, we report results at the closest available bitrate to our target operating points. Our comparison focuses on low-complexity implicit codecs, whose rate–perception–complexity characteristics differ fundamentally from diffusion- or GAN-based methods. Although diffusion models achieve high perceptual scores on metrics like MS-SSIM or LPIPS, Re2IC delivers competitive or even superior FID/DISTS/KID/NIQE/CLIP-IQA/MANIQA performance with far lower complexity, highlighting the effectiveness of implicit codecs for perceptual modeling.

## F.3. Complexity and coding latency

We present the detailed latency and complexity for decoding in Table 8 and encoding in Table 9. We also report encoding latency for CLIC2020 images in Table 10. We can observe that latency increases significantly with resolution, highlighting

*Table 7.* Experiments with more baselines, TACO (Lee et al., 2024) on Kodak, where we select the baseline with the most relevant rate regimes. **Red** and blue indicate the best and second-best results, respectively. HiFiC and TACO are excluded from the high-BPP regime comparisons since their rates are significantly higher or lower. Re$^2$IC consistently delivers **state-of-the-art or leading** performance across FID, DISTS, NIQE, ClipIQA, and MANIQA, even compared with the current strongest model, TACO, at the target medium rate.

| | Model | BPP↓ | PSNR↑ | MS-SSIM↑ | LPIPS↓ | DISTS↓ | FID↓ | NIQE↓ | MUSIQ↑ | ClipIQA↑ | MANIQA↑ |
|---|---|---|---|---|---|---|---|---|---|---|---|
| | VTM | 0.182 | **30.10** | **0.9520** | 0.2538 | 0.1767 | 147.87 | 5.1808 | 70.68 | 0.4827 | 0.4084 |
| | TACO | 0.183 | 28.78 | 0.9399 | **0.0390** | 0.0800 | 41.05 | 4.2099 | 72.7770 | **0.6989** | 0.4028 |
| Medium | C3/WDs | 0.183 | 26.23 | 0.9324 | 0.1150 | 0.0786 | 49.87 | 5.5448 | 68.49 | 0.5062 | 0.3899 |
| | HiFiC | 0.183 | 27.56 | 0.9456 | 0.0665 | 0.0889 | 54.89 | **2.7894** | **73.76** | 0.6562 | **0.4576** |
| | Re$^2$IC | 0.183 | 25.65 | 0.9006 | 0.1017 | **0.0572** | 37.05 | 4.0302 | 71.86 | 0.6149 | 0.4388 |
| | VTM | 0.287 | **31.84** | **0.9694** | 0.1864 | 0.1416 | 106.87 | 4.6125 | 73.13 | 0.5623 | **0.4586** |
| | TACO | 0.226 | 30.08 | 0.9555 | **0.0367** | 0.0746 | 39.65 | 3.9355 | **73.66** | **0.6835** | 0.4213 |
| High | C3/WDs | 0.275 | 27.79 | 0.9529 | 0.0802 | 0.0537 | 35.39 | 4.8479 | 70.15 | 0.5491 | 0.4085 |
| | Re$^2$IC | 0.275 | 26.74 | 0.9231 | 0.0764 | **0.0374** | **26.09** | **3.8981** | 72.27 | 0.6446 | 0.4446 |

the importance of our accelerated convergence. Since our implementation is purely research-oriented and unoptimized, substantial further speed-ups are expected (Blard et al., 2024).

*Table 8.* Decoding cost breakdown of Re$^2$IC on Kodak dataset across different region numbers, using an NVIDIA RTX 3090 GPU and Intel Core i9-10980XE CPU @ 3.00GHz. Note that all results are reported in **CPU computation**. We can see that the complexity is not significantly increased with the increase in region numbers.

| VTM-19.1 | 293.05 ms (CPU) | | | | | |
|---|---|---|---|---|---|---|
| EVC (S/M/L) | 19.89/24.12/31.91 ms (GPU) | | | | | |
| MLIC$^+$+ | 364.06 ms (GPU) | | | | | |
| C3/WDs | 270.55 ms (CPU), with decoding complexity: 2626∼ 2925 MACs/pixel | | | | | |
| Re$^2$IC | Chain coding: 12.86 ms (CPU) | | | | | |
| | Module | ARM | GPM | LPN1 | LPN2 | LPN3 | Total |
| ($N$ = 1 regions) | CPU decoding time (ms) | 204.27(55.3%) | 155.27(42.0%) | 10.12(2.7%) | − | − | 369.66 |
| | Decoding complexity (MACs/pixel) | 1600 | 716 | 33 | − | − | 2349 |
| ($N$ = 2 regions) | CPU decoding time (ms) | 197.91(49.5%) | 181.77(45.5%) | 9.64(2.4%) | 10.45(2.6%) | − | 399.77 |
| | Decoding complexity (MACs/pixel) | 1600 | 959 | 33 | 33 | − | 2625 |
| ($N$ = 3 regions) | CPU decoding time (ms) | 199.70(43.8%) | 227.15(49.8%) | 9.43(2.0%) | 10.08(2.2%) | 9.61(2.1%) | 455.97 |
| | Decoding complexity (MACs/pixel) | 1600 | 1202 | 33 | 33 | 33 | 2901 |

## F.4. More experiments and ablation study

This section provides more quantitative results in Figures 13, 14.

To validate the loss-function design, we first remove the DWT during optimization and observe a substantial degradation in RP performance for both C3 and Re$^2$IC (see Table 12), confirming the effectiveness of our WA-WD formulation.

We then assess the contribution of each mechanism by ablating components one at a time and comparing their performance against the full Re$^2$IC model, with results reported in Table 11. Removing the LPN and region-partition strategy causes the expected similar degradation, discarding LPNs and relying solely on a GPM essentially reduces the model to single-region overfitting. Eliminating the GPM results in a pure dual-overfitting setup similar to C3, yielding an even larger drop in performance, which highlights the importance of local–global perceptual modeling. Together, these ablations validate the effectiveness of our region-based perceptual modeling and demonstrate that each component contributes meaningfully to the overall performance of Re$^2$IC. Interestingly, we find that common randomness (CR) plays a critical role in improving RP performance for Re$^2$IC, consistent with observations in (Ballé et al., 2025). In Re$^2$IC, CR becomes even more influential because it allows more flexible utilization across both spatial and frequency dimensions.

*Table 9.* Encoding cost breakdown of Re$^2$IC on Kodak dataset across different region numbers, using an NVIDIA RTX 3090 GPU and Intel Core i9-10980XE CPU @ 3.00GHz. We can see that the complexity is not significantly increased with the increase in region numbers. Encoding complexity can be approximated as three times the decoding cost. Note that here, the complexity of the feature backbone can vary, as it depends on the backbone and the slice number, where the latency is reported with VGG for both C3/WDs and Re$^2$IC.

| | | | | | | | | |
|---|---|---|---|---|---|---|---|---|
| VTM-19.1 | 87.13 s (CPU) | | | | | | | |
| EVC (S/M/L) | 22.43/36.19/47.87 ms (GPU) | | | | | | | |
| MLIC$^+$+ | 271.91 ms (GPU) | | | | | | | |
| C3/WDs (1k) | 70.24s/1k steps (GPU) | | | | | | | |
| Re$^2$IC (1k) | EML-Net partition time: $\simeq$ 50 ms (GPU), Chain coding: 119.58 ms (CPU) | | | | | | | |
| | **Module** | **ARM** | **GPM** | **LPN1** | **LPN2** | **LPN3** | **Feature backbone** | **Total** |
| ($N=1$ regions) | GPU encoding time (s/1k) | 6.84(6.9%) | 8.11(8.2%) | 1.93(2%) | – | – | 82.07(82.9%) | 98.95 |
| | Encoding complexity (kMACs/pixel) | 4800 | 2148 | 99 | – | – | – | – |
| ($N=2$ regions) | GPU encoding time (s/1k) | 7.45(7.2%) | 12.14(11.7%) | 1.33(1.3%) | 1.45(1.4%) | – | 80.99(78.4%) | 103.36 |
| | Encoding complexity (kMACs/pixel) | 4800 | 2877 | 99 | 99 | – | – | – |
| ($N=3$ regions) | GPU encoding time (s/1k) | 7.09(6.6%) | 15.10(14.1%) | 0.81(0.8%) | 1.12(1.0%) | 1.06(1.0%) | 82.11(76.5%) | 107.29 |
| | Encoding complexity (kMACs/pixel) | 4800 | 3606 | 99 | 99 | 99 | - | – |

*Table 10.* Encoding cost on CLIC2020 using NVIDIA-3090 GPU and Intel i9 CPU.

| | |
|---|---|
| VTM-19.1 | 394.59 s (CPU) |
| EVC (S/M/L) | 59.70/70.78/128.49 ms (GPU) |
| MLIC++ | 495.19 ms (GPU) |
| C3/WDs (1k) | 257.59 s/1k steps (GPU) |
| Re2IC (1k) | 361.74 s/1k steps (GPU) |

*Table 11.* Ablation study of different mechanisms for the RP performance. Higher DISTS, FID, and LPIPS indicate worse RP performance (shown in darker blue).

| Model Variant | DISTS ↓ vs. FID ↓ vs. LPIPS ↓ | | |
|---|---|---|---|
| | bpp≈0.113 | bpp≈0.183 | bpp≈0.275 |
| Proposed Re$^2$IC scheme | 0.086/58.21/0.1432 | 0.0572/37.05/0.1017 | 0.037/26.09/0.0764 |
| ⇒ w/o wavelet transform | 0.104/67.10/0.1547 | 0.066/41.52/0.1044 | 0.042/28.06/0.0765 |
| ⇒ w/o region partition | 0.117/76.58/0.1732 | 0.076/49.37/0.1167 | 0.053/33.45/0.0817 |
| ⇒ w/o LPN | 0.115/74.93/0.1690 | 0.077/49.39/0.1136 | 0.053/34.07/0.0796 |
| ⇒ w/o GPM | 0.131/86.77/0.1941 | 0.087/53.62/0.1303 | 0.057/34.17/0.0952 |
| ⇒ w/o common randomness | 0.151/124.7/0.2613 | 0.110/75.55/0.1852 | 0.081/48.92/0.1285 |

*Table 12.* Ablation study for the role of WA-WD as a loss function across different implicit codecs, where the red background indicates better performance.

| Model Variant | DISTS ↓ vs. FID ↓ vs. LPIPS ↓ | | |
|---|---|---|---|
| | bpp≈0.113 | bpp≈0.183 | bpp≈0.275 |
| Re$^2$IC / WDs | 0.104/67.10/0.1547 | 0.066/41.52/0.1044 | 0.042/28.06/0.0765 |
| ⇒ WA-WD | 0.086/58.21/0.1432 | 0.057/37.05/0.1017 | 0.037/26.09/0.0764 |
| C3 / WDs | 0.118/72.83/0.1717 | 0.079/49.87/0.1150 | 0.054/35.39/0.0802 |
| ⇒ WA-WD | 0.090/62.30/0.1514 | 0.059/40.14/0.1047 | 0.041/28.51/0.0783 |

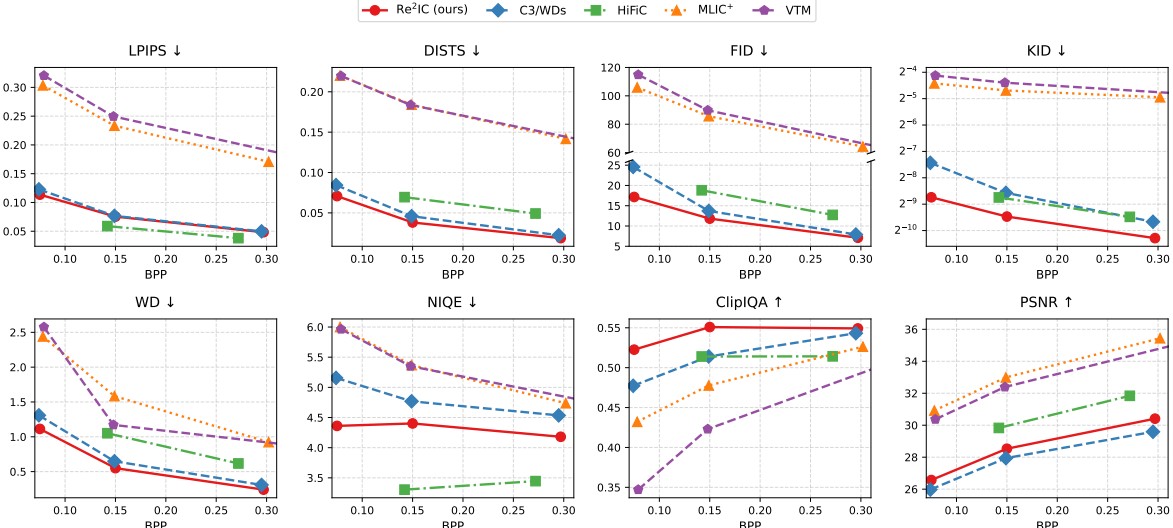

*Figure 13.* Rate-distortion and -perception curves on CLIC2020. Arrows in the title indicate whether lower is better (↓), or higher is better (↑).

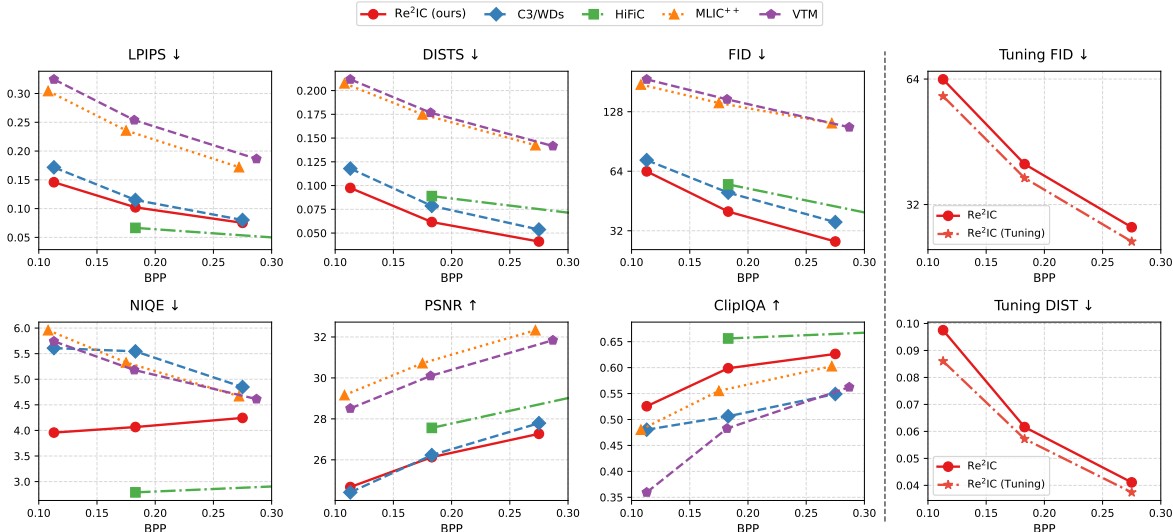

*Figure 14.* Rate-distortion and -perception curves on Kodak. Arrows in the title indicate whether lower is better (↓), or higher is better (↑).

### F.5. Technical comparison for DWT

**Comparison with other DWT-based codecs.** The use of wavelet transforms for frequency-aware optimization has been explored in prior works (Fu et al., 2024; Yin et al., 2025; Ma et al., 2019). However, their roles, objectives, and technical pathways for $Re^2IC$ and these DWT-based codecs are fundamentally different. Prior wavelet-based codecs use the DWT as an analysis transform to decorrelate latent features, produce sparser subbands, and improve entropy coding efficiency. In contrast, Re2IC is an implicit codec. It overfits each individual image and directly models human perception on a per-sample basis. Thus, DWT in Re2IC is not used to improve compression of latent codes, but to improve the perceptual optimization metric itself. In Re2IC, DWT is introduced to: (1). improve the statistical validity of WD approximation. DWT reduces cross-correlations and yields a tighter diagonal-Gaussian approximation, making WD approximation more stable and accurate. (2). Provide a tunable, frequency-aware perceptual metric. WA-WD enables a tunable perception metric and allows flexible balancing of fidelity and realism by adjusting subband weights or $\sigma$-maps. (3). Enable joint spatial–frequency perception modeling. Combined with our region-wise overfitting scheme, WA-WD explicitly models perception along both spatial saliency and frequency structure, which is essential for realism-enhanced implicit codecs.

**Effect of joint optimized entropy models.** Our method adopts a joint entropy model across regions, coupled with a local–global perception modulation mechanism. This section analyzes how different entropy–modeling strategies affect performance. For fairness, we consider 2-region scenarios (with more regions, our benefits are expected to be larger due to the significantly increased bpp costs for entropy models), and match the overall parameter budget across settings (joint entropy: two models with dimensions $16 \times 16$, $16 \times 16$, $16 \times 2$; separate entropy: $24 \times 24$, $24 \times 24$, $24 \times 2$). As shown in Table 13, the joint entropy model achieves better performance, validating our design.

*Table 13.* Ablation study for different settings of entropy models for Kodak images.

| Model Variant | PSNR (dB) ↑ / bpp ↓ | | | |
|---|---|---|---|---|
| Re$^2$IC / joint entropy model | 29.04/0.156 | 32.57/0.361 | 34.59/0.531 | 39.30/1.11 |
| ⇒ separate entropy models | 29.04/0.158 | 32.54/0.364 | 34.58/0.535 | 39.24/1.15 |

*Table 14.* Ablation study for RD performance across different methods on Kodak dataset.

| Model | PSNR (dB) ↑ / bpp ↓ | | | | | | BD-rate over VTM-19.1 % ↓ |
|---|---|---|---|---|---|---|---|
| WeConvene | 30.07/0.163 | 30.88/0.203 | 32.53/0.309 | 34.29/0.452 | 36.04/0.637 | 37.96/0.897 | −8.37 |
| Re$^2$IC / MSE | 29.12/0.151 | 32.61/0.356 | 34.60/0.525 | 36.65/0.743 | 39.33/1.107 | 41.26/1.448 | −1.45 |

**Visualization of Feature Correlations.** To validate that DWT improves the independent-Gaussian assumption underlying WA-WD, we visualize feature correlations before and after wavelet decomposition. As shown in Fig. 15, increasing the number of DWT levels progressively reduces inter-band correlations, producing more independent coefficients and better supporting the diagonal-Gaussian approximation used in WA-WD.

**Frequency-domain optimization objectives.** First, we compare RD performance across two DWT-based methods, and the results are presented in Table 14. We observe that although WeConvene achieves stronger RD performance, it does so at the cost of significantly higher decoding complexity. In contrast, Re$^2$IC delivers competitive RD results, achieving a 6.92% BD-rate reduction relative to VTM-19.1, while maintaining substantially lower complexity. Then, we analyze the RP performance across two DWT methods. As shown in Figs. 16 and 17, Re$^2$IC significantly improves the perception performance under lower decoding complexity and lower bpp cost. Last, we conduct an ablation study over each sub-band for the WA-WD optimization. Figs. 18 and 19 show tunable weights can enhance the RP performance, Table 1 quantifies the performance gain from tuning sub-band weights.

**Wavelet decomposition levels.** Using more wavelet levels can, in principle, yield more independent features and more accurate WA-WD estimation (as illustrated in Fig. 15). However, in practice, higher levels cause VGG-downsampled features to become too small for stable WD estimation (five VGG stages combined with 2–3 DWT levels reduce resolution by factors of 128–256 ×), which also introduces higher computational cost. Although alternative backbones or scale settings could mitigate this, changing the backbone would break fairness with baselines such as C3/WDs. Since a single DWT level already provides strong performance with low overhead, we adopt 1-level DWT in this work and leave multi-level extensions for future study.

**Effect of each sub-band** Figs. 18 and 19 present ablations over different subband configurations. We compare using equal subband weights versus learnable tuned weights, and observe that tuned weights yield better RP performance, confirming the benefit of subband-aware optimization. Removing the LL subband leads to severe degradation: fundamental structures such as lighting, shading, and global geometry collapse, indicating that LL is essential for maintaining fidelity. Removing only LH, HL, or HH subbands can produce acceptable reconstructions but noticeably reduces high-frequency details, consistent with prior findings in (Deng et al., 2019), confirming that high-frequency subbands primarily govern fine textures.

### F.6. Latent bit cost and visualization

We further visualize the latent vectors in Fig. 20 and their bit allocation in Fig. 5.

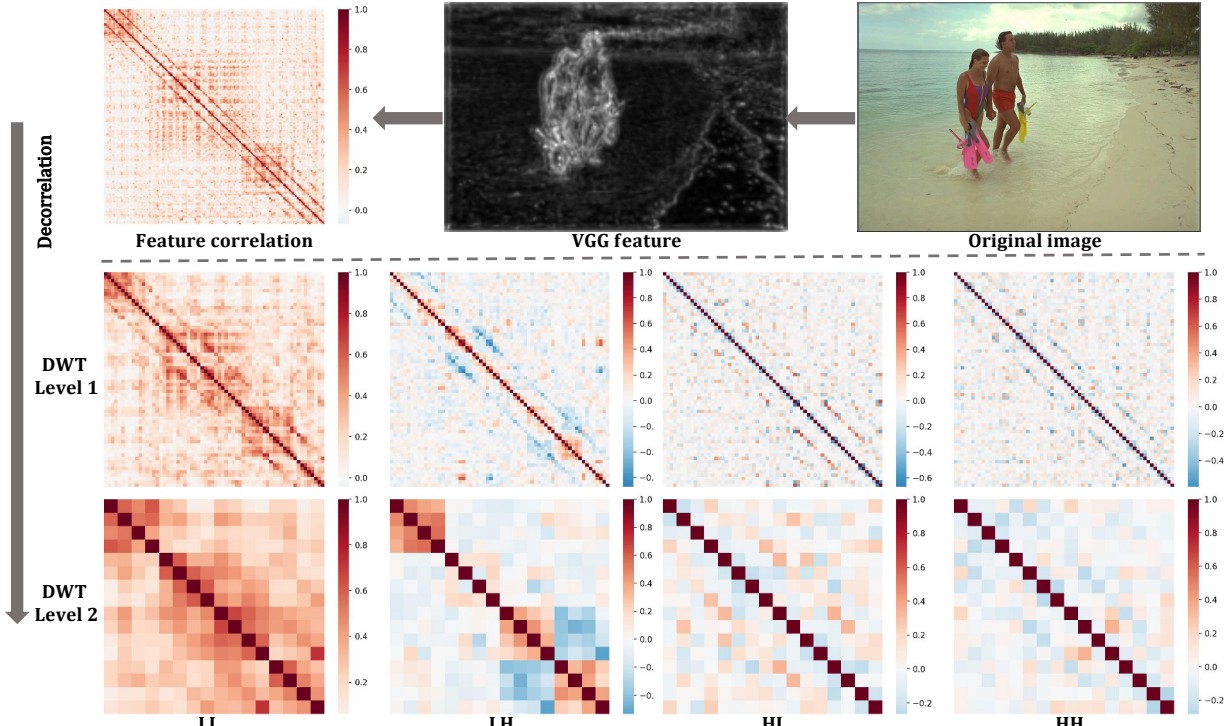

*Figure 15.* Visualization of feature correlations across wavelet subbands. Increasing DWT levels progressively reduces inter-band correlations, yielding more independent coefficients and better supporting the diagonal-Gaussian approximation used in WA-WD.

### F.7. Ablation over region number

Regarding the sensitivity to the region number and saliency quality, we conduct additional experiments. As shown in Table 15, the results exhibit limited sensitivity: 2–3 regions already achieve satisfactory performance, while increasing to 5 regions leads to a slight drop. We further observe that most images contain no more than 3 meaningful salient regions. To assess robustness to saliency quality, we consider two settings: (1) 5-region cases of Table 15, where increasing the number of regions reduces the distinctiveness of saliency, leading to regions that are not strictly salient; and (2) Lossy contour (see experiments in Section 4.4 and Fig. 21), which introduces additional regions that do not strictly correspond to salient structures. In both cases, we observe only minor performance degradation. These results suggest that a simple 2–3 region partition is sufficient for Re2IC in practice, and that the method is robust to imperfect saliency.

### F.8. Artifacts from contour compression

Artifact effects of contour compression is provided in Fig. 21

### F.9. Fast convergence

RD and RP convergence is ploted in Figs. 7 and 22.

### F.10. Kodak dataset

For the Kodak dataset, more visualizations are provided as Figs. 23 25 26 27 28.

### F.11. CLIC2020 dataset

For the CLIC2020 dataset, more visualizations are provided as Figs. 29, 30, 31, 32. Since detailed rates of some CLIC2020 baselines are not released, we report results using images from the same rate regime for visual comparisons.

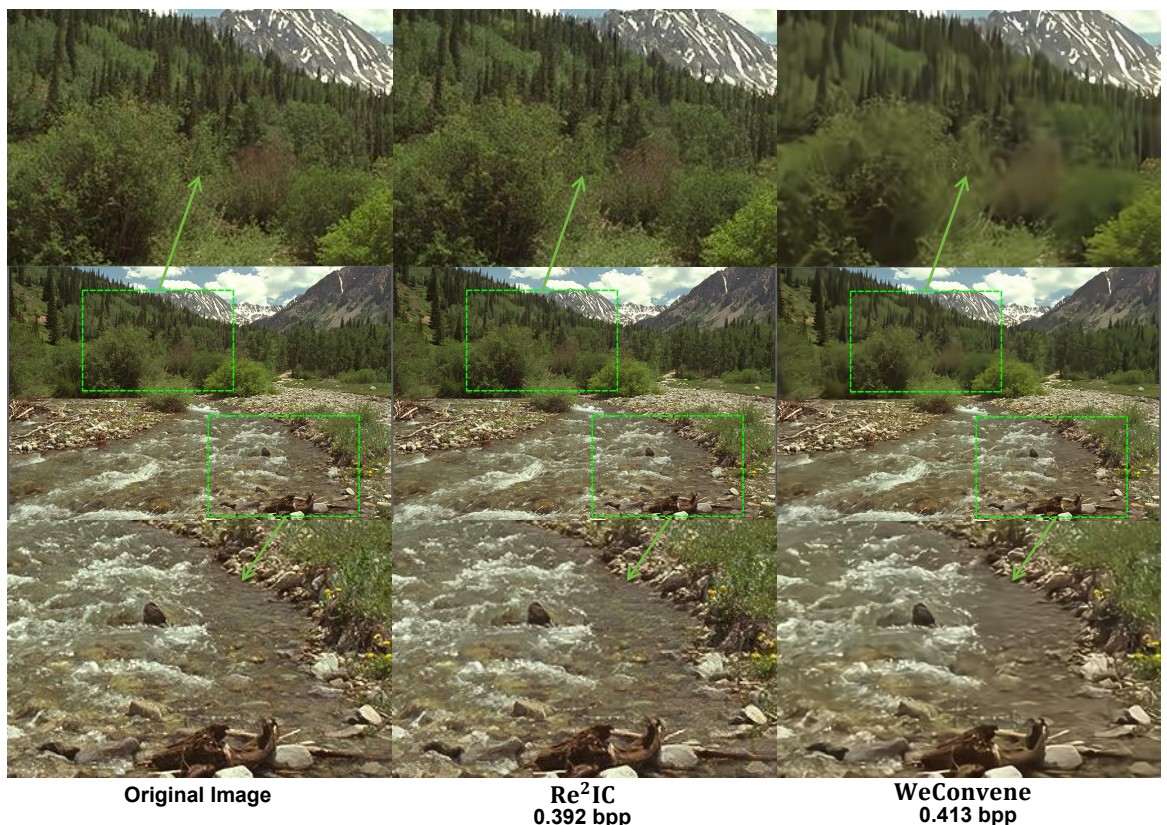

Original Image                    Re²IC                      WeConvene
                               0.392 bpp                    0.413 bpp

*Figure 16.* Visualizations for different DWT-based codecs.

*Table 15.* Quantitative comparison across different numbers of regions (1, 3, 5, 8, 21) on Kodak. For the 5-region setting, the additional region is not strictly defined by saliency, and is included to evaluate robustness to saliency quality.

| | Model | BPP↓ | PSNR↑ | MS-SSIM↑ | LPIPS↓ | DISTS↓ | FID↓ | NIQE↓ | CLIPIQA↑ |
|---|---|---|---|---|---|---|---|---|---|
| Low | Two regions | 0.135 | 22.8264 | 0.8831 | 0.1489 | 0.1018 | 65.4325 | 4.2945 | 0.5170 |
| | Three regions | 0.136 | 22.7360 | 0.8843 | 0.1436 | 0.1012 | 66.0363 | 4.2786 | 0.5197 |
| | Five regions | 0.137 | 22.8128 | 0.8841 | 0.1462 | 0.0982 | 66.7724 | 4.1707 | 0.5156 |
| Medium | Two regions | 0.221 | 24.1870 | 0.9209 | 0.1000 | 0.0615 | 41.6331 | 3.9186 | 0.5657 |
| | Three regions | 0.217 | 24.3468 | 0.9216 | 0.09652 | 0.0638 | 39.3924 | 4.0387 | 0.5759 |
| | Five regions | 0.221 | 24.3107 | 0.9221 | 0.0973 | 0.0587 | 43.1441 | 3.9202 | 0.5809 |
| High | Two regions | 0.323 | 25.6067 | 0.9443 | 0.0702 | 0.0406 | 26.7772 | 4.2079 | 0.5688 |
| | Three regions | 0.326 | 25.6039 | 0.9449 | 0.07043 | 0.0419 | 26.6076 | 4.0804 | 0.5840 |
| | Five regions | 0.329 | 25.4517 | 0.9440 | 0.0719 | 0.03888 | 29.7044 | 4.0319 | 0.5978 |

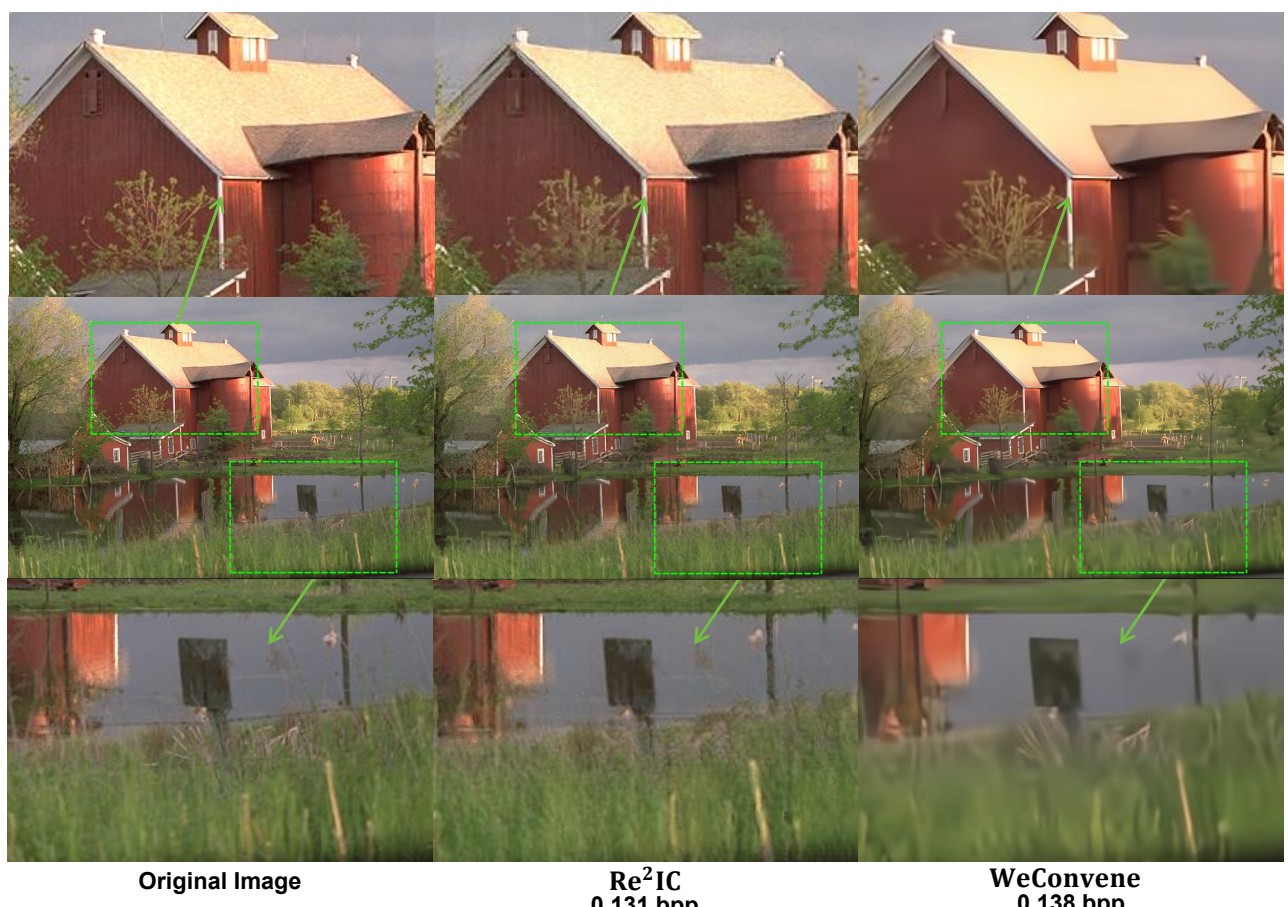

| Original Image | Re$^2$IC | WeConvene |
|:---:|:---:|:---:|
| | 0.131 bpp | 0.138 bpp |

*Figure 17.* Visualizations for different DWT-based codecs.

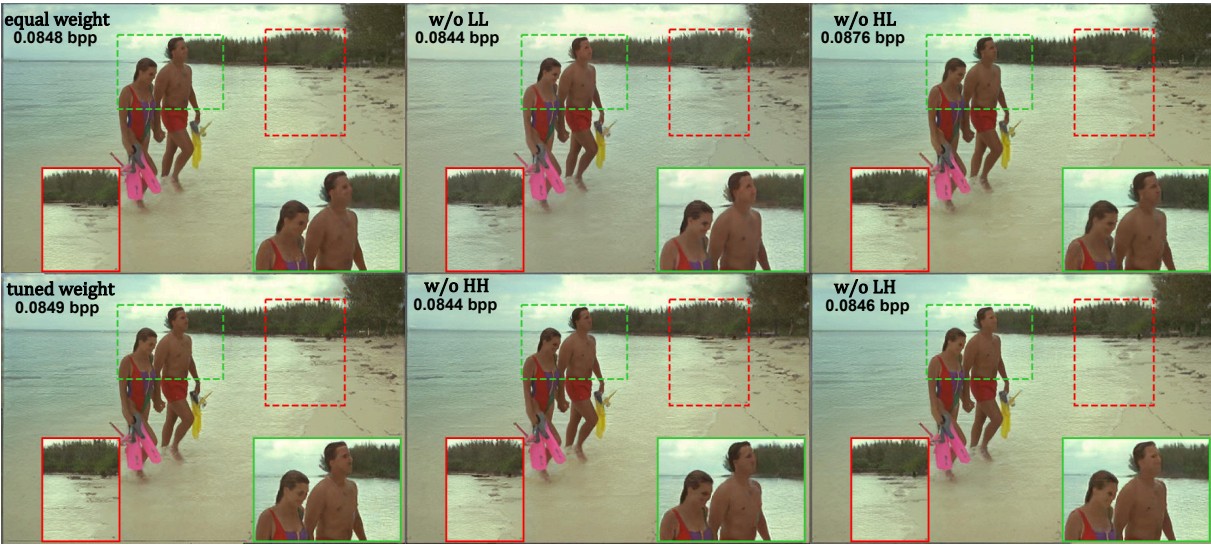

*Figure 18.* Ablation study on WA-WD subbands for kodim 12, where we can observe that learned subband weights outperform equal weighting, yielding higher RP performance. Excluding the LL band causes major degradation in global structure and illumination, whereas excluding LH/HL/HH preserves overall fidelity but reduces high-frequency textures.

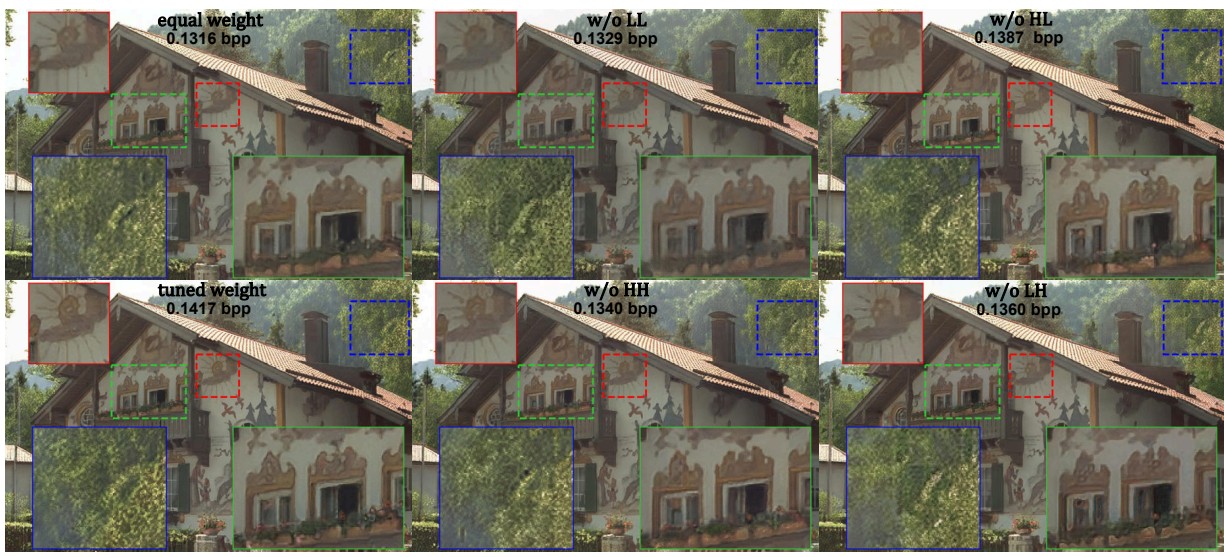

*Figure 19.* Ablation study on WA-WD subbands for kodim 24, where we can observe that learned subband weights outperform equal weighting, yielding higher RP performance. Excluding the LL band causes major degradation in global structure and illumination, whereas excluding LH/HL/HH preserves overall fidelity but reduces high-frequency textures.

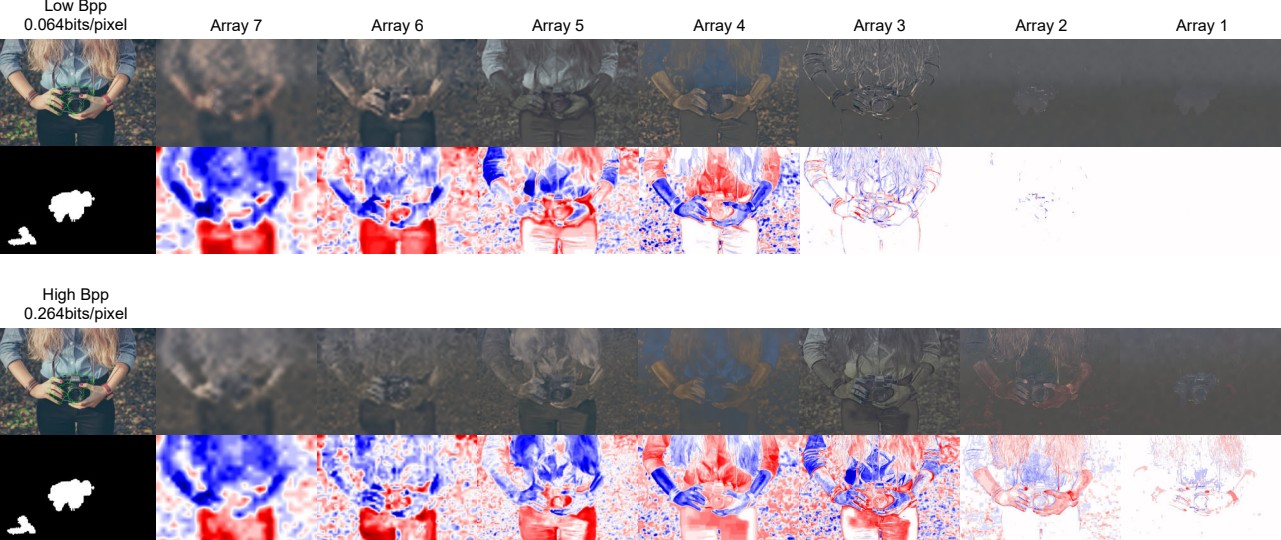

*Figure 20.* Visualization of the latent vectors (two salient regions as an example), where latent vectors are arranged from low to high resolution. The first row shows reconstructions where all but the array are set to zero. The second row displays the raw arrays, upscaled to match the output resolution. At low bit rates, $Re^2IC$ captures rich low-frequency features into low-resolution arrays, then progressively adds high-frequency details with resolution, yielding a more balanced bit allocation across arrays and stronger cross-frequency interaction than prior overfitted codecs, which typically ignore the effect of the lowest resolution arrays/latent vectors, as visualized in (Kim et al., 2024; Wu et al., 2025).

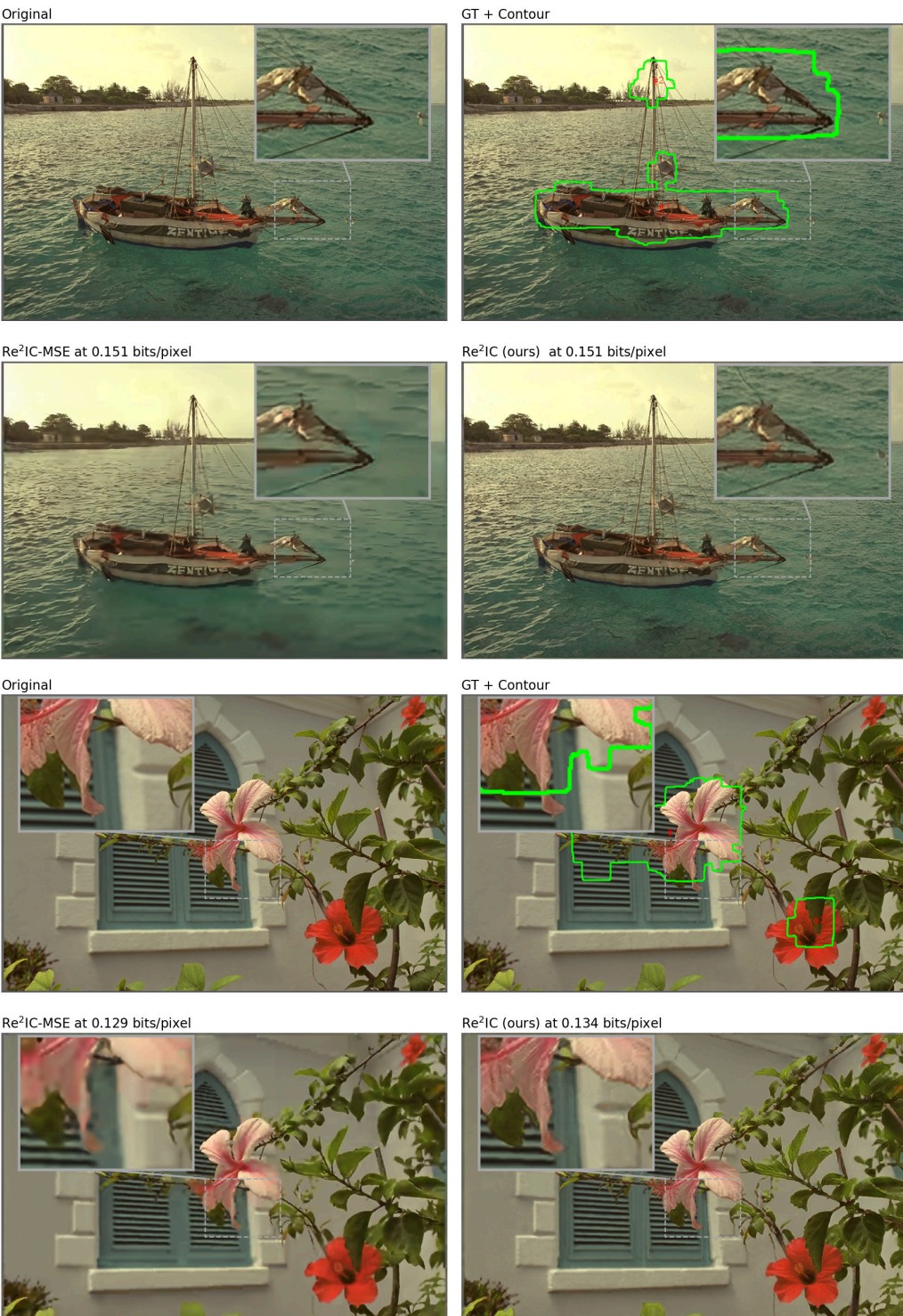

*Figure 21.* Effect of inaccurate contours (green indicates the region for local-fitting): with MSE optimization, artifacts appear around decoded contours where different LPNs meet. In contrast, Re$^2$IC, aided by perceptual modeling and GPM, eliminates these artifacts and achieves perceptual realism closer to the original.

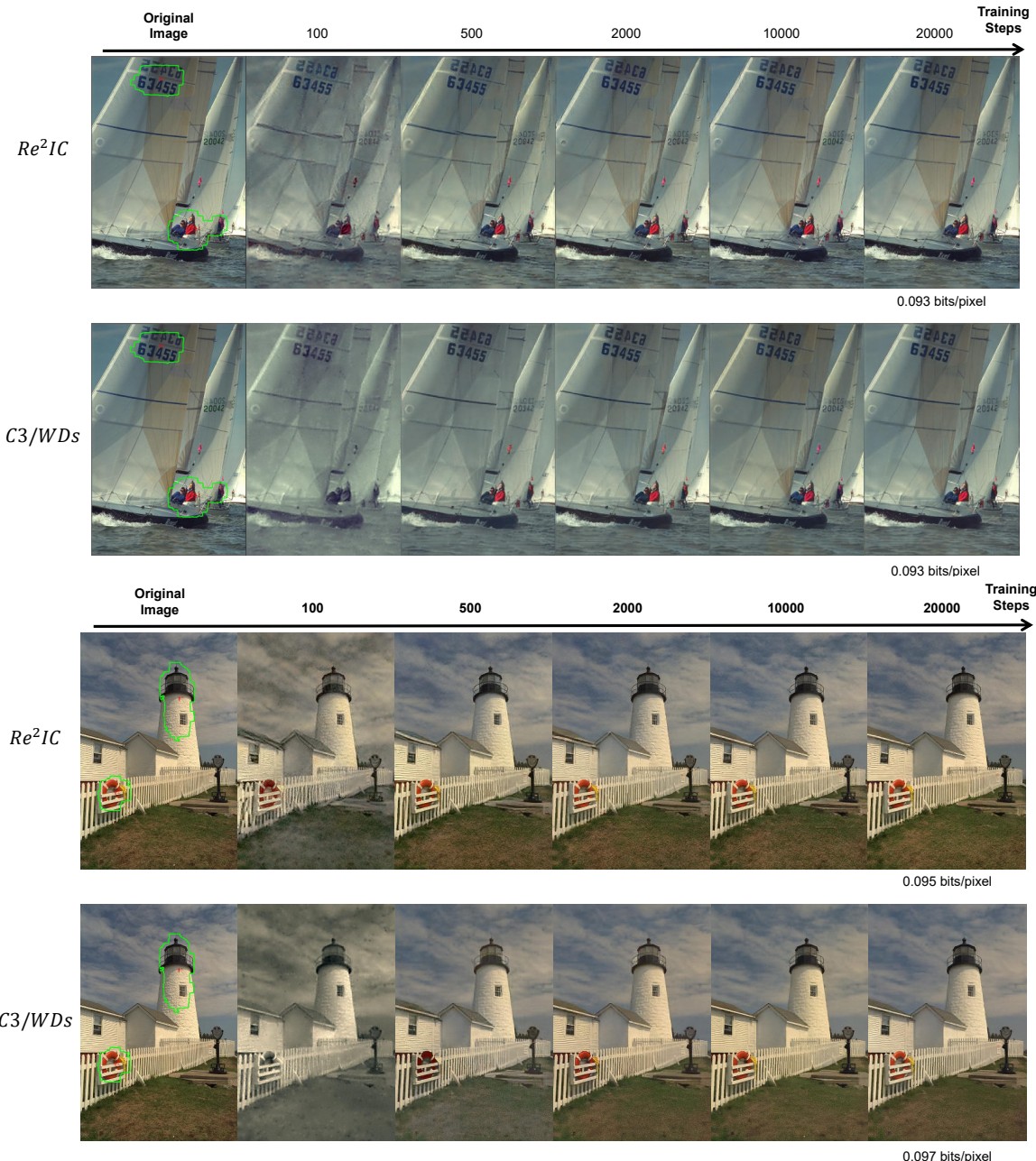

*Figure 22.* RP convergence comparisons (green indicates the partitioned regions): Re²IC converges faster by first fitting salient regions and then refining others.

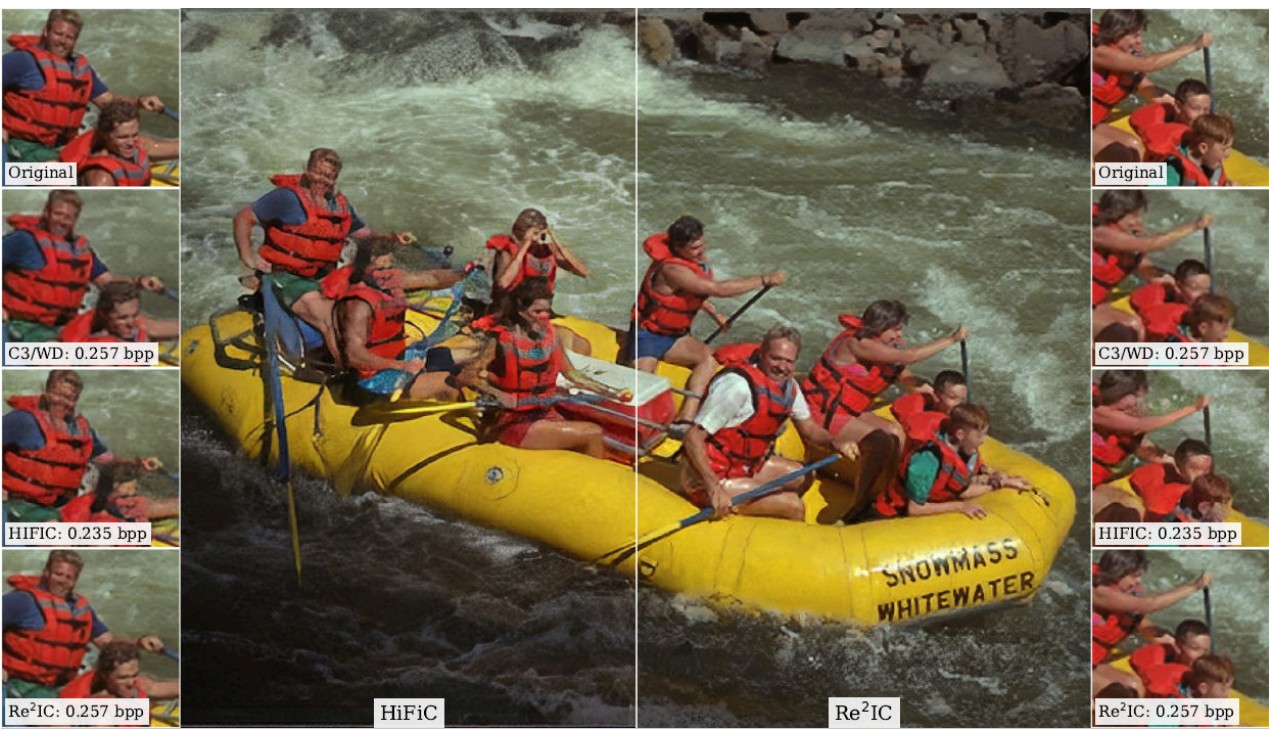

*Figure 23.* More visual comparison between Re$^2$IC, HiFiC, and C3/WDs, where Re$^2$IC can deliver the best perceptual realism, closely matching the original image.

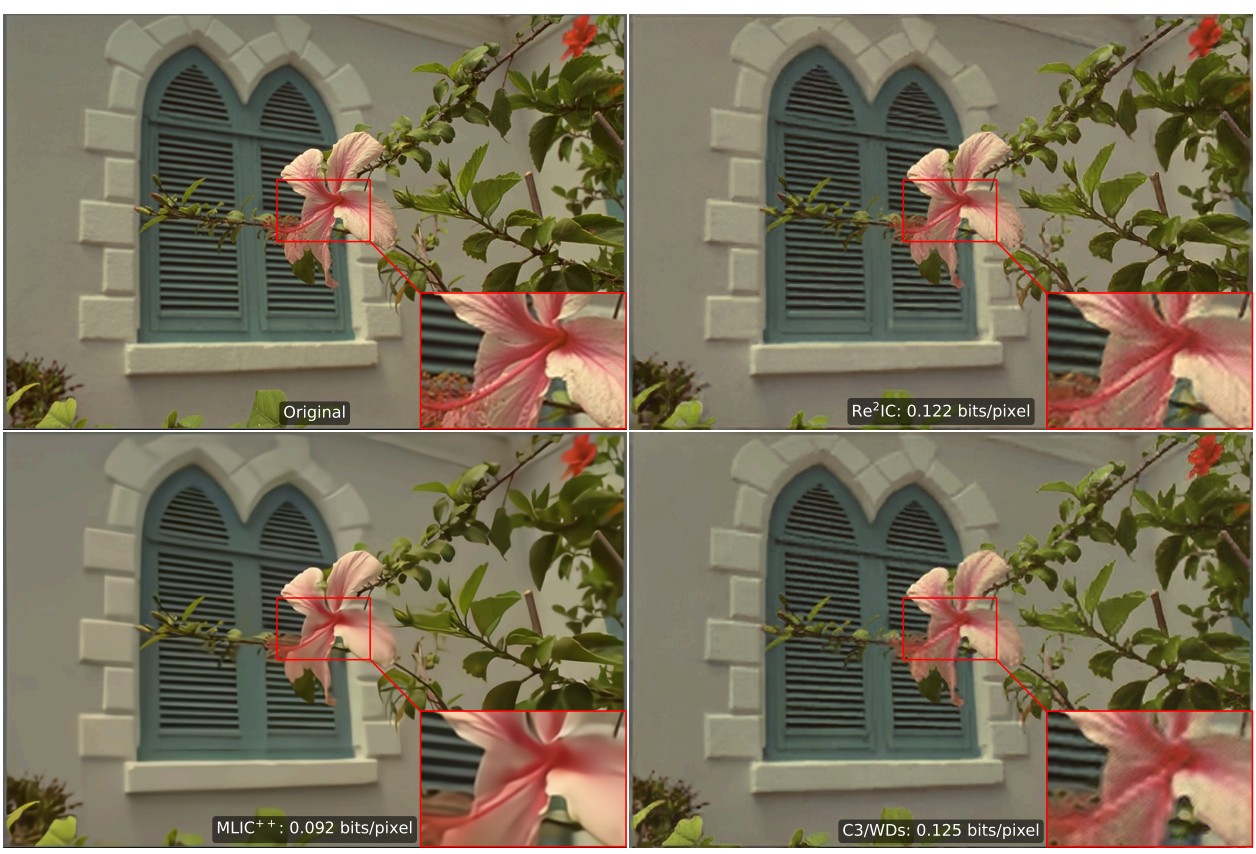

*Figure 24.* More visual comparison between Re$^2$IC, HiFiC, and C3/WDs, where Re$^2$IC can deliver the best perceptual realism, closely matching the original image.

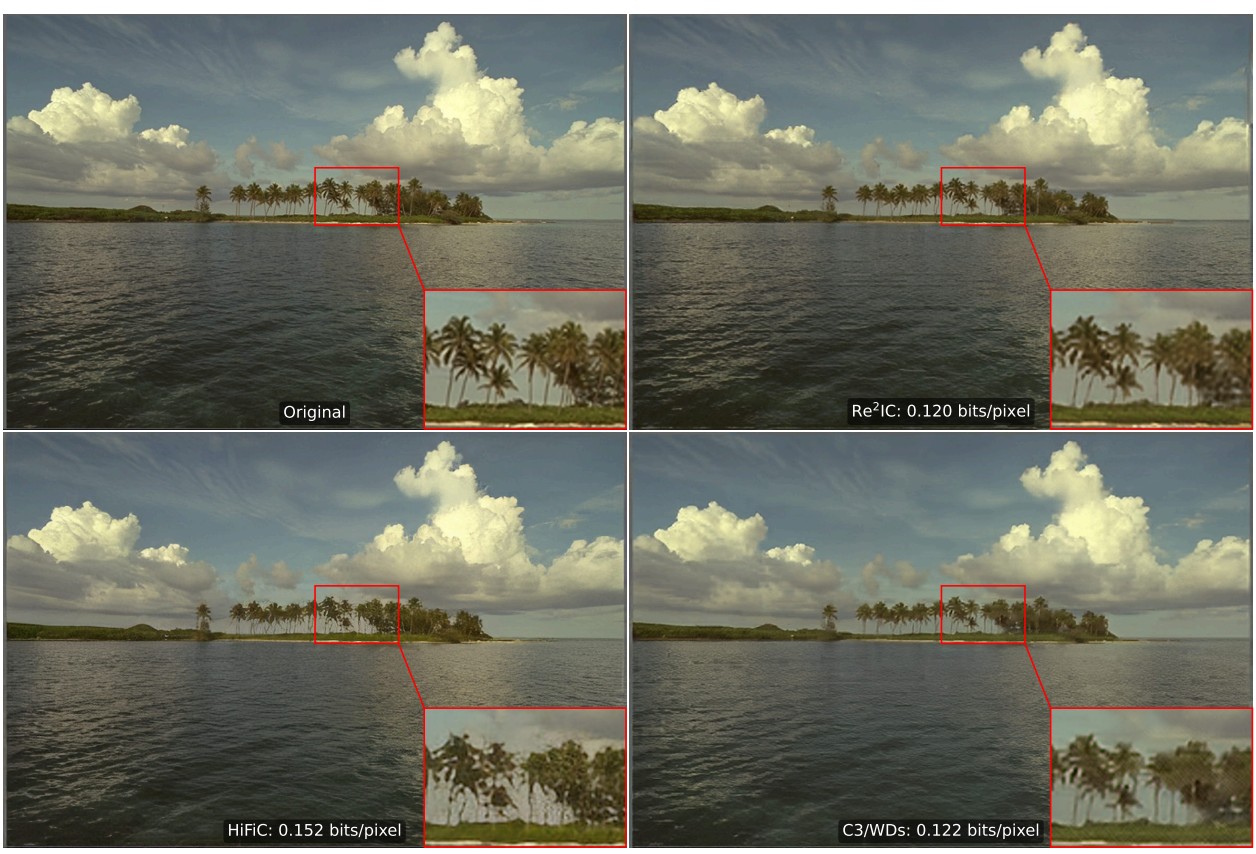

*Figure 25.* More visual comparison between Re$^2$IC, HiFiC, and C3/WDs, where Re$^2$IC can deliver the best perceptual realism, closely matching the original image.

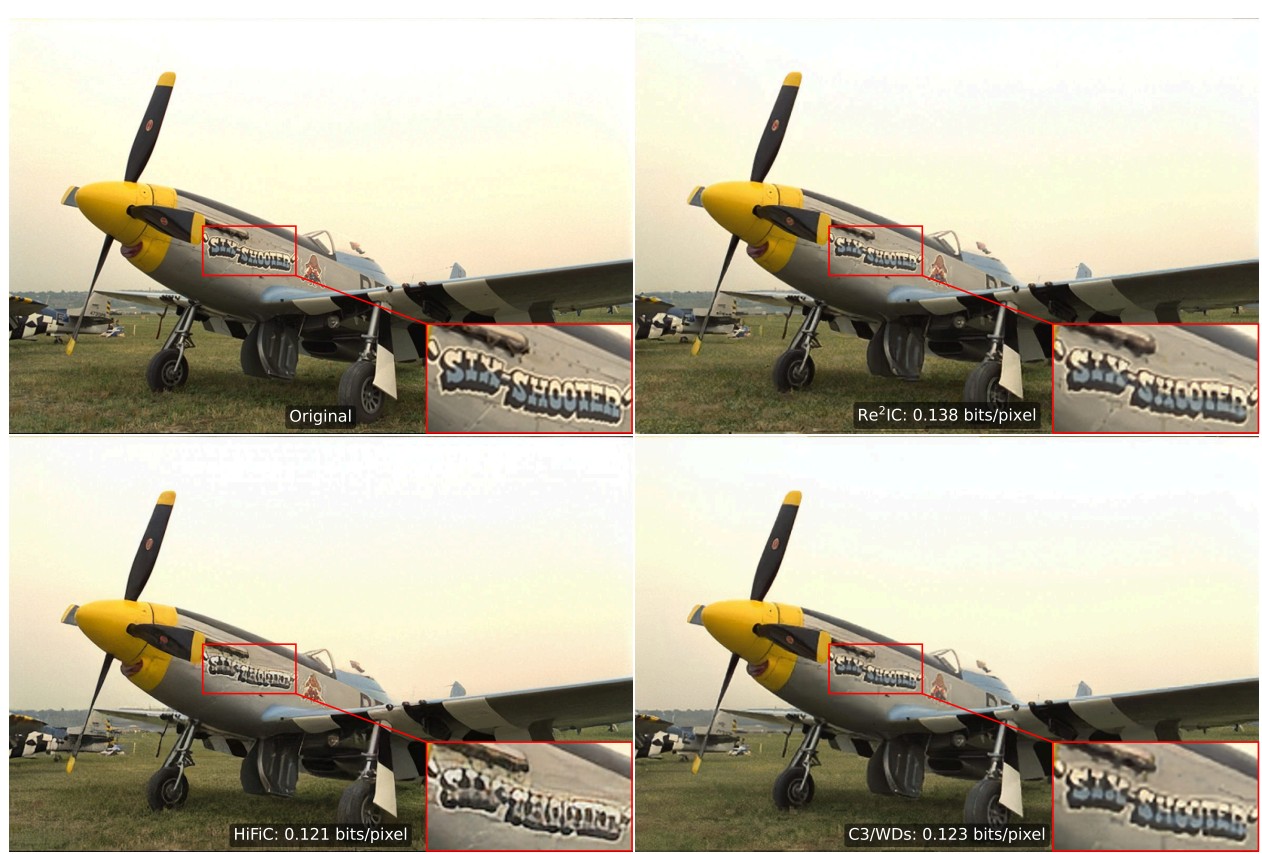

*Figure 26.* More visual comparison between Re$^2$IC, HiFiC, and C3/WDs, where Re$^2$IC can deliver the best perceptual realism, closely matching the original image.

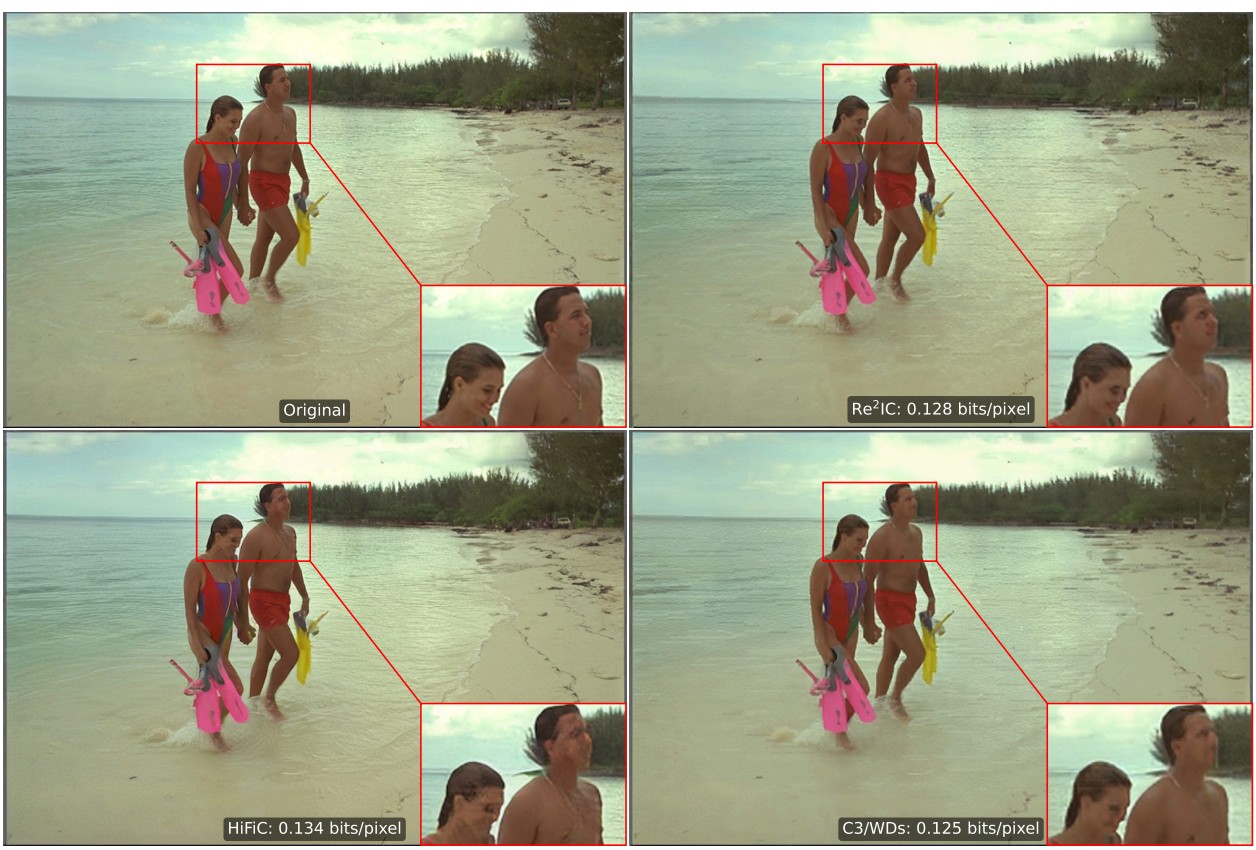

*Figure 27.* More visual comparison between Re²IC, HiFiC, and C3/WDs, where Re²IC can deliver the best perceptual realism, closely matching the original image.

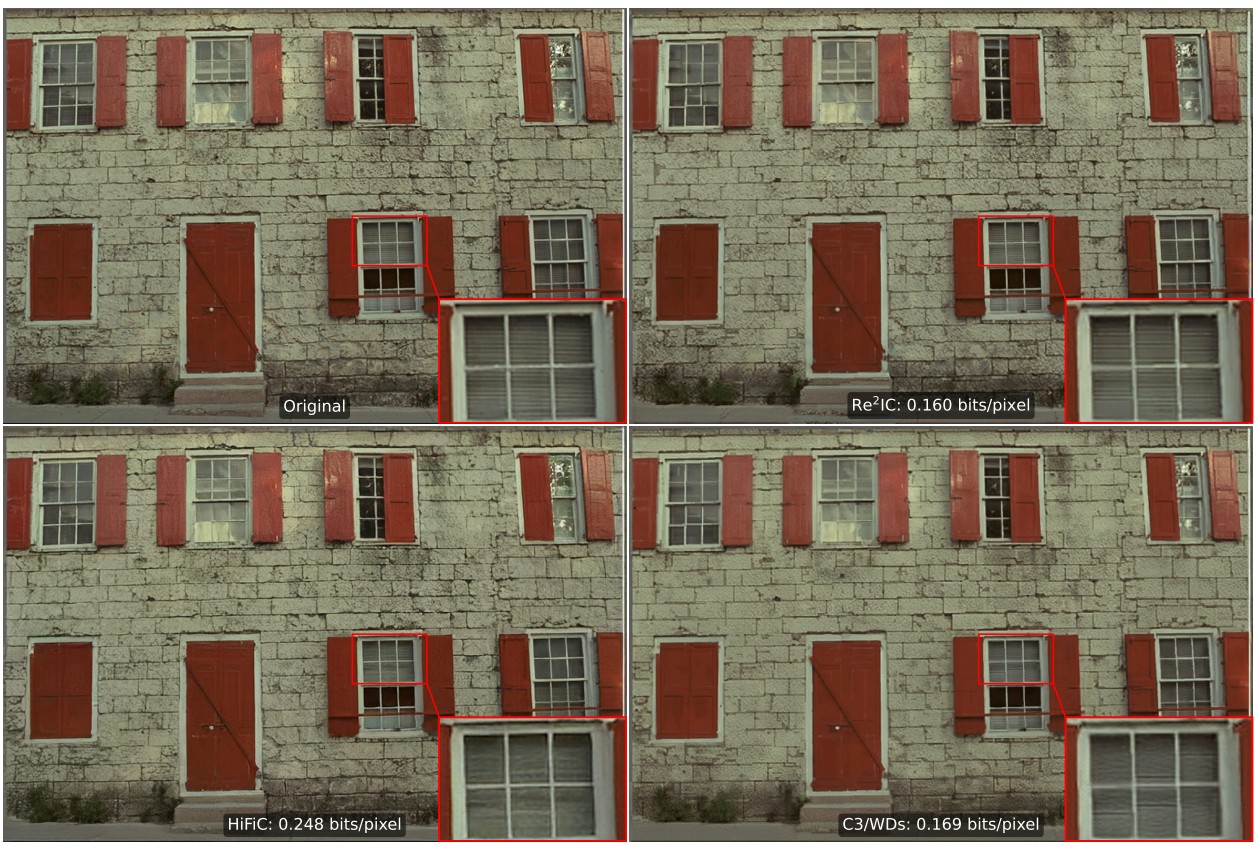

*Figure 28.* More visual comparison between Re$^2$IC, HiFiC, and C3/WDs, where Re$^2$IC can deliver the best perceptual realism, closely matching the original image.

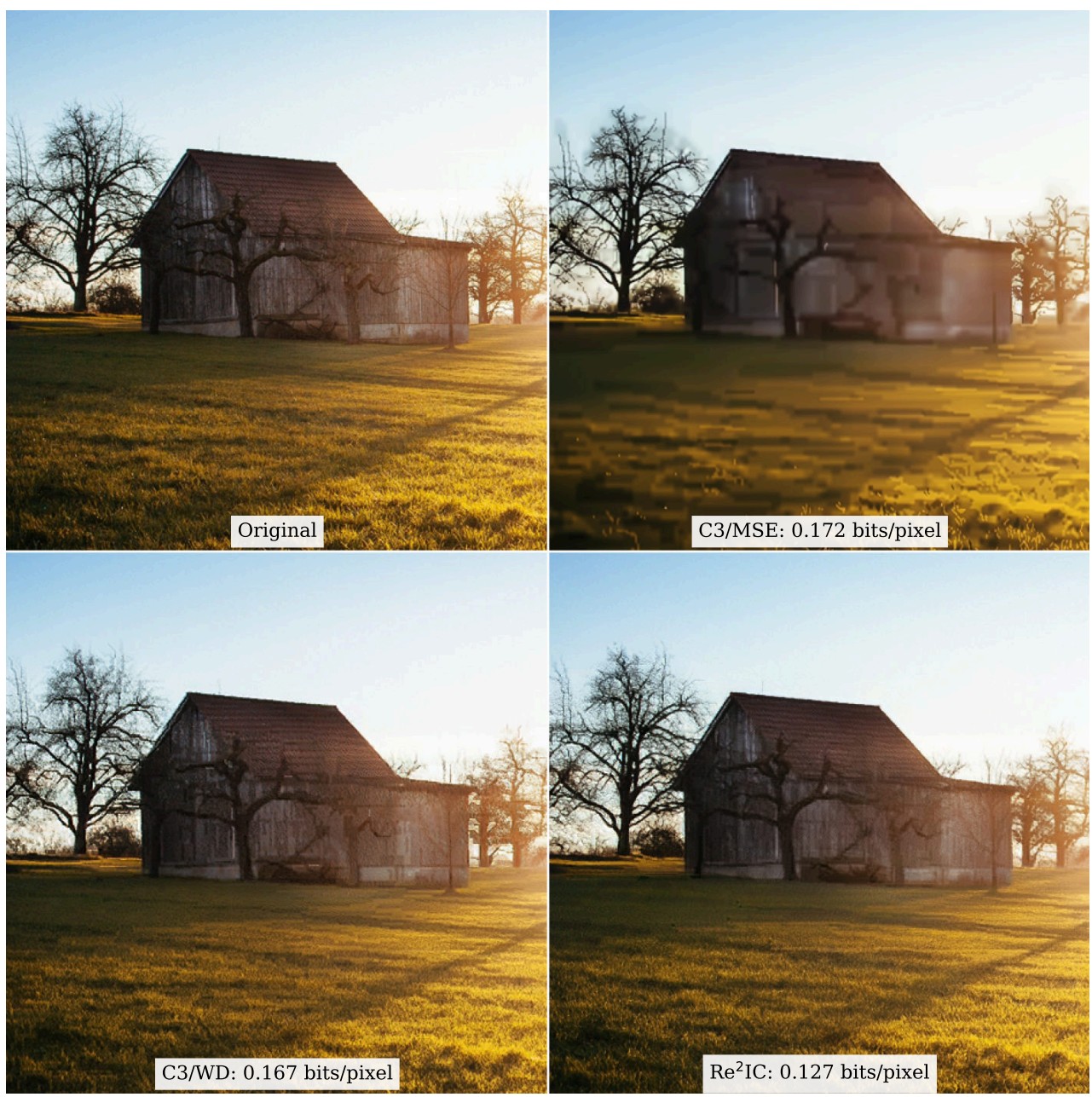

*Figure 29.* Visual comparisons of image 28 from CLIC2020 dataset (best viewed on screen). The MSE-optimized overfitted codec produces flattened textures due to fidelity-focused optimization, while C3/WD can introduce artifacts around trees and wood patterns. In contrast, Re$^2$IC delivers reconstructions with fewer artifacts at a lower bitrate (0.127 bpp), with reduced artifacts.

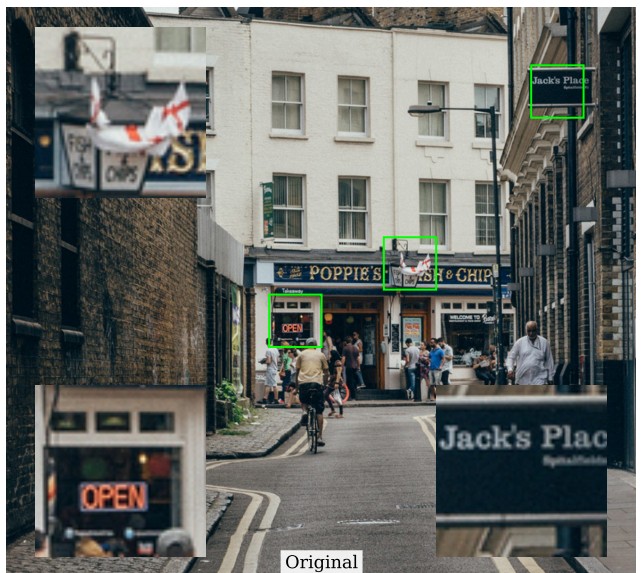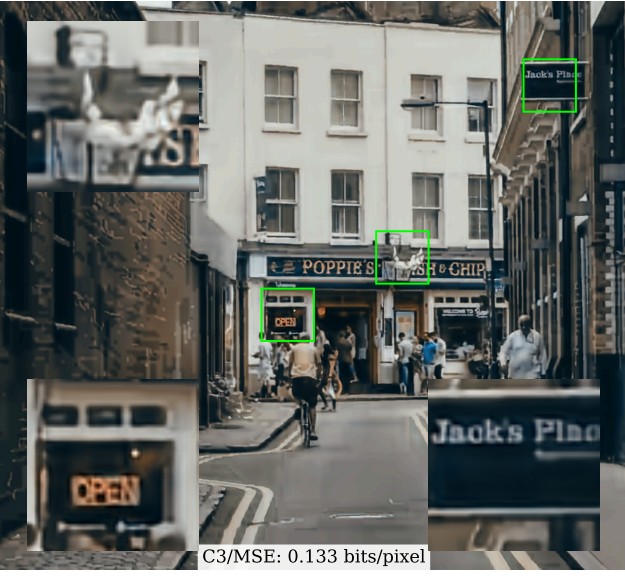

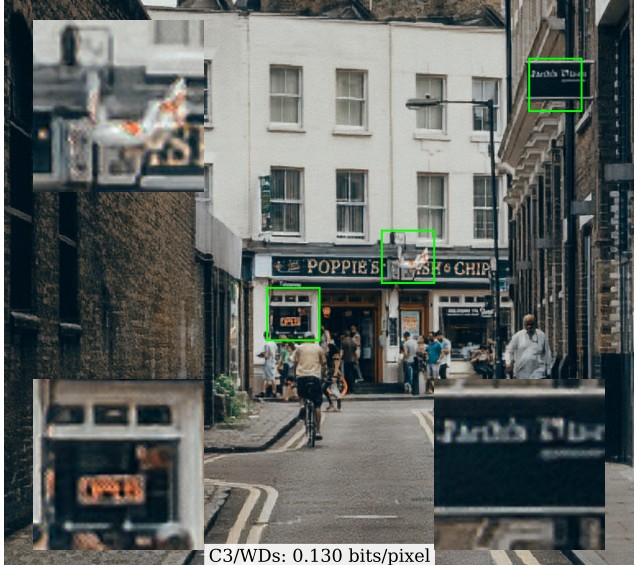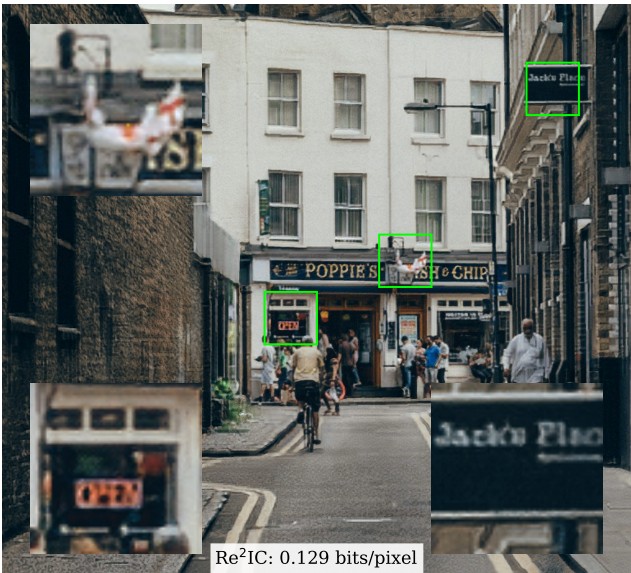

*Figure 30.* Visual comparisons of image 11 from CLIC2020 dataset (best viewed on screen). Similar trends are observed: MSE-optimized codec tends to smooth textures (e.g., roads, walls), while C3/WDs introduces artifacts in non-salient areas (e.g., flags) and loses fidelity in high-frequency content (e.g., text), often reducing realism. Our Re$^2$IC balances both sides.

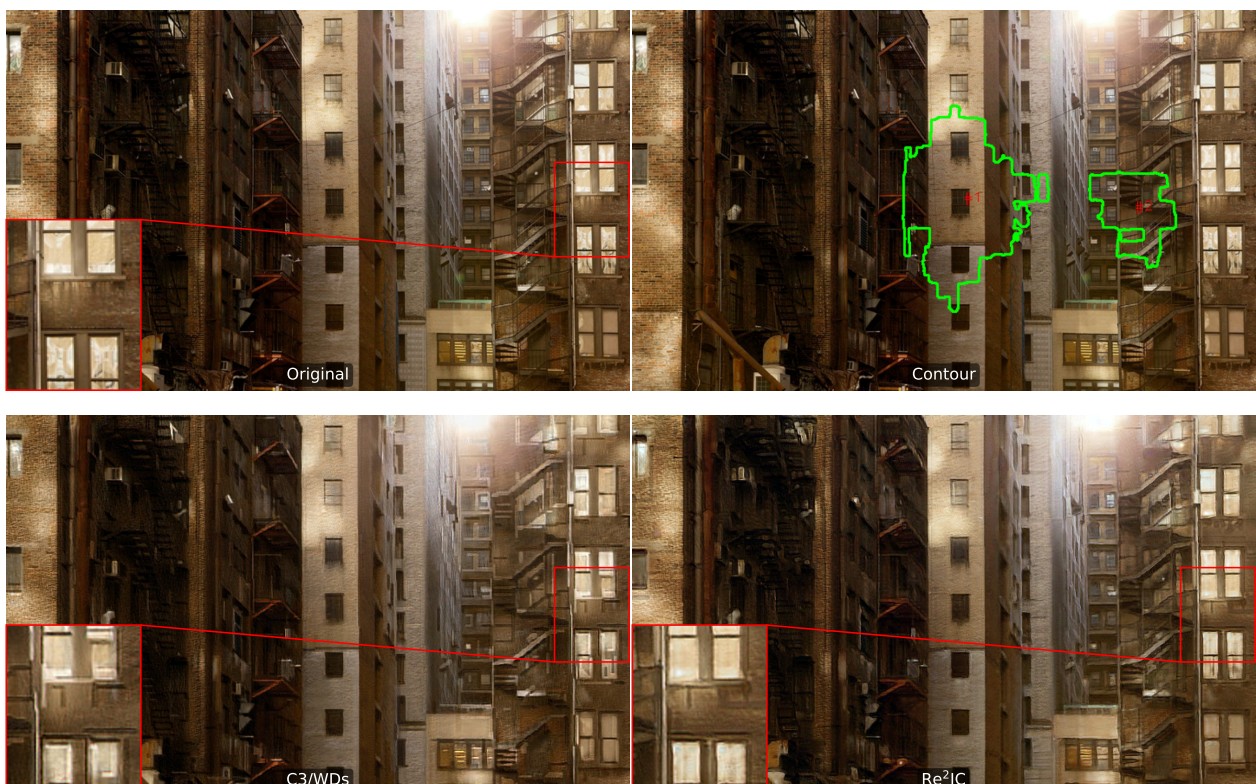

*Figure 31.* Visual compassion for methods at low-bpp regime ($<= 0.075$). We can see that C3/WDs tends to resample the low-frequency information in non-salient areas. This is because it assigns a larger $\sigma$ to non-salient areas without further considering the internal frequency characteristics. As a result, it loses some structural fidelity in non-salient areas, which has a significant impact on perception, affecting user studies and ratings.

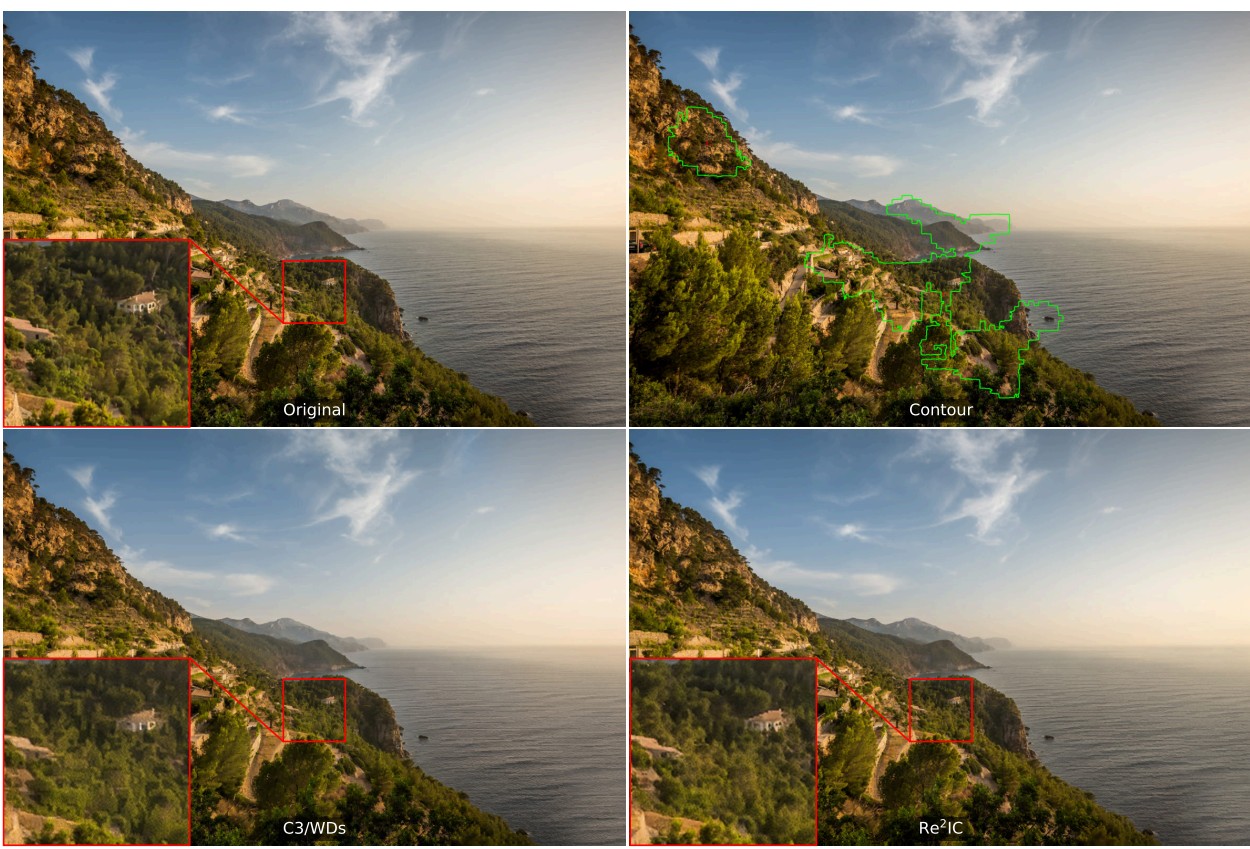

*Figure 32.* Visual comparison at low-bpp regime ($\leq 0.075$). C3/WDs tends to focus on fidelity in salient regions dominated by high frequencies, assigning lower $\sigma$ to salient areas while ignoring internal frequency characteristics. As a result, it struggles to capture high-frequency details under limited bit budgets, reducing perceptual quality.

