# OpenReview forum: "Frequency-Aware Perceptual Optimization for Low-Complexity Implicit Image Compression"
_ICML.cc/2026/Conference — ICML 2026 regular_

### Official Review · Reviewer_jS9x · 2026-03-05

**Soundness:** 3
**Presentation:** 3
**Significance:** 3
**Originality:** 3
**Overall Recommendation:** 5
**Confidence:** 3

**Summary:**

The paper proposes Re2IC, a frequency-aware perceptual optimization framework for low-complexity image compression. It consists five parts: region partition, adaptive chain coding, local-global perceptual network, latent vectors and a shared entropy model. Experiments on Kodak and CLIC2020 report state-of-the-art rate perception trade-off among different codecs, and competitive performance against baseline models.

**Compliance With Llm Reviewing Policy:**

Affirmed.

**Final Justification:**

The rebuttal has addressed my concerns. I give my score of 5.

**Key Questions For Authors:**

Authors are suggested to analyze weaknesses carefully.

**Limitations:**

Yes

**Strengths And Weaknesses:**

Strengths:
The motivation of this paper is clear and compelling. The trade-off between rate, distortion, perception and computing complexity is very important in image compression field. The motivation is well argued in introduction. The proposed Re2IC method offers an insightful bridge between human vision and machine compression. The transition from traditional machine compression to biologically-inspired adaptive framework breaks the boundary between human biological perception and machine-based perception.

Weaknesses:
1. Experimental results demonstrate a performance gap between perceptual metrics and low-level fidelity metrics. While the proposed Re2IC achieves state-of-the-art results on perceptual metrics, the PSNR and MS-SSIM is noticeably lower than other methods like VTM and MLIC+. Please explain the reason.
2. While the theoretical derivation is mathematically rigorous and well presented, I have concerns regarding the overall balance of the manuscript’s structure. Currently, the theoretical part occupies too much space in the main text, leaving only 2 pages for experiment results. It is suggested to rearrange the main text and appendix, moving more quantitative and visualization results to main text.

---

> ### Author Rebuttal · Authors · 2026-03-31
>
> We thank the reviewer for the valuable feedback. Next, we respond to the identified weaknesses and suggestions point by point.
>
> - W1.
> To address this comment, we will incorporate the following statement into the revised manuscript:
>
>   ``The observed gap arises from the **inherent rate–distortion–perception (RDP) trade-off** [1]: at a fixed rate, improving perceptual quality necessarily degrades distortion. This phenomenon has been widely observed in prior perceptual codecs (e.g., C3/WDs, HiFiC).
>
>   Distortion-oriented codecs, such as VTM and MLIC++, are optimized for pixel-wise fidelity (e.g., MSE), which leads to strong PSNR performance but can result in over-smoothed textures under limited bitrates. In contrast, Re2IC emphasizes perceptual quality through distribution alignment and region-aware bit allocation, preserving salient structures while allowing texture resampling in less important regions. This yields more realistic visual quality at the cost of reduced pixel-wise fidelity, leading to lower PSNR/MS-SSIM.''
>
>   [1] Blau, Yochai, and Tomer Michaeli. "Rethinking lossy compression: The rate-distortion-perception tradeoff." International Conference on Machine Learning, 2019.
>
> - W2. We thank the reviewer for this constructive suggestion. In the revision, we will **expand the experimental section**. In particular, if additional page space becomes available in a later revision, we will include more quantitative and visual results in the main text, including Kodak results (**Table 5** of the appendix), bit-allocation analysis (**Fig. 12** of the appendix), ablation studies, and additional visual comparisons (**Fig. 30** of the appendix).
>
>   Meanwhile, we will streamline parts of the theoretical section and move less critical derivations (Corollary  3.4 and Lemma 3.2 of the main text) to the appendix to improve readability. We will also provide clearer **references to the open-sourced supplementary material**, where full comparisons and visual results are presented.

---

> > ### Author Rebuttal · Reviewer_jS9x · 2026-04-02
> >
> > I would keep my original score of 5.

---

### Official Review · Reviewer_vZbF · 2026-03-10

**Soundness:** 3
**Presentation:** 3
**Significance:** 2
**Originality:** 3
**Overall Recommendation:** 5
**Confidence:** 4

**Summary:**

This paper proposes an INR-based, low-complexity image codec that enables optimization for multiple rate-distortion-perception (RDP) trade-offs. Overall, it presents a substantial and highly practical improvement over the C3/WD baseline.

**Compliance With Llm Reviewing Policy:**

Affirmed.

**Final Justification:**

This work presents an effective frequency-aware, low-complexity approach to implicit image compression. I appreciate the substantial effort reflected in the work, the clear and well-organized presentation, and the release of source code, which greatly facilitates reproducibility and follow-up research. The experimental results also demonstrate reasonable performance gains. Considering the overall quality, including its significance and originality, I believe this work solidly meets the standard of a borderline accept. I have therefore raised my rating to 5 (Weak Accept).

**Key Questions For Authors:**

In general, it is a good paper, but I have the following questions:
1. It is unclear whether the saliency-guided coding scheme consistently guarantees perceptual quality gains. Specifically, if an encoded image lacks a clear focal point or salient object, does this region partitioning approach actually degrade performance?
2. Could you provide a direct comparison of the end-to-end encoding time for high-resolution images, such as those in the CLIC2020 dataset?
3. You mention that 1 or 2 sub-regions are typically sufficient. However, what is the RD/RP impact if N is forced higher (e.g., N=5)? Does the chain-coding and network parameter overhead lead to a sharp collapse in performance in such edge cases?
4. I recommend including a comparison with more recent generative methods, such as Generative Latent Coding (GLC) [1].
5. I am curious about the performance of the MSE-optimized Re2IC on higher-resolution datasets. Could the authors provide a direct RD performance comparison against VTM?

[1] Jia Z, Li J, Li B, et al. Generative latent coding for ultra-low bitrate image compression, CVPR 2024.

**Limitations:**

yes

**Strengths And Weaknesses:**

Strengths:
1. The paper features comprehensive experiments and demonstrates excellent decoding efficiency (low MACs).
2. The introduction of saliency-guided region partitioning is a novel and effective approach for implicit codecs.

Weaknesses:
1. The encoding complexity and time overhead remain significant concerns for practical deployment.
2. Figure 12 shows that “networks + contours” consume a substantial fixed proportion of the bit budget. It is unclear how gracefully the method degrades at ultra-low bitrates (e.g., < 0.05 bpp) where this fixed overhead might dominate the bitstream.

---

> ### Author Rebuttal · Authors · 2026-03-31
>
> We thank the reviewer for the valuable feedback. Accordingly, we address the commented weakness and the question point by point, and will incorporate the following discussion into the revised manuscript.
>
> - W1. We acknowledge this limitation, noting that high encoding complexity is inherent to **all** implicit codecs, **not unique** to Re2IC. As such, it is particularly suitable for massive on-demand streaming applications (encode once, decode many times). Prior work shows that meta-learning [1] and engineering optimizations [2] can yield 10x–100x speedups. In this context, Re2IC **highlights** an additional advantage: **accelerated convergence** via region-wise perception modeling (Fig.6 of manuscript). Combined with future optimizations, this represents meaningful progress toward practical implicit codecs.
>
>   [1] "COIN++: Neural Compression Across Modalities." Transactions on Machine Learning Research.
>
>   [2] "Overfitted image coding at reduced complexity." EUSIPCO.
>
> - W2. In the standard bitrate regime, contour and parameter costs are small and thus not explicitly optimized in RP loss, though they become more noticeable at ultra-low bitrates. Both can be significantly compressed (lossy contours, smaller models) with limited RD/RP impact. More advanced strategies, such as Bayesian INR combined with channel simulation [3], could further reduce overhead. We also provide experiments in a lower bitrate regime (**Table R4**), showing controllable overhead and a preference for lightweight architectures.
>
>   [3] "Compression with bayesian implicit neural representations." NeurIPS.
>
> **Table R4 Different architectures on Kodak (1,2,3,6,7) at low bit-rate.**
> |BPP:0.058|Bit cost (latent/network/contour)|PSNR|MS-SSIM|LPIPS|DISTS|FID|NIQE|CLIPIQA|
> |-|-|-|-|-|-|-|-|-|
> |Arm-8,GPM-12|62.67%/35.16%/2.17%|22.95|0.7881|0.1886|0.1219|110.34|3.9678|0.5701|
> |Arm-16,GPM-12|45.74%/52.15%/2.11%|22.83|0.7781|0.1969|0.1245|111.20|3.8819|0.5062|
>
> Answers to questions:
>
> - Q1. Saliency-guided partitioning and WA-WD are tightly coupled. Without saliency, WA-WD reduces to uniform weighting (i.e., MSE), thereby reducing to fidelity optimization. To address the question, we evaluate region partitioning by comparing single-region and multi-region cases in this setting (**Table R5**). Results show improved performance from region-wise modeling even in the case without saliency.
>
> **Table.R5 Single-region vs. Multi-region on Kodak (3,4,10,12,14) without saliency.**
> |Region setting|Lambda|8e-3|4e-3|2e-3|1e-3|5e-4|3e-4|1e-4|BD-rate (vs one region)|
> |-|-|-|-|-|-|-|-|-|-|
> |One Region|PSNR (dB)|30.27|31.75|33.38|35.30|37.20|38.55|41.47|0%|
> | |Rate (bpp)|0.105|0.162|0.249|0.373|0.548|0.712|1.173|  |
> |Two Region|PSNR (dB)|30.33|31.83|33.48|35.34|37.24|38.67|41.63|-1.91%|
> |  |Rate (bpp)|0.108|0.165|0.250|0.377|0.542|0.701|1.145| |
> |Three Region|PSNR (dB) |30.34|31.85|33.51|35.36|37.29|38.70|41.65|-2.53%|
> ||Rate (bpp)|0.110|0.167|0.248|0.377|0.545|0.698|1.131|  |
>
> - Q2. We report encoding latency for CLIC2020 images in **Table R6**. We can observe that latency increases significantly with resolution, highlighting the importance of our accelerated convergence (key advantage of Re2IC). The reported results are based on an unoptimized code, where region-wise coding can be parallelized and VGG can be lightweighted for substantial speedups.
>
> **Table R6 Encoding cost on CLIC2020 using NVIDIA 3090 and Intel i9.**
> |Method|Encoding cost|
> |-|-|
> |VTM-19.1|394.59 s (CPU)|
> |EVC (S/M/L)|59.70/70.78/128.49 ms (GPU)|
> |MLIC++|495.19 ms (GPU)|
> |C3/WDs (1k)|257.59 s/1k steps (GPU)|
> |Re2IC (1k)|361.74 s/1k steps (GPU)|
>
> - Q3. The choice of 1–2 regions is motivated by both perceptual intuition and empirical evidence: natural images typically contain only a few salient regions, and 1–2 regions already yield significant RP gains with lower computational cost.
>
>   We further provide ablation studies over the number of regions (please see **Table R1** in our response to [G85m]). Results show that increasing $N$ does not cause notable degradation, where the overhead from contours and parameters remains limited.
>
> - Q4. We thank the reviewer for highlighting this recent work. GLC targets an ultra-low bitrate regime, which lies outside our evaluation scope. Following standard CLIC protocols, we evaluate at 0.75/0.3/0.15 bpp to ensure fair user studies and quantitative comparisons, consistent with C3/WDs. For completeness, we include GLC in the related work and highlight the fundamental architectural differences.
>
> - Q5. The performance of MSE-optimized Re2IC (ARM24) on CLIC2020 dataset is reported in **Table R7**, showing competitive results. Note that the region partitioning is based on saliency from EML-Net, leaving room for further optimization.
>
> **Table R7 MSE-optimized Re2IC on CLIC2020.**
> |Lambda|1e-2|5e-3|1e-3|2e-4|1e-4|BD-rate (vs VTM)|
> |-|-|-|-|-|-|-|
> |PSNR (dB)|30.63|32.08|35.93|39.69|41.30|-7.62%|
> |Rate (bpp)|0.088|0.142|0.359|0.796|1.084| |

---

> > ### Author Rebuttal · Reviewer_vZbF · 2026-04-01
> >
> > I thank the authors for their detailed response and thorough efforts during the rebuttal. All of my concerns have been adequately addressed. Accordingly, **I have decided to raise my score to 5 (Accept)**.

---

### Official Review · Reviewer_G85m · 2026-03-12

**Soundness:** 3
**Presentation:** 3
**Significance:** 2
**Originality:** 2
**Overall Recommendation:** 4
**Confidence:** 4

**Summary:**

This paper proposes Re2IC, an implicit neural representation (INR) based image compression method designed to improve perceptual quality at low complexity. The approach partitions an image into salient regions and models each region using a local neural network, combined with a global perceptual modulator. The authors also introduce wavelet–Wasserstein distortion (WA-WD), which decomposes perceptual loss into frequency bands to allow finer control over the tradeoff between realism and fidelity. Experiments compare perceptual quality to several existing implicit codecs and some generative approaches.

**Compliance With Llm Reviewing Policy:**

Affirmed.

**Final Justification:**

The authors addressed most of my primary concerns in their rebuttal. I hope they will include the new results on complexity and comparisons to additional methods in the camera-ready version.

**Key Questions For Authors:**

1. Which component contributes most to the improvements; region partitioning or wavelet–Wasserstein distortion? A more targeted ablation would help clarify this.
2. Since the codec still overfits a model per image, how does encoding time compare with other implicit codecs in practice?
3. Could the authors clarify how computational budgets can be compared when evaluating against generative perceptual codecs?

**Limitations:**

Yes

**Strengths And Weaknesses:**

Strengths:
- The paper addresses an important challenge with a clear motivation: improving perceptual quality in implicit codecs while keeping the decoding complexity low. The combination of region-based modeling and frequency-aware perceptual optimization is intuitive and well motivated by human visual perception.
- The authors provide extensive experiments, including several perceptual metrics and user studies.
- The proposed wavelet–Wasserstein distortion is a well thought-out extension to Wasserstein-based perceptual losses and it seems to correlate well with human preference.
- The paper is well written and the method is described clearly

Weaknesses:
- Many components of the method build on existing ideas (region partitioning, Wasserstein distortion, wavelet decompositions, local/global modeling). The contribution mainly lies in combining these elements rather than introducing a fundamentally new compression paradigm.
- The method contains multiple components (region partitioning, perceptual modulation, wavelet distortion), but it is difficult to determine which part contributes most to the reported gains. Ablation studies do show incremental improvements from MSE → WD → region coding → WA-WD. But there is no clear test with region partitioning removed but WA-WD retained, no sensitivity analysis for number of regions and little discussion of saliency quality.
- Evaluation setup can be clearer. The comparisons with generative perceptual codecs are a little difficult to interpret since these approaches operate under different assumptions and computational budgets. Further, comparisons with other implicit codecs that explicitly optimize perceptual objectives are limited.

---

> ### Author Rebuttal · Authors · 2026-03-31
>
> We thank the reviewer for the valuable feedback. Accordingly, we address the commented weaknesses and the questions point by point.
>
> - W1. We agree that our method builds on several established components. However, the contribution goes beyond their straightforward combination by introducing a **new implicit-compression perspective**. Region-based coding for arbitrary shapes is previously impractical in transform coding due to the lack of suitable transforms for irregular regions. INRs uniquely support this, and coupling region-wise INR coding with WA-WD yields a frequency-aware perceptual optimization that can be independently tuned per region; something no prior system achieves. By shifting to region-wise perception modeling, R2IC provides: (1) improved perceptual quality with lower decoding complexity, surpassing generative methods (HiFiC); (2) a general, interpretable metric (WA-WD) with standalone value (IQA); and (3) faster convergence, alleviating a core bottleneck of implicit codecs.
>
> - W2. Regarding the experiment of WA-WD without region partitioning, we refer to **Table 10** (line 1623 of the manuscript), which reports the **effect of each module** (with darker colors indicating larger impact). The results show that removing region partitioning leads to a larger performance drop than removing WA-WD, highlighting its critical role.
>
>   Regarding sensitivity to the region number and saliency quality, we conduct experiments with varying region numbers. The results in **Table R1** exhibit limited sensitivity: 2–3 regions can achieve satisfactory performance (most images contain no more than 3 meaningful salient regions). To assess robustness to saliency quality, we consider: (1) 5-region case (**Table R1**), where saliency becomes less distinct; and (2) lossy contours case (line 409 of the manuscript), introducing non-salient parts. In both cases, we observe only minor performance degradation. These results suggest that the method is robust to imperfect saliency and region number. We will include these analysis in the revised manuscript.
>
> - W3. The reviewer is correct that generative perceptual codecs operate under different computational budgets. We include them to contextualize the RP performance of implicit codecs relative to current leading methods. To improve clarity, we now report their computational budgets (**Table R2**) and clarify intended scope of these comparisons in the revised manuscript.
>
>   Regarding comparisons with other perceptual implicit codecs, we currently adopt C3/WDs, the current strongest perceptual codecs, as the primary baseline. For other implicit codecs (e.g., COIN/COIN++, Cool-Chic), directly optimizing perceptual objectives typically requires substantial architectural changes or additional mechanisms (e.g., common randomness, network architecture). As such, fair comparison would involve non-trivial redesign beyond their original formulations. We therefore focus on existing perceptual baselines and will clarify this point in the revised manuscript.
>
> **Table.R1 Quantitative comparison across different numbers of regions on Kodak (1,3,5,8,21).**
> ||Region|BPP|PSNR|MS-SSIM|LPIPS|DISTS$\downarrow$| FID$\downarrow$|NIQE$\downarrow$|CLIPIQA$\uparrow$|
> |-|--|-|-|-|-|-|-|-|-|
> ||2|0.135|22.8264|0.8831|0.1489|0.1018|65.4325|4.2945|0.5170|
> |Low|3|0.136|22.7360|0.8843|0.1436|0.1012|66.0363| 4.2786|0.5197|
> ||5|0.137|22.8128|0.8841|0.1462|0.0982|66.7724|4.1707|0.5156|
> ||2|0.221|24.1877|0.9209|0.1000| 0.0615|41.6331|3.9186|0.5657|
> |Medium|3|0.217|24.3468|0.9216|0.0965|0.0638|39.3924|4.0387|0.5759|
> ||5|0.221|24.3107|0.9221| 0.0973|0.0587|43.1441|3.9202|0.5809|
> ||2|0.323|25.6067|0.9443|0.0702| 0.0406|26.7772|4.2079|0.5688|
> |High|3|0.326|25.6039|0.9449|0.0704| 0.0419|26.6076|4.0804|0.5840|
> ||5|0.329|25.4517|0.9440|0.0719|0.0388|29.7044|4.0319|0.5978|
>
> **Table. R2 Model size and decoding time of diffusion baseline on Kodak (NVIDIA 3090 GPU).**
> |  |CDC (1 step)|CDC (17 steps)|CDC (65 steps)|
> |-|-|-|-|
> |Parameters (M)|53.8|53.8|53.8|
> |Decoding Time (ms)|85.0845|1132.0039|4272.1439|
>
> Answers to questions:
>
> - Q1. We refer to our response to W2 and **Table 10**. All components contribute significantly, with common randomness having the largest impact, followed by region partitioning and then WA-WD.
>
> - Q2. We report the encoding time of implicit codecs in **Table R3**. Re2IC incurs slightly higher per-step cost (but requires fewer steps). Its runtime scales well with region count. Perceptual optimization (vs. MSE) is slower due to VGG-based computations. Our Re2IC is based on an unoptimized implementation and can be further improved (see response to W1 of [vZbf]).
>
> - Q3. We refer the reviewer to our response to W3 (**Table R2**), with computational budgets reported.
>
> **Table. R3 GPU encoding time of different implicit codecs on Kodak.**
> |Implicit codecs|sec/1k steps|
> |-|-|
> |Cool-Chic (mse/WDs) |17.87/69.98|
> |C3 (mse/WDs) |21.59/70.24|
> |Re2IC (mse/WA-WD)|26.72/107.29|

---

> > ### Author Rebuttal · Reviewer_G85m · 2026-04-03
> >
> > I thank the authors for their detailed response, most of my primary concerns have been addressed.  I hope the authors will include the new complexity results and comparisons in the camera-ready version. I will maintain my rating.

---

### Decision · Program_Chairs · 2026-04-30

**Decision:**

Accept (regular)

**Comment:**

After the rebuttal and discussion, the reviewers converged toward a positive recommendation. Therefore, the AC agrees that the paper should be accepted. We encourage the authors to incorporate the additional rebuttal results into the camera-ready version.